# From Lazy to Rich: Exact Learning Dynamics in Deep Linear Networks

**Clémentine C. J. Dominé**[1,†]**, Nicolas Anguita**[2,†]**, Alexandra M. Proca**[2]**, Lukas Braun**[3]**,**
**Daniel Kunin**[4]**, Pedro A. M. Mediano**[2,5,‡]**, Andrew M. Saxe**[1,6,7,‡]

[1] Gatsby Computational Neuroscience Unit, University College London, UK

[2] Department of Computing, Imperial College London, UK

[3] Department of Experimental Psychology, University of Oxford, UK

[4] Institute for Computational and Mathematical Engineering, Stanford University, USA

[5] Division of Psychology and Language Sciences, University College London, UK

[6] Sainsbury Wellcome Centre, University College London, UK

[7] CIFAR Azrieli Global Scholar, CIFAR, Toronto, Canada

† Co-first author

‡ Co-senior author.

`clementine.domine.20@ucl.ac.uk, na658@cam.ac.uk`

## Abstract

Biological and artificial neural networks develop internal representations that enable them to perform complex tasks. In artificial networks, the effectiveness of these models relies on their ability to build task-specific representations, a process influenced by interactions among datasets, architectures, initialization strategies, and optimization algorithms. Prior studies highlight that different initializations can place networks in either a *lazy* regime, where representations remain static, or a *rich* (or *feature-learning*) regime, where representations evolve dynamically. Here, we examine how initialization influences learning dynamics in deep linear neural networks, deriving exact solutions for '*λ-balanced*' initializations—defined by the relative scale of weights across layers. These solutions capture the evolution of representations and the Neural Tangent Kernel across the spectrum from the *rich* to the *lazy* regimes. Our findings deepen the theoretical understanding of the impact of weight initialization on learning regimes, with implications for continual learning, reversal learning, and transfer learning, relevant to both neuroscience and practical applications.

## 1 Introduction

Biological and artificial neural networks learn internal representations that enable complex tasks such as categorization, reasoning, and decision-making. Both systems often develop similar representations from comparable stimuli, suggesting shared information processing mechanisms (Yamins et al., 2014). This similarity, though not fully understood, has drawn interest from neuroscience, AI, and cognitive science (Haxby et al., 2001; Laakso & Cottrell, 2000; Morcos et al., 2018; Kornblith et al., 2019; Moschella et al., 2022). The success of neural models relies on their ability to extract relevant features from data to build internal representations, a complex process that in machine learning is defined by two regimes: *lazy* and *rich* (Saxe et al., 2014; Pennington et al., 2017; Chizat et al., 2019; Bahri et al., 2020).

**Lazy regime.** The *lazy* regime follows from a fundamental phenomenon in overparameterized neural networks: during training, these networks frequently remain near their linearized form, undergoing minimal changes in the parameter space (Chizat et al., 2019). Consequently, they adopt learning dynamics akin to kernel regression, characterized by the Neural Tangent Kernel (NTK) matrix and exhibiting exponential learning behavior (Du et al., 2018; Jacot et al., 2018; Du et al., 2019; Allen-Zhu et al., 2019a;b; Zou et al., 2020). This behavior, known as the *lazy* or kernel regime, typically occurs in infinitely wide architectures and can be triggered by large variance initialization (Jacot

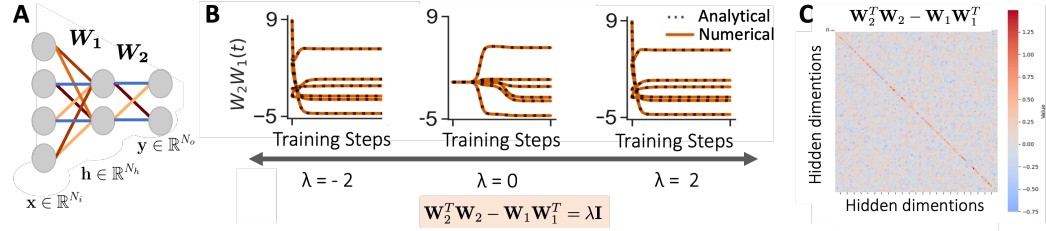

Figure 1: A minimal model of the *rich* and *lazy* regimes. **A.** We examine a deep and wide linear network trained using gradient descent starting from an initialization characterized by a relative scale parameter $\lambda$ — which characterizes the difference in the weight covariance between the first and second layers. **B.** Network output for an example task over training time, starting from a range of relative scale values. The dynamics are influenced by the initialization. Solid lines represent simulations, while dotted lines indicate the analytical solutions derived in this work. **C.** A network with LeCun weight initialization (LeCun et al., 1998) in the infinite width limit becomes $\lambda$-balanced, as $\mathbf{W}_2^T\mathbf{W}_2 - \mathbf{W}_1\mathbf{W}_1^T$ approaches the scaled identity matrix.

et al., 2018; Chizat et al., 2019). While the *lazy* regime offers valuable insights into how networks converge to a global minimum, it does not fully account for the generalization capabilities of neural networks. It is, therefore, widely believed that another regime, driven by small or vanishing initializations, underpins some of the successes of neural networks.

**Rich regime.** In contrast, the *rich* feature-learning regime is characterized by a NTK that evolves throughout training, accompanied by non-convex dynamics that navigate saddle points (Baldi & Hornik, 1989; Saxe et al., 2014; 2019b; Jacot et al., 2021). This regime features sigmoidal learning curves and simplicity biases, such as low-rankness (Li et al., 2020) or sparsity (Woodworth et al., 2020). Numerous studies have shown that the *absolute scale* of initialization drives the *rich* regime, which typically emerges at small initialization scales (Chizat et al., 2019; Geiger et al., 2020). However, even at small initialization scales, differences in weight magnitudes between layers can induce the *lazy* learning regime (Azulay et al., 2021; Kunin et al., 2024) – highlighting the significance of both *absolute scale* (initialization variance) and *relative scale* (difference in weight magnitude between layers) in generating diverse learning dynamics (Atanasov et al., 2022). Beyond *absolute scale* and *relative scale*, additional aspects of initialization can profoundly affect feature learning, including the effective rank of the weight matrices Liu et al. (2023b), layer-specific initialization variances Yang & Hu (2020); Luo et al. (2021); Yang et al. (2022), and the use of large learning rates Lewkowycz et al. (2020); Ba et al. (2022); Zhu et al. (2023); Cui et al. (2024). These findings illustrate the effect of initialization on inducing complex learning behavior through the resulting dynamics. Here we develop a solvable model which captures these diverse phenomena.

Despite significant advances, these learning regimes and their characterization are not yet fully understood and would benefit from clearer theoretical predictions, particularly regarding the influence of prior knowledge (initialization) on the learning regime. In this work, we address this gap by deriving exact solutions for the learning dynamics in deep linear networks as a function of network initialization, providing one of the few analytical models of the *rich* and *lazy* regimes in wide and deep neural networks (Xu & Ziyin, 2024; Kunin et al., 2024; Tu et al., 2024). To illustrate the dramatic effect of initialization and the kind of phenomenon we build a theory for, we consider a two-layer linear network parameterized by an encoding layer $\mathbf{W}_1$ and a decoding layer $\mathbf{W}_2$ (Fig.1A). This network can be initialized with different relative scalings, such that $\mathbf{W}_1\mathbf{W}_1^T \succ \mathbf{W}_2^T\mathbf{W}_2$, $\mathbf{W}_1\mathbf{W}_1^T \prec \mathbf{W}_2^T\mathbf{W}_2$, or $\mathbf{W}_1\mathbf{W}_1^T = \mathbf{W}_2^T\mathbf{W}_2$, while maintaining the same absolute scale. As shown in Fig.1B, the choice of relative scaling can result in drastically different learning trajectories and representations and the theory we develop over the course of this paper describes these effects. Through these solutions, we aim to gain insights into the *rich* and *lazy* regimes, as well as the transition between them during training, by examining the impact of relative scaling. As shown in Fig.1C and further proved in Appendix A.3, initialization methods used in practice, such as Le-Cun initialization in wide networks, approximate the relative scaling initialization explored in this paper, making it relevant to machine learning community as further demonstrated by Kunin et al. (2024). We consider applications relevant to machine learning and neuroscience, including continual learning (Kirkpatrick et al., 2017; Zenke et al., 2017; Parisi et al., 2019), reversal learning (Erdeniz

& Atalay, 2010) and transfer learning (Taylor & Stone, 2009; Thrun & Pratt, 2012; Lampinen & Ganguli, 2018; Gerace et al., 2022).

**Our contributions.**

- We derive explicit solutions for the gradient flow, internal representational similarity, and finite-width NTK in unequal-input-output two-layer deep linear networks, under a broad range of $\lambda$-balanced initialization conditions (Section 4).

- We model the full range of learning dynamics from *lazy* to *rich*, showing that this transition is influenced by a complex interaction of architecture, *relative scale*, and *absolute scale*, extending beyond just initialization *absolute scale* (Section 5).

- We present applications of these solutions relevant to both the neuroscience and machine learning communities, providing exact solutions for continual learning dynamics, reversal learning dynamics, and transfer learning (Section 6).

## 2 RELATED WORK

**Linear networks.** Our work builds upon a rich body of research on deep linear networks, which, despite their simplicity, have proven to be valuable models for understanding more complex neural networks (Baldi & Hornik, 1989; Fukumizu, 1998; Saxe et al., 2014). Previous research has extensively analyzed convergence (Arora et al., 2018a; Du & Hu, 2019), generalization properties (Lampinen & Ganguli, 2018; Poggio et al., 2018; Huh, 2020), and the implicit bias of gradient descent (Arora et al., 2019a; Woodworth et al., 2020; Chizat & Bach, 2020; Kunin et al., 2022) in linear networks. These studies have also revealed that deep linear networks have intricate fixed point structures and nonlinear learning dynamics in parameter and function space, reminiscent of phenomena observed in nonlinear networks (Arora et al., 2018b; Lampinen & Ganguli, 2018). Seminal work by Saxe et al. (2014) laid the groundwork by providing exact solutions to gradient flow dynamics under task-aligned initializations, demonstrating that the largest singular values are learned first during training. This analysis has been extended to deep linear networks (Arora et al., 2018b; 2019a; Ziyin et al., 2022) with more flexible initialization schemes (Gidel et al., 2019; Tarmoun et al., 2021; Gissin et al., 2019). This work directly builds on the matrix Riccati formulation proposed by Fukumizu (1998) and Braun et al. (2022) which extends these solutions to wide networks. We extend and refine these results to obtain the dynamics for a wider class of $\lambda$-balanced networks to more clearly demonstrate the impact of initialization on *rich* and *lazy* learning regimes also developed in Tu et al. (2024) for a set of orthogonal initalizations. Our work extends previous analyses (Xu & Ziyin, 2024; Kunin et al., 2024) of these regimes to wide networks. Previous studies leveraged these solutions primarily to characterize convergence rates; however, our work goes beyond this by providing a comprehensive characterization of the complete dynamics of the system (Tarmoun et al., 2021).

**Infinite-width networks.** Recent advances in understanding the *rich* regime have largely stemmed from examining how the initialization variance and layer-wise learning rates must scale in the infinite-width limit to maintain consistent behavior in activations, gradients, and outputs. Several studies have employed statistical mechanics tools to derive analytical solutions for the *rich* population dynamics of two-layer nonlinear neural networks initialized using a *mean-field* parameterization (Mei et al., 2018; Rotskoff & Vanden-Eijnden, 2018; Chizat & Bach, 2018; Sirignano & Spiliopoulos, 2020; Rotskoff & Vanden-Eijnden, 2022; Sirignano & Spiliopoulos, 2020). Other methods for analyzing deep network dynamics include the NTK limit, where the network effectively performs kernel regression without feature learning (Jacot et al., 2018; Lee et al., 2019; Arora et al., 2019b). Furthermore, these approaches typically require numerical integration or operate within a limited learning regime, and are unable to describe the learning dynamics of hidden representations. Instead, our work provides explicit analytical solutions for the dynamics of the network and its NTK in the finite-width case (Jacot et al., 2021; Xu & Ziyin, 2024; Kunin et al., 2024; Chizat et al., 2019).

## 3 PRELIMINARIES

Consider a supervised learning task where input vectors $\mathbf{x}_n \in \mathbb{R}^{N_i}$, from a set of $P$ training pairs $\{(\mathbf{x}_n, \mathbf{y}_n)\}_{n=1}^{P}$, need to be mapped to their corresponding target output vectors $\mathbf{y}_n \in \mathbb{R}^{N_o}$. We learn this task with a two-layer linear network model that produces the output prediction

$$\hat{\mathbf{y}}_n = \mathbf{W}_2 \mathbf{W}_1 \mathbf{x}_n, \tag{1}$$

with weight matrices $\mathbf{W}_1 \in \mathbb{R}^{N_h \times N_i}$ and $\mathbf{W}_2 \in \mathbb{R}^{N_o \times N_h}$, where $N_h$ is the number of hidden units. The network's weights are optimized using full batch gradient descent with learning rate $\eta$ (or respectively time constant $\tau = \frac{1}{\eta}$) on the mean squared error loss $\mathcal{L}(\hat{\mathbf{y}}, \mathbf{y}) = \frac{1}{2} \langle ||\hat{\mathbf{y}} - \mathbf{y}||^2 \rangle$, where $\langle \cdot \rangle$ denotes the average over the dataset. Our objective is to describe the entire dynamics of the network's output and internal representations based on the input covariance and input-output cross-covariance matrices of the dataset, defined as

$$\tilde{\mathbf{\Sigma}}^{xx} = \frac{1}{P} \sum_{n=1}^{P} \mathbf{x}_n \mathbf{x}_n^T \in \mathbb{R}^{N_i \times N_i} \quad \text{and} \quad \tilde{\mathbf{\Sigma}}^{yx} = \frac{1}{P} \sum_{n=1}^{P} \mathbf{y}_n \mathbf{x}_n^T \in \mathbb{R}^{N_o \times N_i}, \tag{2}$$

and the initialization $\mathbf{W}_2(0), \mathbf{W}_1(0)$. We employ an approach first introduced in the foundational work of Fukumizu (1998) and extended in recent work by Braun et al. (2022), which instead of studying the parameters directly, considers the dynamics of a matrix of the important statistics. In particular, defining $\mathbf{Q} = \begin{bmatrix} \mathbf{W}_1 & \mathbf{W}_2^T \end{bmatrix}^T \in \mathbb{R}^{(N_i + N_o) \times N_h}$, we consider the $(N_i + N_o) \times (N_i + N_o)$ matrix

$$\mathbf{Q}\mathbf{Q}^T(t) = \begin{bmatrix} \mathbf{W}_1^T \mathbf{W}_1(t) & \mathbf{W}_1^T \mathbf{W}_2^T(t) \\ \mathbf{W}_2 \mathbf{W}_1(t) & \mathbf{W}_2 \mathbf{W}_2^T(t) \end{bmatrix}, \tag{3}$$

which is divided into four quadrants with interpretable meanings, and where $t \in \mathbb{R}$ represents training time. The approach monitors several key statistics collected in the matrix. The off-diagonal blocks contain the network function $\hat{\mathbf{Y}}(t) = \mathbf{W}_2 \mathbf{W}_1(t) \mathbf{X}$. The on-diagonal blocks capture the correlation structure of the weight matrices, allowing for the calculation of the temporal evolution of the network's internal representations. This includes the representational similarity matrices (RSM) of the neural representations within the hidden layer, as first defined by Braun et al. (2022),

$$\text{RSM}_I = \mathbf{X}^T \mathbf{W}_1^T \mathbf{W}_1(t) \mathbf{X}, \quad \text{RSM}_O = \mathbf{Y}^T (\mathbf{W}_2 \mathbf{W}_2^T(t))^+ \mathbf{Y}, \tag{4}$$

where $+$ denotes the pseudoinverse; and the network's finite-width NTK (Jacot et al., 2018; Lee et al., 2019; Arora et al., 2019b)

$$\text{NTK} = \mathbf{I}_{N_o} \otimes \mathbf{X}^T \mathbf{W}_1^T \mathbf{W}_1(t) \mathbf{X} + \mathbf{W}_2 \mathbf{W}_2^T(t) \otimes \mathbf{X}^T \mathbf{X}, \tag{5}$$

where $\mathbf{I}_{N_o}$ is the $N_o \times N_o$ identity matrix and $\otimes$ is the Kronecker product. Hence, the dynamics of $\mathbf{Q}\mathbf{Q}^T$ describes the important aspects of network behaviour.

## 4 EXACT LEARNING DYNAMICS

We derive an exact solution for $\mathbf{Q}\mathbf{Q}^T$ offering insight into the learning dynamics, convergence behavior, and generalization properties of two-layer linear networks with prior knowledge.

**Assumptions.** To derive these solutions we make the following assumptions:

- **A1** (*Whitened input*). The input data is whitened, i.e. $\tilde{\mathbf{\Sigma}}^{xx} = \mathbf{I}$.
- **A2** ($\lambda$-*Balanced*). The network's weight matrices are $\lambda$-balanced at the beginning of training, i.e. $\mathbf{W}_2^T \mathbf{W}_2(0) - \mathbf{W}_1 \mathbf{W}_1(0)^T = \lambda \mathbf{I}$. If this condition holds at initialization, it will persist throughout training (Saxe et al., 2014; Arora et al., 2018a). For completeness, we prove this in Appendix A.
- **A3** (*Dimensions*). The hidden dimension of the network is defined as $N_h = \min(N_i, N_o)$, ensuring the network is neither bottlenecked ($N_h < \min(N_i, N_o)$) nor overparameterized ($N_h > \min(N_i, N_o)$).

These assumptions are strictly weaker than prior works (Fukumizu, 1998; Braun et al., 2022; Kunin et al., 2024; Xu & Ziyin, 2024). The main distinction between our work and prior works is that both Fukumizu (1998) and Braun et al. (2022) assumed zero-balanced weights ($\mathbf{W}_1(0)\mathbf{W}_1(0)^T = \mathbf{W}_2(0)^T \mathbf{W}_2(0)$), while we relax this assumption to $\lambda$-balanced. The zero-balanced condition restricts the networks to a *rich* setting. We develop solutions to explore the continuum between the *rich* and the *lazy* regime. While some works, such as Tarmoun et al. (2021), have considered removing this constraint, their solutions remain in an unstable and mixed form. Other studies, such as Xu & Ziyin (2024) and Kunin et al. (2024), have similarly relaxed the balanced assumption but were limited to single-output neuron settings. See Appendix A.2 for a further discussion on each of these works' assumptions and their relationship to ours.

**Lemmas and definitions.** To derive exact solutions we start by presenting the main lemmas which we prove in the appendix.

**Lemma 4.1.** *Under assumptions 1 and 2, the gradient flow dynamics of $\mathbf{Q}\mathbf{Q}^T(t)$, with initialization $\mathbf{Q}\mathbf{Q}^T(0) = \mathbf{Q}(0)\mathbf{Q}(0)^T$ can be written as a differential matrix Riccati equation*

$$\tau \frac{d}{dt}(\mathbf{Q}\mathbf{Q}^T) = \mathbf{F}\mathbf{Q}\mathbf{Q}^T + \mathbf{Q}\mathbf{Q}^T\mathbf{F} - (\mathbf{Q}\mathbf{Q}^T)^2, \quad where \; \boldsymbol{F} = \begin{pmatrix} -\frac{\lambda}{2}\mathbf{I}_{N_i} & (\tilde{\boldsymbol{\Sigma}}^{yx})^T \\ \tilde{\boldsymbol{\Sigma}}^{yx} & \frac{\lambda}{2}\mathbf{I}_{N_o} \end{pmatrix}. \quad (6)$$

As derived in Fukumizu (1998) and extended in Braun et al. (2022), whenever $\boldsymbol{F}$ is symmetric and diagonalizable such that $\boldsymbol{F} = \boldsymbol{P}\boldsymbol{\Lambda}\boldsymbol{P}^T$, where $\boldsymbol{P}$ is an orthonormal matrix and $\boldsymbol{\Lambda}$ is a diagonal matrix, then the unique solution to this matrix Riccatti equation is given by

$$\mathbf{Q}\mathbf{Q}^T(t) = e^{\mathbf{F}\frac{t}{\tau}}\mathbf{Q}(0)\left[\mathbf{I} + \mathbf{Q}(0)^T\boldsymbol{P}\left(\frac{e^{2\boldsymbol{\Lambda}\frac{t}{\tau}} - \mathbf{I}}{2\boldsymbol{\Lambda}}\right)\boldsymbol{P}^T\mathbf{Q}(0)\right]^{-1}\mathbf{Q}(0)^Te^{\mathbf{F}\frac{t}{\tau}}. \quad (7)$$

In Appendix B.2 we prove that this equation is the unique solution to the initial value problem derived in Lemma 4.1 for any value of $\Lambda$. However, as discussed in Braun et al. (2022), the solution in this form is not very useable or interpretable due to the matrix inverse mixing the blocks of $\mathbf{Q}\mathbf{Q}^T$. Additionally, we need to diagonalize $\boldsymbol{F}$. To do so we consider the compact singular value decomposition $\text{SVD}(\tilde{\boldsymbol{\Sigma}}^{yx}) = \tilde{\mathbf{U}}\tilde{\mathbf{S}}\tilde{\mathbf{V}}^T$. Here, $\tilde{\mathbf{U}} \in \mathbb{R}^{N_o \times N_h}$ denotes the left singular vectors, $\tilde{\mathbf{S}} \in \mathbb{R}^{N_h \times N_h}$ the square matrix with ordered, non-zero eigenvalues on its diagonal, and $\tilde{\mathbf{V}} \in \mathbb{R}^{N_i \times N_h}$ the corresponding right singular vectors. For unequal input-output dimensions ($N_i \neq N_o$), the right and left singular vectors are not square. Accordingly, for the case $N_i > N_h = N_o$, we define $\tilde{\mathbf{U}}_\perp \in \mathbb{R}^{N_o \times |N_o - N_i|}$ as a matrix containing orthogonal column vectors that complete the basis for $\tilde{\mathbf{U}}$, i.e., make $[\tilde{\mathbf{U}} \; \tilde{\mathbf{U}}_\perp]$ orthonormal, and $\tilde{\mathbf{V}}_\perp \in \mathbb{R}^{N_i \times |N_o - N_i|}$ as a matrix of zeros. Conversely, when $N_i = N_h < N_o$, then $\tilde{\mathbf{V}}_\perp$ is a matrix containing orthogonal column vectors that complete the basis for $\tilde{\boldsymbol{V}}$ and $\tilde{\mathbf{U}}_\perp$ is a matrix of zeros. Using this SVD structure we can now describe the eigendecomposition of $\mathbf{F}$.

**Lemma 4.2.** *Under assumptions 3, the eigendecomposition of $\mathbf{F} = \mathbf{P}\boldsymbol{\Lambda}\mathbf{P}^T$ is*

$$\mathbf{P} = \frac{1}{\sqrt{2}}\begin{pmatrix} \tilde{\boldsymbol{V}}(\tilde{\boldsymbol{G}} - \tilde{\boldsymbol{H}}\tilde{\boldsymbol{G}}) & \tilde{\boldsymbol{V}}(\tilde{\boldsymbol{G}} + \tilde{\boldsymbol{H}}\tilde{\boldsymbol{G}}) & \sqrt{2}\tilde{\boldsymbol{V}}_\perp \\ \tilde{\boldsymbol{U}}(\tilde{\boldsymbol{G}} + \tilde{\boldsymbol{H}}\tilde{\boldsymbol{G}}) & -\tilde{\boldsymbol{U}}(\tilde{\boldsymbol{G}} - \tilde{\boldsymbol{H}}\tilde{\boldsymbol{G}}) & \sqrt{2}\tilde{\boldsymbol{U}}_\perp \end{pmatrix}, \quad \boldsymbol{\Lambda} = \begin{pmatrix} \tilde{\boldsymbol{S}}_\lambda & 0 & 0 \\ 0 & -\tilde{\boldsymbol{S}}_\lambda & 0 \\ 0 & 0 & \boldsymbol{\lambda}_\perp \end{pmatrix}, \quad (8)$$

*where the matrices $\tilde{\boldsymbol{S}}_\lambda$, $\boldsymbol{\lambda}_\perp$, $\tilde{\boldsymbol{H}}$, and $\tilde{\boldsymbol{G}}$ are diagonal matrices defined as:*

$$\tilde{\boldsymbol{S}}_\lambda = \sqrt{\tilde{\boldsymbol{S}}^2 + \frac{\lambda^2}{4}\mathbf{I}}, \quad \boldsymbol{\lambda}_\perp = \text{sgn}(N_o - N_i)\frac{\lambda}{2}\mathbf{I}_{|N_o-N_i|}, \quad \tilde{\boldsymbol{H}} = \text{sgn}(\lambda)\sqrt{\frac{\tilde{\boldsymbol{S}}_\lambda - \tilde{\boldsymbol{S}}}{\tilde{\boldsymbol{S}}_\lambda + \tilde{\boldsymbol{S}}}}, \quad \tilde{\boldsymbol{G}} = \frac{1}{\sqrt{\mathbf{I} + \tilde{\boldsymbol{H}}^2}}. \quad (9)$$

**Main theorem.** Thanks to the eigendecomposition of $\boldsymbol{F}$ we can separate the solution provided in equation 7 into four quadrants. Following an approach used in Braun et al. (2022), we will find it useful to define the following variables of the initialization that will allow us to define the product $\boldsymbol{P}^T\boldsymbol{Q}(0)$ more succinctly,

$$\mathbf{B} = \mathbf{W}_2(0)^T\tilde{\boldsymbol{U}}(\tilde{\boldsymbol{G}} + \tilde{\boldsymbol{H}}\tilde{\boldsymbol{G}}) + \mathbf{W}_1(0)\tilde{\boldsymbol{V}}(\tilde{\boldsymbol{G}} - \tilde{\boldsymbol{H}}\tilde{\boldsymbol{G}}) \in \mathbb{R}^{N_h \times N_h}, \quad (10)$$

$$\mathbf{C} = \mathbf{W}_2(0)^T\tilde{\boldsymbol{U}}(\tilde{\boldsymbol{G}} - \tilde{\boldsymbol{H}}\tilde{\boldsymbol{G}}) - \mathbf{W}_1(0)\tilde{\boldsymbol{V}}(\tilde{\boldsymbol{G}} + \tilde{\boldsymbol{H}}\tilde{\boldsymbol{G}}) \in \mathbb{R}^{N_h \times N_h}, \quad (11)$$

$$\boldsymbol{D} = \mathbf{W}_2(0)^T\tilde{\mathbf{U}}_\perp + \mathbf{W}_1(0)\tilde{\mathbf{V}}_\perp \in \mathbb{R}^{N_h \times |N_o - N_i|}. \quad (12)$$

Using these variables of the initialization, this brings us to our main theorem:

**Theorem 4.3.** *Under the assumptions of whitened inputs (1), $\lambda$-balanced weights (2), and no bottleneck (3), the temporal dynamics of $\mathbf{Q}\mathbf{Q}^T$ are*

$$\mathbf{Q}\mathbf{Q}^T(t) = \begin{pmatrix} \boldsymbol{Z_1}(t)\boldsymbol{A}^{-1}(t)\boldsymbol{Z_1^T}(t) & \boldsymbol{Z_1}(t)\boldsymbol{A}^{-1}(t)\boldsymbol{Z_2^T}(t) \\ \boldsymbol{Z_2}(t)\boldsymbol{A}^{-1}(t)\boldsymbol{Z_1^T}(t) & \boldsymbol{Z_2}(t)\boldsymbol{A}^{-1}(t)\boldsymbol{Z_2^T}(t) \end{pmatrix},$$

*with the time-dependent variables $\boldsymbol{Z_1}(t) \in \mathbb{R}^{N_i \times N_h}$, $\boldsymbol{Z_2}(t) \in \mathbb{R}^{N_o \times N_h}$, and $\boldsymbol{A}(t) \in \mathbb{R}^{N_h \times N_h}$:*

$$\boldsymbol{Z_1}(t) = \frac{1}{2}\tilde{\boldsymbol{V}}(\tilde{\boldsymbol{G}} - \tilde{\boldsymbol{H}}\tilde{\boldsymbol{G}})e^{\tilde{\boldsymbol{S}}_\lambda \frac{t}{\tau}}\boldsymbol{B}^T - \frac{1}{2}\tilde{\boldsymbol{V}}(\tilde{\boldsymbol{G}} + \tilde{\boldsymbol{H}}\tilde{\boldsymbol{G}})e^{-\tilde{\boldsymbol{S}}_\lambda \frac{t}{\tau}}\boldsymbol{C}^T + \tilde{\mathbf{V}}_\perp e^{\boldsymbol{\lambda}_\perp \frac{t}{\tau}}\boldsymbol{D}^T, \quad (13)$$

$$\boldsymbol{Z_2}(t) = \frac{1}{2}\tilde{\boldsymbol{U}}(\tilde{\boldsymbol{G}} + \tilde{\boldsymbol{H}}\tilde{\boldsymbol{G}})e^{\tilde{\boldsymbol{S}}_\lambda \frac{t}{\tau}}\boldsymbol{B}^T + \frac{1}{2}\tilde{\boldsymbol{U}}(\tilde{\boldsymbol{G}} - \tilde{\boldsymbol{H}}\tilde{\boldsymbol{G}})e^{-\tilde{\boldsymbol{S}}_\lambda \frac{t}{\tau}}\boldsymbol{C}^T + \tilde{\mathbf{U}}_\perp e^{\boldsymbol{\lambda}_\perp \frac{t}{\tau}}\boldsymbol{D}^T, \quad (14)$$

$$\boldsymbol{A}(t) = \mathbf{I} + \boldsymbol{B}\left(\frac{e^{2\tilde{\boldsymbol{S}}_\lambda \frac{t}{\tau}} - \mathbf{I}}{4\tilde{\boldsymbol{S}}_\lambda}\right)\boldsymbol{B}^T - \boldsymbol{C}\left(\frac{e^{-2\tilde{\boldsymbol{S}}_\lambda \frac{t}{\tau}} - \mathbf{I}}{4\tilde{\boldsymbol{S}}_\lambda}\right)\boldsymbol{C}^T + \boldsymbol{D}\left(\frac{e^{\boldsymbol{\lambda}_\perp \frac{t}{\tau}} - \mathbf{I}}{\boldsymbol{\lambda}_\perp}\right)\boldsymbol{D}^T. \quad (15)$$

The proof of Theorem 4.3 is in Appendix B. With this solution we can calculate the exact temporal dynamics of the loss, network function, RSMs and NTK (Fig. 2A, C) over a range of $\lambda$-balanced initializations.

**Implementation and simulation.** One issue with the expression we derived in Theorem 4.3 is that it can be numerically unstable when simulating it for long time $t \gg 0$ as it involves taking the inverse of terms that involve exponentials that are diverging with $t$. If we make the additional assumption that $\boldsymbol{B}$ is invertible, then we can rearrange this expression to only use exponentials with negative coefficients, which we derive in Appendix B.5. In the next section we will discuss the significance of $\boldsymbol{B}$ being invertible at initialization on the convergence of the dynamics. Simulation details are in Appendix E.7.

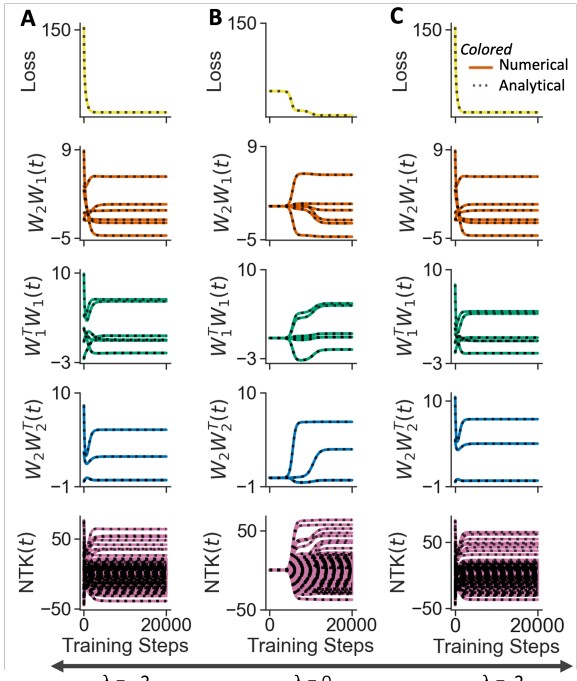

Figure 2: **A.** The temporal dynamics of the numerical simulation (colored lines) of the loss, network function, correlation of input and output weights, and the NTK (row 1-5 respectively) are exactly matched by the analytical solution (black dotted lines) for $\lambda = -2$. **B.** $\lambda = 0$ Large initial weight values. **C.** $\lambda = 2$ initial weight values initialized as described in E.7.

## 5   RICH AND LAZY LEARNING

Next, we use these solutions to gain a deeper understanding of the transition between the *rich* and *lazy* regimes by examining the dynamics as a function of $\lambda$ – the *relative scale* – as it varies between positive and negative infinity. We investigate four key indicators of the learning regimes: the dynamics of singular vectors, the structure and robustness of the representations, and the evolution of the NTK.

**Dynamics of the singular values.** Here we examine a $\lambda$-*balanced* linear network initialized with *task-aligned* weights. Previous research (Saxe et al., 2019a) has demonstrated that initial weights that are aligned with the task remain aligned throughout training, restricting the learning dynamics to the singular values of the network. This setting offers a valuable opportunity to build intuition about the impact of $\lambda$ on the dynamics of learning regimes, extending beyond previous solutions (Tarmoun et al., 2021; Varre et al., 2024).

**Theorem 5.1.** *Under the assumptions of Theorem 4.3 and with a task-aligned initialization, as defined in Saxe et al. (2013), the network function is given by the expression $\boldsymbol{W}_2\boldsymbol{W}_1(t) = \tilde{\boldsymbol{U}}\boldsymbol{S}(t)\tilde{\boldsymbol{V}}^T$ where $\boldsymbol{S}(t) \in \mathbb{R}^{N_h \times N_h}$ is a diagonal matrix of singular values with elements $s_\alpha(t)$ that evolve according to the equation,*

$$s_\alpha(t) = s_\alpha(0) + \gamma_\alpha(t; \lambda)\left(\tilde{s}_\alpha - s_\alpha(0)\right), \tag{16}$$

*where $\tilde{s}_\alpha$ is the $\alpha$ singular value of $\tilde{\boldsymbol{S}}$ and $\gamma_\alpha(t; \lambda)$ is a $\lambda$-dependent monotonic transition function for each singular value that increases from $\gamma_\alpha(0; \lambda) = 0$ to $\lim_{t \to \infty} \gamma_\alpha(t; \lambda) = 1$ defined explicitly in Appendix C.1. We find that under different limits of $\lambda$, the transition function converges pointwise to the sigmoidal ($\lambda \to 0$) and exponential ($\lambda \to \pm\infty$) transition functions,*

$$\lim_{\lambda \to 0} \gamma_\alpha(t; \lambda) \to \frac{e^{2\tilde{s}_\alpha \frac{t}{\tau}} - 1}{e^{2\tilde{s}_\alpha \frac{t}{\tau}} - 1 + \frac{\tilde{s}_\alpha}{s_\alpha(0)}}, \qquad \lim_{\lambda \to \pm\infty} \gamma_\alpha(t; \lambda) \to 1 - e^{-|\lambda|\frac{t}{\tau}}. \tag{17}$$

The proof for Theorem 5.1 can be found in Appendix C.1. As shown in Fig. 3B, as $\lambda$ approaches zero, the dynamics resemble sigmoidal learning curves that traverse between saddle points, characteristic of the *rich* regime (Braun et al., 2022). In this regime the network learns the most salient features first, which can be beneficial for generalization (Lampinen & Ganguli, 2018). Conversely, as shown in Fig. 3A and C, as the magnitude of $\lambda$ increases, the dynamics become exponential, characteristic of the *lazy* regime. In this regime, all features are treated equally and the network's

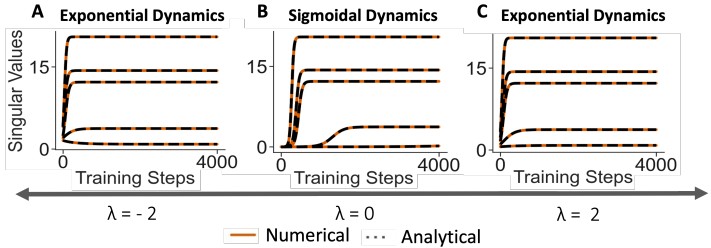

Figure 3: Simulated and analytical dynamics of the singular values of the network function with *relative scale* of **A.** $\lambda = -2$, **B.** $\lambda = 0$, or **C.** $\lambda = 2$, initialized as described in Appendix E.7.

dynamics resemble that of a shallow network. Notably, similar effects have been observed previously in the context of large *absolute scales* (Saxe et al., 2019a) independently of the *relative scale*. Overall, our results highlight the critical influence the *relative scale* $\lambda$ has in shaping the learning dynamics, from sigmoidal to exponential, steering the network between the *rich* and *lazy* regimes.

**The dynamics of the representations.** We now consider how the representations of the individual parameters $\boldsymbol{W}_1$ and $\boldsymbol{W}_2$ change through training. We note that under $\lambda$-balanced initializations there is a simple structure which persists throughout training that allows us to recover the dynamics of the parameters up to a time-dependent orthogonal transformation from the dynamics of $\mathbf{Q}\mathbf{Q}^T(t)$.

**Theorem 5.2.** *Under assumptions 2, if the network function* $\boldsymbol{W}_2\boldsymbol{W}_1(t) = \boldsymbol{U}(t)\boldsymbol{S}(t)\boldsymbol{V}^T(t)$ *is full rank, then we can recover the parameters* $\boldsymbol{W}_2(t) = \boldsymbol{U}(t)\boldsymbol{S}_2(t)\boldsymbol{R}^T(t)$ *and* $\boldsymbol{W}_1(t) = \boldsymbol{R}(t)\boldsymbol{S}_1(t)\boldsymbol{V}^T(t)$ *up to time-dependent orthogonal transformation* $\boldsymbol{R}(t) \in \mathbb{R}^{N_h \times N_h}$, *where* $\boldsymbol{S}_\lambda(t) = \sqrt{\boldsymbol{S}^2(t) + \frac{\lambda^2}{4}\mathbf{I}}$ *and*

$$\boldsymbol{S}_1(t) = \left( \left( \boldsymbol{S}_\lambda(t) - \frac{\lambda\mathbf{I}}{2} \right)^{\frac{1}{2}}, 0_{\max(0, N_i - N_o)} \right), \quad \boldsymbol{S}_2(t) = \left( \left( \boldsymbol{S}_\lambda(t) + \frac{\lambda\mathbf{I}}{2} \right)^{\frac{1}{2}}; 0_{\max(0, N_o - N_i)} \right). \quad (18)$$

The effective singular values $\boldsymbol{S}_\lambda$ of the corresponding weights are either up-weighted or down-weighted depending on the magnitude and sign of $\lambda$, splitting the representation into two parts. This division is reflected in the network's internal representations. With our solution, $\mathbf{Q}\mathbf{Q}^T(t)$, which captures the temporal dynamics of the similarity between hidden layer activations, we can analyze the network's internal representations in relation to the task. This allows us to determine whether the network adopts a *rich* or *lazy* representation, depending on the value of $\lambda$. Assuming convergence to the global minimum, which is guaranteed when the matrix $\mathbf{B}$ is non-singular, the internal representation satisfies $\mathbf{W}_1^T\mathbf{W}_1 = \tilde{\mathbf{V}}\tilde{\mathbf{S}}_1^2\tilde{\mathbf{V}}^T$ and $\mathbf{W}_2\mathbf{W}_2^T = \tilde{\mathbf{U}}\tilde{\mathbf{S}}_2^2\tilde{\mathbf{U}}^T$ with $\mathbf{W}_2\mathbf{W}_1 = \tilde{\mathbf{U}}\tilde{\mathbf{S}}\tilde{\mathbf{V}}^T$. Theorem C.3 in the Appendix provides a detailed proof of this limiting behavior. To illustrate this, we consider a hierarchical semantic learning task[1], introduced in Saxe et al. (2014); Braun et al. (2022), where living organisms are organized according to their features (Fig. 4A). The representational similarity of the task's inputs ($\tilde{\mathbf{V}}\tilde{\mathbf{S}}\tilde{\mathbf{V}}^T$) reflects this hierarchical structure (Fig.4A). Similarly, the representational similarity of the task's target values ($\tilde{\mathbf{U}}\tilde{\mathbf{S}}\tilde{\mathbf{U}}^T$) highlights the primary groupings of items. When training a two-layer network with *relative scale* $\lambda$ equal to zero and task-agnostic initialization (Mishkin & Matas, 2015), the input and output representational similarity matrices (Fig.4B) match the task's structure upon convergence. As derived in Theorem C.4 the network is guaranteed to find a *rich* solution regardless of the *absolute scale*, meaning $\mathbf{W}_1^T\mathbf{W}_1 = \tilde{\mathbf{V}}\tilde{\mathbf{S}}\tilde{\mathbf{V}}^T$ and $\mathbf{W}_2\mathbf{W}_2^T = \tilde{\mathbf{U}}\tilde{\mathbf{S}}\tilde{\mathbf{U}}^T$, as shown in Fig. 4C. Hence, the network learns task-specific representations. We also show that as $\lambda$ approaches either positive or negative infinity, the network symmetrically transitions into the *lazy* regime. As demonstrated in Theorem C.4 and illustrated in Fig. 4, the representations converge to an identity matrix for both large positive and large negative values of $\lambda$— emerging in the output representations for large positive $\lambda$ and input representations for large negative $\lambda$. This convergence indicates that the network adopts task-agnostic representations. Meanwhile, the other respective RSMs become negligible, with scales proportional to $1/\lambda$. Therefore, as shown in Theorem C.5, the NTK becomes static and equivalent to the identity matrix in the limit as $\lambda$ approaches infinity. However, the downscaled representations of the network remain structured and task-specific. Intuitively, in this

---

[1]In this setting, the network has equal input and output dimensions

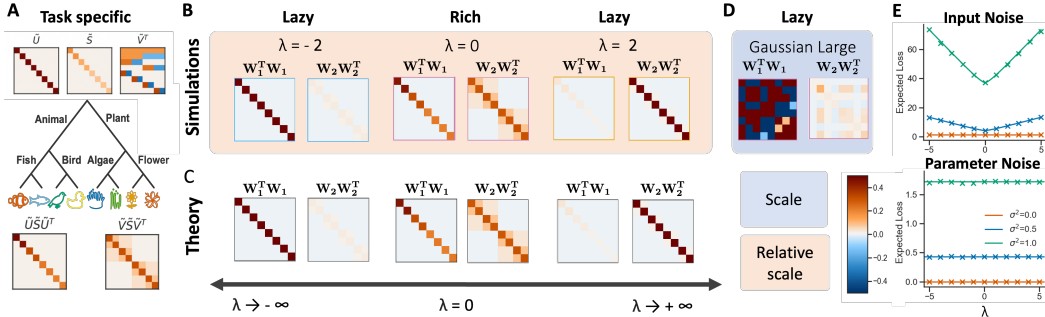

Figure 4: **A.** A semantic learning task with the SVD of the input-output correlation matrix of the task. (top) $U$ and $V$ represent the singular vectors, and $S$ contains the singular values. (bottom) The respective RSMs as $USU^\top$ for the input and $VSV^\top$ for the output task. **B.** Simulation results and **C.** Theoretical input and output representation matrices after training, showing convergence when initialized with values of $\lambda$ equal to $-2$, $0$, and $2$, according to the initialization scheme described in Appendix E.7. **D.** Final RSMs matrices after training converged when initialised from random large weights. **E.** After convergence, the network's sensitivity to input noise (top panel) is invariant to $\lambda$, but the sensitivity to parameter noise increases as $\lambda$ becomes smaller (or larger) than zero.

setup, the larger weights function as an identity-like projection, while the smaller weights adapt and align to the task. However, due to their relative scale compared to the larger weights, their contribution to the NTK remains negligible. This property could be beneficial if the weights are later rescaled, such as during fine-tuning, potentially enhancing generalization and transfer learning, as we will demonstrate in Section 6. We compare this to the scenario where both weights are initialized with large Gaussian values, leading to *lazy* learning that maintains a fixed NTK but lacks any structural representation, as illustrated in Fig. 4D. We further discuss the relationship between the scale and the relative scale in appendix A.4. Furthermore, in the infinite-width regime, where weights are initialized from a Gaussian distribution with large variance, averaging effects cause both input and output representations to approximate identity matrices. In this scenario, the network learns in the lazy regime with a fixed Neural Tangent Kernel (NTK). This behavior contrasts with the dynamics observed in the current setting since both input and output representations are task agnostic. Consequently, we propose a new *lazy* regime, which we refer to as the *semi-structured lazy* regime. We note that these existing regimes preserve only the input or output representation, resulting in a partial loss of structural information. In the nonlinear setting, this behavior is not expected to hold, as an additional factor comes into play in the computation of the NTK: the activation coefficients of the nonlinearity, as demonstrated in Kunin et al. (2024). In that case large relative weight (large positive $\lambda$) leads to a rapid rich regime. All together, we find that initialization will determine which layer in the network the task specification features resides in: layers initialized with large values will be task-agnostic, while those initialized with small values will be task-specific.

**Representation robustness and sensitivity to noise.** Here we examine the relationship between the learning regime and the robustness of the learned representations to added noise in the inputs and parameters. The expected post-convergence loss with added noise to the inputs is determined by the norm of the network function (Braun et al., 2025), which in our setting is independent of $\lambda$. Specifically, if we add zero-centered noise $\xi_{\mathbf{X}}$ with variance $\sigma_{\mathbf{X}}^2$ to the inputs, then the expected loss is $\langle \mathcal{L} \rangle_{\xi_{\mathbf{X}}} = \sigma_{\mathbf{X}}^2 \sum_{i=1}^{N_h} \tilde{\mathbf{S}}_i^2 + c$, where $c$ is a constant that depends solely on the statistics of the training data (Figure 4E, Appendix C.3). However, if instead noise is added to the parameters, the expected loss scales quadratically with the norm of the weight matrices (Braun et al., 2025), which in our setting depend on $\lambda$. In particular, zero-centered parameter noise $\xi_{\mathbf{W}_1}$ and $\xi_{\mathbf{W}_2}$ with variance $\sigma_{\mathbf{W}}^2$ results in an expected loss of $\langle \mathcal{L} \rangle_{\xi_{\mathbf{W}_1}, \xi_{\mathbf{W}_2}} = \frac{1}{2} N_i \sigma_{\mathbf{W}}^2 \|\mathbf{W}_2\|_F^2 + \frac{1}{2} N_o \sigma_{\mathbf{W}}^2 \|\mathbf{W}_1\|_F^2 + \frac{1}{2} N_i N_h N_o \sigma^4 + c$, with norms $\|\mathbf{W}_1\|_F^2 = \frac{1}{2} \sum_{i=1}^{N_h} \left( \sqrt{4\tilde{\mathbf{S}}_i^2 + \lambda^2} + \lambda \right)$ and $\|\mathbf{W}_2\|_F^2 = \frac{1}{2} \sum_{i=1}^{N_h} \left( \sqrt{4\tilde{\mathbf{S}}_i^2 + \lambda^2} - \lambda \right)$. This implies that, under the assumption of equal input-output dimensions, networks initialized with weights such that $\lambda = 0$, corresponding to the rich regime, converge to solutions that are most robust to parameter noise (Figure 4E, Appendix C.3). In practice, parameter noise could be interpreted as

Figure 5: **A.** Schematic representations of the network architectures considered, from left to right: funnel network, square network, and inverted-funnel network. **B.** The plot shows the NTK kernel distance from initialization, as defined in Fort et al. (2020) across the three architecture depicted schematically. **C.** The NTK kernel distance away from initialization over training time.

the noise occurring within the neurons of a biological network. Hence, a rich solution may enable a more robust representation in such systems.

**The impact of the architecture.** Thus far, we have found that the magnitude of the *relative scale* parameter $\lambda$ determines the extent of rich and lazy learning. Here, we explore how a network's learning regime is also shaped by the interaction of its architecture and the sign of the *relative scale*. We consider three types of network architectures, depicted in Fig. 5A: *funnel networks*, which narrow from input to output ($N_i > N_h = N_o$); *inverted-funnel networks*, which expand from input to output ($N_i = N_h < N_o$); and *square networks*, where input and output dimensions are equal ($N_i = N_h = N_o$). Our solution, $\mathbf{Q}\mathbf{Q}^T$, captures the dynamics of the NTK across these different network architectures. To examine the NTK's evolution under varying $\lambda$ initializations, we compute the kernel distance from initialization, as defined in Fort et al. (2020). As shown in Fig. 5B, we observe that funnel networks enter the *lazy* regime as $\lambda \to \infty$, while inverted-funnel networks do so as $\lambda \to -\infty$. The NTK remains static during the initial phase, rigorously confirming the rank argument first introduced by Kunin et al. (2024) for the multi-output setting. In the opposite limits of $\lambda$, these networks transition from a *lazy* regime to a *rich* regime. During this second alignment phase, the NTK matrix undergoes changes, indicating an initial *lazy* phase followed by a *delayed rich* phase. We further investigate and quantify this *delayed rich* regime, showing the NTK movement over training in Fig. 5C. This behavior is also quantified in Theorem C.6, which describes the rate of learning in this network. In Appendix A.4, we further explore the impact of the architecture as a function of the absolute and relative scale. Intuitively, the delayed onset of the rich regime occurs because no least-squares solution exists within the span of the network at initialization. In such cases, the network enters a delayed rich phase, where $\lambda$ tends toward infinity, with the magnitude of $\lambda$ determining the length of the delay. At first, the network exhibits *lazy* dynamics, striving to approximate the solution. However, as constraints necessitate adjustments in its directions, the network gradually transitions into the *rich* phase. For square networks this behavior is discussed in Section 5. Across all architectures, as $\lambda \to 0$, the networks consistently transition into the *rich* regime. Altogether, we further characterize the *delayed rich* regime in wide networks.

## 6 APPLICATIONS

In this section, we apply the exact solutions for the learning dynamics in deep linear networks described in Section 4 to illustrate several phenomena relevant to machine learning and neuroscience.

**Continual learning.** Continual learning, as thoroughly reviewed by Parisi et al. (2019), has long posed a significant challenge for neural network models in contrast to biological networks, particularly due to the issue of catastrophic forgetting (McCloskey & Cohen, 1989; Ratcliff, 1990; French, 1999). Similarly to the framework presented by Braun et al. (2022), our approach describes the exact solutions of the networks dynamics trained across a sequence of tasks describing the entire continual learning process. As detailed in Appendix D.1, we demonstrate that, regardless of the chosen value of $\lambda$, training on subsequent tasks can result in the overwriting of previously acquired knowledge, leading to catastrophic forgetting.

**Reversal learning.** During reversal learning, previously acquired knowledge must be relearned, necessitating the overcoming of an earlier established relationship between inputs and outputs. As demonstrated in Braun et al. (2022), reversal learning theoretically does not succeed in deep linear networks as the initalization aligns with the separatrix of a saddle point. While simulations show

that the learning dynamics can escape the saddle point due to numerical imprecision, the process is catastrophically slowed in its vicinity. However, when $\lambda$ is non-zero, reversal learning dynamics consistently succeed, as they avoid passing through the saddle point due to the initialization scheme. This is both theoretically proven and numerically illustrated in Appendix D.2. We also present a spectrum of reversal learning behaviors controlled by the *relative scale* $\lambda$, ranging from *rich* to *lazy* learning regimes. This spectrum has the potential to explain the diverse dynamics observed in animal behavior, offering insights into the learning regimes relevant toneuroscience experiments.

**Transfer learning.** We consider how different $\lambda$ initializations influence generalization to a new feature after being trained on an initial task. As detailed in Appendix D.3, we first train each network on the hierarchical semantic learning task described in Fig. 4. We then add a new feature to the dataset, and train the network specifically on the corresponding item while keeping the rest of the network parameters unchanged. Afterwards, we evaluate the generalization to the other items. We observe in Appendix Fig. D.3 that the hierarchical structure of the data is effectively transferred to the new feature when the representation is task-specific and $\lambda$ is zero. Conversely, when the input feature representation is *lazy* ($\lambda \leq 0$), meaning the hidden representation lacks adaptation, no hierarchical generalization is observed. Strikingly, when $\lambda$ is positive, the hierarchical structure in the input weights remains small but structured, while the output weights exhibit a *lazy* representation and the network generalizes hierarchically. Specifically Fig. D.3 shows that the generalization loss on untrained items with the new feature decreases as a function of increasing $\lambda$. Therefore, as $\lambda$ increases, networks more effectively transfer the hierarchical structure of the network to the new feature for untrained items, leading to an increase in generalization performance. This indicates that the *lazy* regime structure (large $\lambda$ values) can be beneficial for transfer learning.

**Fine-tuning** It is a common practice to pre- train neural networks on a large auxiliary task before fine-tuning them on a downstream task with limited samples. Despite the widespread use of this approach, the dynamics and outcomes of this method remain poorly understood. Our study establishes a theoretical basis for the success of fine-tuning, particularly how changes in $\lambda$-balancedness initialisation after pre-training affect performance on new datasets (see Appendix D.4). We consistently find that finetuning performance improves and converges more quickly as networks are re-balanced to larger values of $\lambda_{FT}$ and, conversely, decreases as $\lambda_{FT}$ approaches 0 as shown in Fig. D.4. Our work examines the fine-tuning dynamics of two-layer linear networks. While simple, these architectures are commonly utilized for fine-tuning large pre-trained language and vision models, notably in Low-Rank Adapters (LoRA) (Hu et al., 2022) as further discussed in Appendix D.4. While a detailed exploration of fine-tuning performance in practice as a function of initialization lies beyond the scope of this work, it remains an important direction for future research.

## 7 DISCUSSION

We derive exact solutions to the learning dynamics of deep linear networks. While our findings extend the range of analytically describable two-layer linear network problems, they are still limited by a set of assumptions. In particular, relaxing the assumptions that input covariance must be white and that initialization must be $\lambda$-*balanced* could bring the analysis closer to practical applications in machine learning and neuroscience. Moving towards the nonlinear setting would also make the findings more applicable in real-world scenarios. Despite these limitations, our solutions provide valuable insights into network behavior. We examine the transition between the *rich* and *lazy* regimes by analyzing the dynamics as a function of $\lambda$—the *relative scale*—across its full range from positive to negative infinity. Our analysis demonstrates that the *relative scale*, $\lambda$, is pivotal in managing this transition. Notably, we identify a structured *lazy* regime that promotes transfer learning. Building on previous work (Kunin et al., 2024) that shows these findings extend to basic nonlinear settings and practical scenarios, our theory suggests that further exploration of unbalanced initialization could optimize efficient feature learning. We leave for future work, the extension of this initialization to deep networks. Future work will focus on extending the application of the solution to the dynamics of fine-tuning and linear autoencoders.

ACKNOWLEDGMENTS

This research was funded in whole, or in part, by the Wellcome Trust [216386/Z/19/Z]. For the purpose of Open Access, the author has applied a CC BY public copyright licence to any Author Accepted Manuscript version arising from this submission. C.D. and A.S. were supported by the Gatsby Charitable Foundation (GAT3755). Further, A.S. was supported by the Sainsbury Wellcome Centre Core Grant (219627/Z/19/Z) and A.S. is a CIFAR Azrieli Global Scholar in the Learning in Machines & Brains program. A.M.P. was supported by the Imperial College London President's PhD Scholarship. L.B. was supported by the Woodward Scholarship awarded by Wadham College, Oxford. and the Medical Research Council [MR/N013468/1]. D.K. thanks the Open Philanthropy AI Fellowship for support. This research was funded in whole, or in part, by the Wellcome Trust [216386/Z/19/Z].

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

## A PRELIMINARIES

### A.1 APPENDIX: BALANCED CONDITION

**Definition A.1** (Definition of $\lambda$-*balanced* property (Saxe et al. (2013), Marcotte et al. (2023))). The weights $\boldsymbol{W_1}, \boldsymbol{W_2}$ are $\lambda$-*balanced* if and only if there exists a **Balanced Coefficient** $\lambda \in \mathbb{R}$ such that:

$$B(\boldsymbol{W_1}, \boldsymbol{W_2}) = \boldsymbol{W_2}^T \boldsymbol{W_2} - \boldsymbol{W_1}\boldsymbol{W_1}^T = \lambda \mathbf{I} \tag{19}$$

where $B$ is called the **Balanced Computation**.
For $\lambda = 0$ we have **Zero-Balanced** given as **A4** (). $\mathbf{W}_1(0)\mathbf{W}_1(0)^T = \mathbf{W}_2(0)^T\mathbf{W}_2(0)$.

**Theorem A.2.** *Balanced Condition Persists Through Training*
*Suppose at initialization*

$$\boldsymbol{W_2}(0)^T \boldsymbol{W_2}(0) - \boldsymbol{W_1}(0)\boldsymbol{W_1}(0)^T = \lambda \mathbf{I} \tag{20}$$

*Then for all $t \geq 0$*

$$\boldsymbol{W_2}(t)^T \boldsymbol{W_2}(t) - \boldsymbol{W_1}(t)\boldsymbol{W_1}(t)^T = \lambda \mathbf{I} \tag{21}$$

*Proof.* Consider:

$$
\begin{aligned}
\tau \frac{d}{dt}\left[\boldsymbol{W_2}(t)\boldsymbol{W_2}(t)^T - \boldsymbol{W_1}(t)\boldsymbol{W_1}(t)^T\right] &= \left(\tau\frac{d}{dt}\boldsymbol{W_2}(t)\right)^T \boldsymbol{W_2}(t) + \boldsymbol{W_2}(t)^T\left(\tau\frac{d}{dt}\boldsymbol{W_2}(t)\right) \\
&\quad - \left(\tau\frac{d}{dt}\boldsymbol{W_1}(t)\right)\boldsymbol{W_1}(t)^T - \boldsymbol{W_1}(t)\left(\tau\frac{d}{dt}\boldsymbol{W_1}(t)\right)^T \\
&= \boldsymbol{W_1}(t)\left(\tilde{\boldsymbol{\Sigma}}^{yx} - \boldsymbol{W_2}(t)\boldsymbol{W_1}(t)\tilde{\boldsymbol{\Sigma}}^{xx}\right)^T \boldsymbol{W_2}(t) \\
&\quad + \boldsymbol{W_2}(t)^T\left(\tilde{\boldsymbol{\Sigma}}^{yx} - \boldsymbol{W_2}(t)\boldsymbol{W_1}(t)\tilde{\boldsymbol{\Sigma}}^{xx}\right)\boldsymbol{W_1}(t) \\
&\quad - \boldsymbol{W_2}(t)^T\left(\tilde{\boldsymbol{\Sigma}}^{yx} - \boldsymbol{W_2}(t)\boldsymbol{W_1}(t)\tilde{\boldsymbol{\Sigma}}^{xx}\right)\boldsymbol{W_1}(t) \\
&\quad - \boldsymbol{W_1}(t)\left(\tilde{\boldsymbol{\Sigma}}^{yx} - \boldsymbol{W_2}(t)\boldsymbol{W_1}(t)\tilde{\boldsymbol{\Sigma}}^{xx}\right)\boldsymbol{W_2}(t) \\
&= \mathbf{0}
\end{aligned}
$$

Note that $\boldsymbol{W_2}(t)^T\boldsymbol{W_2}(t) - \boldsymbol{W_1}(t)\boldsymbol{W_1}(t)^T$ is conserved for any initial value $\lambda$. $\quad\square$

### A.2 DISCUSSION ASSUMPTIONS

**Whittened Inputs.** Although the whitened input assumption is quite strong, it is commonly used in analytical work to obtain exact solutions, and much of the existing literature relies on these solutions Fukumizu (1998); Braun et al. (2022); Kunin et al. (2024). While relaxing this assumption prevents the exact description of network dynamics, Kunin et al. (2024) examine the implicit bias of the training trajectory without relying on whitened inputs. If the interpolating manifold is one-dimensional, the solution can be solved exactly in terms of $\lambda$. Their findings demonstrate a similar quantitative dependence on $\lambda$, governing the implicit bias transition between rich and lazy regimes. Furthermore, recent advancements, such as the "decorrelated backpropagation" technique introduced by Dalm et al. (2024) which whitens inputs during training, showing that optimizing for whitened inputs can actually be done in practice and improve efficiency in real-world applications. Importantly, This study highlights that in certain real-world scenarios, whitening can provide a more optimal learning condition. This approaches emphasize the potential advantages of input whitening for downstream tasks, reinforcing the validity of our assumption.

**Dimension.** Previous works imposed specific dimensionality constraints. For example: Fukumizu (1998) assumed equal input and output dimensions ($N_i = N_o$) while allowing a bottleneck in the hidden dimension ($N_h \leq N_i = N_o$). Braun et al. (2022) extended these solutions to cases with unequal input and output dimensions ($N_i \neq N_o$) but restricted bottleneck networks ($N_h = min(N_i, N_o)$) and introduced an additional invertibility condition on the **B**. In our work we allow for unequal input and output $N_i \neq N_o$ and do not introduce an additional invertibility assumption. This flexibility expands the applicability of our framework to a wider range of architectures.

**Full rank**  Previous work by Braun et al. (2022), imposed a full-rank initialization condition, defined as $\mathrm{rank}(\mathbf{W}_2(0)\mathbf{W}_1(0)) = N_i = N_o$. However, this assumption is not necessary in our framework.

**Balancedness Assumption**  A significant departure from prior works is the relaxation of the balancedness assumption: Earlier studies, such as Fukumizu (1998) and Braun et al. (2022) assumed strict zero-balancedness ($\mathbf{W}_1(0)\mathbf{W}_1(0)^T = \mathbf{W}_2(0)^T\mathbf{W}_2(0)$), which constrained the networks to the *rich* regime. Our approach generalizes this to $\lambda$-balancedness, enabling exploration of the continuum between the *rich* and *lazy* regimes. While some efforts, such as Tarmoun et al. (2021), have explored removing the zero-balanced constraint, their solutions were limited to unstable or mixed forms. In contrast, our methodology systematically studies different learning regimes by varying initialization properties, particularly through the relative scale parameter. This allows controlled transitions between regimes, advancing understanding of neural network behavior across the spectrum. Other studies, such as Kunin et al. (2024) and Xu & Zheng (2024) have also relaxed the balancedness assumption, though their analysis was restricted to single-output neuron settings. We emphasize the importance of this balanced quantity by rigorously proving that, in the averaging limit, standard network initializations (e.g., LeCun initialization LeCun et al. (1998), He initialization) lead to $\lambda$-balanced behavior in the infinite-width limit. Specifically, the term $\mathbf{W}_1(0)\mathbf{W}_1(0)^T = \mathbf{W}_2(0)^T\mathbf{W}_2(0)$ converges to a scaled identity matrix. Furthermore, previous studies have demonstrated that the relative scaling of $\lambda$ significantly impacts the learning regime in practical scenarios, highlighting the crucial role of dynamical studies of networks as a function of this parameter.

## A.3 Random weight initialisations and $\lambda$-Balanced Property

Throughout this work, we assume that initial weights are $\lambda$-Balanced. However, in practice, weights are not initialized with that goal in mind. Usually, a weight matrix $\mathbf{W}$ is initialized with some random distribution centered around 0, with variance inversely proportional to the number of layers on which $\mathbf{W}$ has a direct effect (Glorot & Bengio (2010), LeCun et al. (1998), He et al. (2015)). In this section, we show that many common initialization techniques lead to $\lambda$-Balanced weights in expectation. Furthermore, as the size of a network tends to infinity, these random weights are $\lambda$-Balanced in probability.

We do this by first finding the expectation and variance of the balance computation for two adjacent weight matrices, $\mathbf{W}_{i+1}$ and $\mathbf{W}_i$, initialized under a normal distribution with zero mean. Subsequently, we describe how network structure and size can impact the expectation and variance of the balance computation.

**Theorem A.3.** *[Random Weight Initialization Leads to Balanced Condition] Consider a fully connected neural network with $L$ layers. Each layer has $N_i$ neurons, and the weights of each layer $\boldsymbol{W_i}$ is a matrix of dimension $(N_i, N_{i+1})$. The matrix $\boldsymbol{W_i} = (w^i_{N,m})$ where $w^i_{N,m} \sim \mathcal{N}(0, \sigma_i^2)$, where $\sigma_i^2$ is determined based on the initialization technique. Then the following hold for all $i \in [1, L-1]$:*

1. $\mathbb{E}\left[\boldsymbol{W_{i+1}^T W_{i+1}} - \boldsymbol{W_i W_i^T}\right] = (N_{i+2}\sigma_{i+1}^2 - N_i\sigma_i^2)\mathbf{I}$
2. $Var\left[\boldsymbol{W_{i+1}^T W_{i+1}} - \boldsymbol{W_i W_i^T}\right] = (N_{i+2}\sigma_{i+1}^4 + N_i\sigma_i^4)\mathbb{B}$, *where $\mathbb{B}$ is a square matrix with fours across the diagonal and ones everywhere else.*

Note that in the case $L = 3$, $N_0 = i$, $N_1 = h$, $N_2 = o$ with $i, h, o$ being the input, hidden and output dimensions respectively as defined in the main text.

*Proof of Theorem A.3.*

$$\text{Let } \boldsymbol{W_{i+1}} = \begin{pmatrix} w_{1,1} & w_{1,2} & \cdots & w_{1,N_{i+2}} \\ w_{2,1} & w_{2,2} & \cdots & w_{2,N_{i+2}} \\ \vdots & \vdots & \ddots & \vdots \\ w_{N_{i+1},1} & w_{N_{i+1},2} & \cdots & w_{N_{i+1},N_{i+2}} \end{pmatrix} \tag{22}$$
$$= (\overline{w}_1 \quad \overline{w}_2 \quad \ldots \quad \overline{w}_{N_{i+2}})$$

with $\overline{w}_j = (w_{1,j}, w_{2,j}, \ldots, w_{N_{i+1},j})^T$.

Then,

$$
\boldsymbol{W_{i+1}^T W_{i+1}} = \begin{pmatrix} \overline{w}_1^T \\ \overline{w}_2^T \\ \vdots \\ \overline{w}_{N_{i+2}}^T \end{pmatrix} \begin{pmatrix} \overline{w}_1 & \overline{w}_2 & \cdots & \overline{w}_{N_{i+2}} \end{pmatrix}
$$

$$
= \begin{pmatrix} \langle \overline{w}_1, \overline{w}_1 \rangle & \langle \overline{w}_1, \overline{w}_2 \rangle & \cdots & \langle \overline{w}_1, \overline{w}_{N_{i+2}} \rangle \\ \langle \overline{w}_2, \overline{w}_1 \rangle & \langle \overline{w}_2, \overline{w}_2 \rangle & \cdots & \langle \overline{w}_2, \overline{w}_{N_{i+2}} \rangle \\ \vdots & \vdots & \ddots & \vdots \\ \langle \overline{w}_{N_{i+2}}, \overline{w}_1 \rangle & \langle \overline{w}_{N_{i+2}}, \overline{w}_2 \rangle & \cdots & \langle \overline{w}_{N_{i+2}}, \overline{w}_{N_{i+2}} \rangle \end{pmatrix}
$$

Now, consider $\langle \overline{w}_i, \overline{w}_j \rangle$ with $i \neq j$,

$$
\langle \overline{w}_i, \overline{w}_j \rangle = \sum_{k=1}^{N_{i+2}} w_{k,i} w_{k,j}
$$

$$
\mathbb{E}\left[\langle \overline{w}_i, \overline{w}_j \rangle\right] = \mathbb{E}\left[\sum_{k=1}^{N_{i+2}} w_{k,i} w_{k,j}\right]
$$

$$
= \sum_{k=1}^{N_{i+2}} \mathbb{E}[w_{k,i} w_{k,j}]
$$

$$
= \sum_{k=1}^{N_{i+2}} \mathbb{E}[w_{k,i}]\mathbb{E}[w_{k,j}] = 0 \quad \text{(by independence)}
$$

$$
\mathrm{Var}\left[\langle \overline{w}_i, \overline{w}_j \rangle\right] = \mathbb{E}\left[\langle \overline{w}_i, \overline{w}_j \rangle^2\right] - \left[\mathbb{E}\left[\langle \overline{w}_i, \overline{w}_j \rangle\right]\right]^2
$$

$$
= \mathbb{E}\left[\left(\sum_{k=1}^{N_{i+2}} w_{k,i} w_{k,j}\right)^2\right]
$$

$$
= \mathbb{E}\left[\sum_{k=1}^{N_{i+2}} w_{k,i}^2 w_{k,j}^2 + 2\sum_{k=1}^{N_{i+2}}\sum_{l>k} w_{k,i} w_{k,j} w_{l,i} w_{l,j}\right]
$$

$$
= \sum_{k=1}^{N_{i+2}} \mathbb{E}[w_{k,i}^2 w_{k,j}^2] + 2\sum_{k=1}^{N_{i+2}}\sum_{l>k} \mathbb{E}[w_{k,i}]\mathbb{E}[w_{k,j}]\mathbb{E}[w_{l,i}]\mathbb{E}[w_{l,j}]
$$

$$
= \sum_{k=1}^{N_{i+2}} \mathbb{E}[w_{k,i}^2]\mathbb{E}[w_{k,j}^2]
$$

$$
= (N_{i+2})\sigma_{i+1}^4
$$

Similarly, consider $\langle \overline{w}_i, \overline{w}_i \rangle$:

$$\langle \overline{w}_i, \overline{w}_i \rangle = \sum_{k=1}^{N_{i+2}} w_{k,i}^2$$

$$\mathbb{E}\left[\langle \overline{w}_i, \overline{w}_i \rangle\right] = \mathbb{E}\left[\sum_{k=1}^{N_{i+2}} w_{k,i}^2\right] = N_{i+2}\mathbb{E}\left[w_{k,i}^2\right] = N_{i+2}\sigma_{N_{i+1}}^2$$

$$\text{Var}\left[\langle \overline{w}_i, \overline{w}_i \rangle\right] = \mathbb{E}\left[\left(\langle \overline{w}_i, \overline{w}_i \rangle\right)^2\right] - \mathbb{E}\left[\langle \overline{w}_i, \overline{w}_i \rangle\right]^2$$

$$= \mathbb{E}\left[\left(\sum_{k=1}^{N_{i+2}} w_{k,i}^2\right)^2\right] - N_{i+2}^2\sigma_{N_{i+1}}^4$$

$$= \mathbb{E}\left[\left(\sum_{k=1}^{N_{i+2}} w_{k,i}^2\right)^2\right] - N_{i+2}^2\sigma_{N_{i+1}}^4$$

$$= \mathbb{E}\left[\sum_{k=1}^{N_{i+2}} w_{k,i}^4 + 2\sum_{k=1}^{N_{i+2}}\sum_{l=k+1}^{N_{i+2}} w_{k,i}^2 w_{l,i}^2\right] - N_{i+2}^2\sigma_{N_{i+1}}^4$$

$$= \sum_{k=1}^{N_{i+2}} \mathbb{E}\left[w_{k,i}^4\right] + 2\sum_{k=1}^{N_{i+2}}\sum_{l=k+1}^{N_{i+2}} \mathbb{E}\left[w_{k,i}^2\right]\mathbb{E}\left[w_{l,i}^2\right] - N_{i+2}^2\sigma_{N_{i+1}}^4$$

$$= N_{i+2}(3\sigma_{N_{i+1}}^4) + (N_{i+2}^2 - N_{i+2})\sigma_{N_{i+1}}^4 - N_{i+2}^2\sigma_{N_{i+1}}^4$$

$$= 4N_{i+2}\sigma_{N_{i+1}}^4$$

Hence

$$\mathbb{E}\left[\boldsymbol{W_{i+1}^T W_{i+1}}\right] = \left(N_{i+2}\sigma_{i+1}^2\right)\mathbf{I}$$

$$\text{Var}\left[\boldsymbol{W_{i+1}^T W_{i+1}}\right] = 4\left(N_{i+2}\right)\sigma_{i+1}^4\mathbb{B}$$

For the case for $\boldsymbol{W_i}$, notice we can express $\boldsymbol{W_i W_i^T}$ as $(\boldsymbol{W_i^T})^T(\boldsymbol{W_i^T})$. Hence we can use the proof above, with $\boldsymbol{W_{i+1}'} = \boldsymbol{W_i^T}$. In this case the matrix $\boldsymbol{W_{i+1}'}$ has shape $(N_i, N_{i+1})$, and each element of the matrix has variance $\sigma_i^2$. We have:

$$\mathbb{E}\left[\boldsymbol{W_i W_i^T}\right] = N_i\sigma_i^2\mathbf{I}$$

$$\text{Var}\left[\boldsymbol{W_i W_i^T}\right] = N_i\sigma_i^4\mathbb{B}$$

By assumption, $\boldsymbol{W_i}, \boldsymbol{W_{i+1}}$ are independent. Hence $Cov(\boldsymbol{W_i}, \boldsymbol{W_{i+1}}) = 0$. We can use this property together with linearity of expectation:

$$\mathbb{E}\left[\boldsymbol{W_{i+1}^T W_{i+1}} - \boldsymbol{W_i^T W_i}\right] = \left(N_{i+2}\sigma_{i+1}^2 - N_i\sigma_i^2\right)\mathbf{I}$$

$$\text{Var}\left[\boldsymbol{W_{i+1}^T W_{i+1}} - \boldsymbol{W_i^T W_i}\right] = \left(N_{i+2}\sigma_{i+1}^4 + N_i\sigma_i^4\right)\mathbb{B}$$

This completes the proof. $\square$

In neural network training, proper weight initialization is crucial for ensuring stable gradients during backpropagation, which helps to avoid issues such as vanishing and exploding gradients. The goal of weight scaling is to maintain appropriate variance across layers, enabling efficient and effective learning (Glorot & Bengio (2010)). The weights are typically initialized to be random and centered around 0 to break symmetry and ensure that different neurons learn different features.

Some of the most commonly used initialization methods are detailed below:

- **LeCun Initialization (LeCun et al. (1998))**: Weights are initialized using a normal distribution with a mean of 0 and a variance of $\frac{1}{N_i}$, where $N_i$ is the number of input units in the layer. Mathematically, the weights $w$ are drawn from $\mathcal{N}(0, \frac{1}{N_i})$.

- **Glorot Initialization (Glorot & Bengio (2010))**: Weights are initialized using a normal distribution with a mean of 0 and a variance of $\frac{2}{N_i+N_{i+1}}$, where $N_i$ is the number of input units and $N_{i+1}$ is the number of output units. This method balances the variance between layers with different widths. Mathematically, the weights $w$ are drawn from $\mathcal{N}(0, \frac{2}{N_i+N_{i+1}})$.

- **He Initialization (He et al. (2015))**: Weights are initialized using a normal distribution with a mean of 0 and a variance of $\frac{2}{N_i}$, where $N_i$ is the number of input units in the layer. This method is particularly suited for layers with ReLU activation functions. Mathematically, the weights $w$ are drawn from $\mathcal{N}(0, \frac{2}{N_i})$.

- **Scaled Initialization (Rahnamayan & Wang (2009))**: Weights are initialized using a normal distribution with a mean of 0 and a variance of $\frac{\alpha_i}{N_i}$, where $N_i$ is the number of input units in the layer and $\alpha_i$ is a parameter specific to each layer. Mathematically, the weights $w$ are drawn from $\mathcal{N}(0, \frac{\alpha_i}{N_i})$.

These initialization methods help ensure that the network starts with weights that facilitate stable and efficient learning, avoiding the common pitfalls of poorly initialized neural networks. Using (A.3), we can calculate the respective Expectations and Variances of the Balanced Computation under the different initialisations:

| Initialization | $\text{Var}(w_{N,m}^{i+1})$ | $\text{Var}(w_{N,m}^{i})$ | $\mathbb{E}[\mathbf{W}_{i+1}^T\mathbf{W}_{i+1} - \mathbf{W}_i\mathbf{W}_i^T]$ | $\text{Var}[\mathbf{W}_{i+1}^T\mathbf{W}_{i+1} - \mathbf{W}_i\mathbf{W}_i^T]$ |
|---|---|---|---|---|
| LeCun | $\frac{1}{N_{i+1}}$ | $\frac{1}{N_i}$ | $\left(\frac{N_{i+2}}{N_{i+1}} - 1\right)\mathbf{I}$ | $\left(\frac{N_{i+2}}{N_{i+1}^2} + \frac{1}{N_i}\right)\mathbb{B}$ |
| Glorot | $\frac{2}{N_{i+1}+N_{i+2}}$ | $\frac{2}{N_{i+1}+N_i}$ | $2\left(\frac{N_{i+2}}{N_{i+1}+N_{i+2}} - \frac{N_i}{N_{i+1}+N_i}\right)\mathbf{I}$ | $(N_{i+2}\left(\frac{2}{N_{i+1}+N_{i+2}}\right)^2 + N_i\left(\frac{2}{N_i+N_{i+1}}\right)^2)\mathbb{B}$ |
| He | $\frac{2}{N_{i+1}}$ | $\frac{2}{N_i}$ | $2\left(\frac{N_{i+2}}{N_{i+1}} - 1\right)\mathbf{I}$ | $4\left(\frac{N_{i+2}}{N_{i+1}^2} + \frac{1}{N_i}\right)\mathbb{B}$ |
| Scaled | $\frac{\alpha_{i+1}^2}{N_{i+1}}$ | $\frac{\alpha_i^2}{N_i}$ | $\left(\frac{N_{i+2}}{N_{i+1}}\alpha_{i+1}^2 - \alpha_i^2\right)\mathbf{I}$ | $(N_{i+2}\left(\frac{\alpha_{i+1}^2}{N_{i+1}}\right)^2 + N_i\left(\frac{\alpha_i^2}{N_i}\right)^2)\mathbb{B}$ |

Table 1: Comparison of Variance and Expectation of Balanced Computation for Different Weight Initializations

Fig. 6 shows a numerical example of how the Balanced computation would look like for initialising weights with LeCun, He, Scaled and Glorot initialisations using $N_i = 160, N_{i+1} = 80, N_{i+2} = 120$. With these numbers of nodes in each layer one can appreciate how the Balanced Computation on the weights is visually similar to a scaled identity matrix. In Fig. 7 We show how the sign of the scaled identity changes with the dimentions $N_i, N_{i+1} N_{i+2}$.

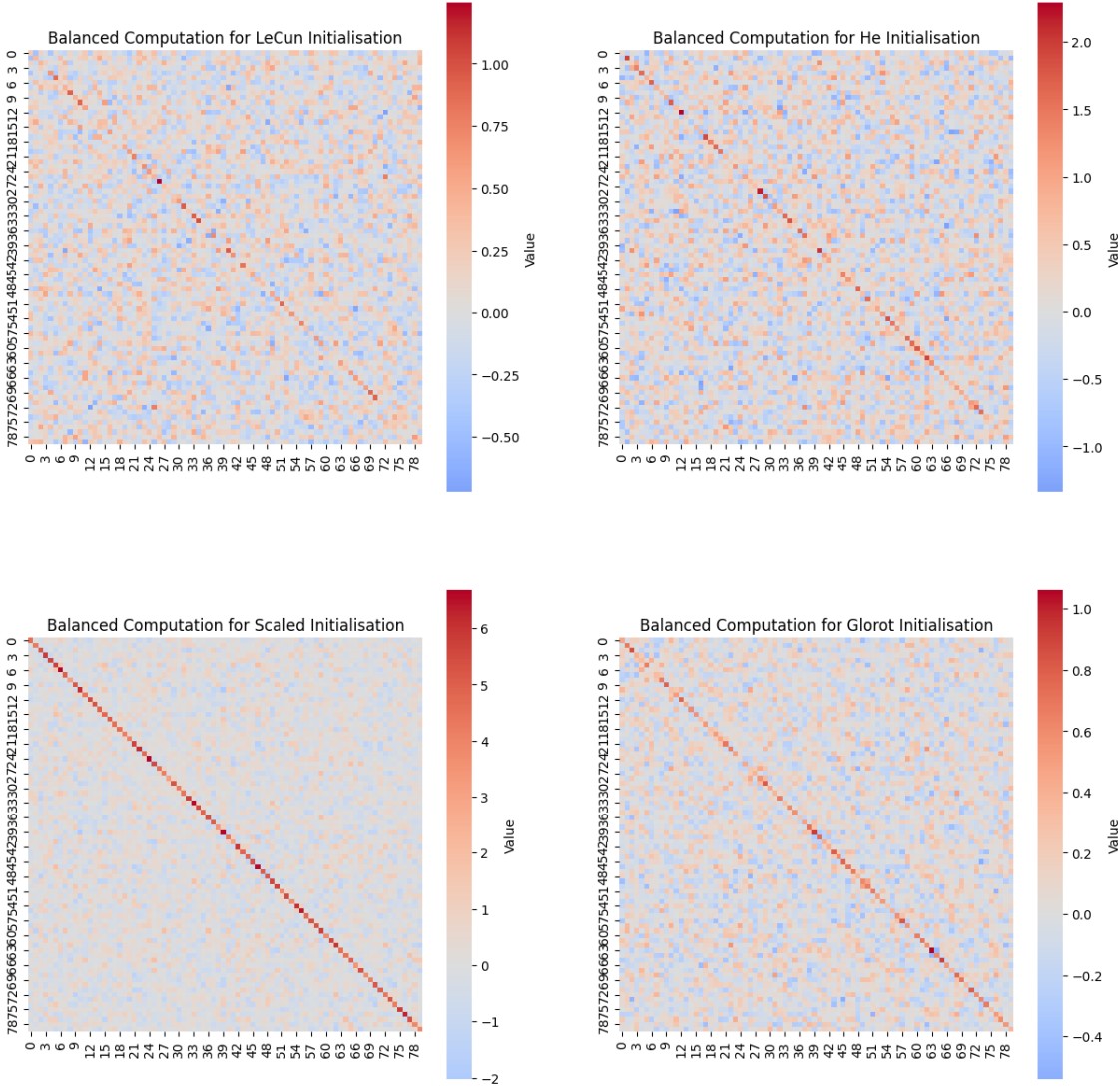

Figure 6: Balanced computation for different weight initializationsusing $N_i = 160, N_{i+1} = 80, N_{i+2} = 120, \alpha_1 = 1, \alpha_2 = 2$. The figure compares LeCun, He, Scaled, and Glorot initializations, showing how the balanced computation on the weights visually resembles a scaled identity matrix.

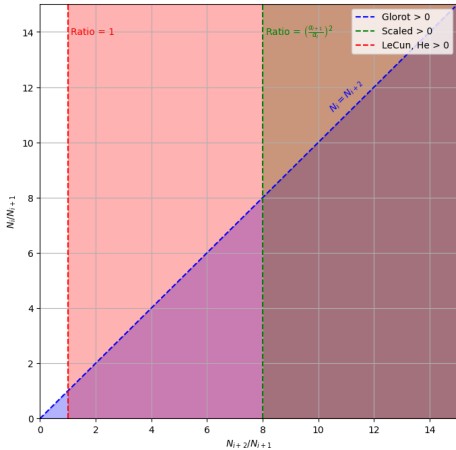

Figure 7: Impact of Network Structure on sign of balanced coefficient for different initialisations. Lecun, He and Scaled initialisations depend solely on the ratio of the sizes of the output and hidden layers. Scaled initialisation's dependence is affected by the parameters $\alpha_{i+1}, \alpha_i$. Glorot initialisation's sign depends on both ratios.

State-of-the-art models such as (Krizhevsky et al. (2012), Liu et al. (2023a)) use more than 10,000 nodes in each hidden layer. This implies that if we wished to perform some mathematical analysis on these models, assuming the initial random weights are $\lambda$-Balanced would be a very close depiction to reality. In addition, these models often have a different number of nodes per layer, so understanding the effect of the relationship between $N_i, N_{i+1}$, and $N_{i+2}$ is crucial.

Specifically, we aim to understand how changes in the relative width of layers $i, i+1, i+2$ affect the Balanced Computation: suppose $N_{i+1} = kN_i$ for some $k \in \mathbb{R}^+$ and $N_{i+2} = rN_i$ for some $r \in \mathbb{R}^+$. Then we can express the table above in terms of $k, r$, and $N_i$ only:

| Initialization | $\text{Var}(w_{N,m}^{i+1})$ | $\text{Var}(w_{N,m}^i)$ | $\mathbb{E}[\boldsymbol{W}_{i+1}^T\boldsymbol{W}_{i+1} - \boldsymbol{W}_i\boldsymbol{W}_i^T]$ | $\text{Var}[\boldsymbol{W}_{i+1}^T\boldsymbol{W}_{i+1} - \boldsymbol{W}_i\boldsymbol{W}_i^T]$ |
|---|---|---|---|---|
| LeCun | $\frac{1}{kN_i}$ | $\frac{1}{N_i}$ | $\left(\frac{r}{k} - 1\right)\mathbf{I}$ | $\frac{1}{N_i}\left(\frac{r}{k^2} + 1\right)\mathbb{B}$ |
| Glorot | $\frac{2}{N_i(k+r)}$ | $\frac{2}{N_i(k+1)}$ | $2\left(\frac{r}{k+r} - \frac{1}{k+1}\right)\mathbf{I}$ | $\frac{4}{N_i}\left(\frac{r}{(r+k)^2} + \frac{1}{(k+1)^2}\right)\mathbb{B}$ |
| He | $\frac{2}{kN_i}$ | $\frac{2}{N_i}$ | $2\left(\frac{r}{k} - 1\right)\mathbf{I}$ | $\frac{4}{N_i}\left(\frac{r}{k^2} + 1\right)\mathbb{B}$ |
| Scaled | $\frac{\alpha_{i+1}^2}{kN_i}$ | $\frac{\alpha_i^2}{N_i}$ | $\left(\frac{r}{k}\alpha_{i+1}^2 - \alpha_i^2\right)\mathbf{I}$ | $\frac{1}{N_i}\left(\frac{r\alpha_{i+1}^2}{k^2} + \alpha_i^2\right)\mathbb{B}$ |

Table 2: Comparison of Variance and Expectation for Different Initializations

From Table 2, we can observe that as the number of nodes in each layer tends to infinity, while the ratios between the number of nodes in each layer $(r, k)$ are maintained, the variance of the Balanced Computation tends to zero. Hence the Balanced Computation converges in probability to $\lambda$-Balanced weights.

Further, Table 2 shows that a larger value of $r$ will lead to a higher value of $\lambda$ in every one of the initializations displayed. Moreover, a larger $k$ has the opposite effect. In addition, we can observe that in some initializations there are limits as to what value $\lambda$ can take (such as in LeCun $\lambda \geq 0$).

Some special cases of $r$ and $k$ are interesting to consider to gain an intuition of how changing these values influences the Balancedness of the Weights. In the table below, we consider the cases:

1. $r = k$: the three layers of the network have the same number of nodes.

2. $r \to 0$, $k$ fixed: $N_i >> N_{i+2}, N_{i+1}$, the first of the three layers is much larger than the other two layers.

3. $r \to \infty$, $k$ fixed: $N_{i+2} >> N_i, N_{i+1}$, the last of the three layers is much larger than the other two layers ($k$ is fixed).

4. $k \to 0$, $r$ fixed: the middle layer is exceedingly small, $N_{i+1} << N_i, N_{i+2}$

5. $k \to \infty$, $r$ fixed: the middle layer is much bigger than the other two layers, $N_{i+1} >> N_i, N_{i+2}$

6. $r = \frac{\alpha_i^2}{\alpha_{i+1}^2}$: the inner and outer layers have a ratio proportional to the scale factors of each weight layer. This case is important for Scaled initialization.

| Initialization | $r = k$ | $r \to 0$ | $r \to \infty$ | $k \to \infty$ | $k \to 0$ | $r = \frac{\alpha_i^2}{\alpha_{i+1}^2}k$ |
|---|---|---|---|---|---|---|
| LeCun | $0$ | $-\mathbf{I}$ | $\frac{r}{k}\mathbf{I}$ | $-\mathbf{I}$ | $\frac{r}{k}\mathbf{I}$ | - |
| Glorot | $0$ | $2\mathbf{I}$ | $2\mathbf{I}$ | $0$ | $-\frac{2}{k+1}\mathbf{I}$ | - |
| He | $0$ | $-2\mathbf{I}$ | $2\left(\frac{r}{k}\right)\mathbf{I}$ | $-2\mathbf{I}$ | $2\left(\frac{r}{k}\right)\mathbf{I}$ | - |
| Scaled | $(\alpha_{i+1}^2 - \alpha_i^2)\mathbf{I}$ | $-\alpha_i^2\mathbf{I}$ | $\frac{r}{k}\alpha_{i+1}^2\mathbf{I}$ | $-\alpha_i^2\mathbf{I}$ | $\frac{r}{k}\alpha_{i+1}^2\mathbf{I}$ | $0$ |

Table 3: Comparison of Variance and Expectation under Different Conditions

From Table 3 above one can appreciate the impact network structure can have on the Balanced Computation of the weights of each layer. One can also see that there are many cases when the Balanced computation does not equal 0, both in the limit of $r, k$ and not in the limit.

We have showed that although the Balanced property is only preserved in Linear Networks, it occurs at initialisation for large networks which utilise some of the most common weight initialisation techniques.

These findings provide motivation to better understand the relation between the Balanced Computation of a Network, its structure and the regime it will learn in. If we are able to understand the relation between $\lambda$-Balanced weights and Rich and Lazy Learning in Linear Networks, one might be able to approximate these results to the nonlinear case.

A possible future application might be the ability to heavily influence a network's learning regime by altering the relative width of its layers, its activation functions or weight initialisation techniques used for each layer.

In order to better understand the effects of $\lambda$ on the learning dynamics and learning regime of the network, we first study aligned $\lambda$-Balanced networks.

### A.4 SCALE VS RELATIVE SCALE

A straightforward intuition for the scale and the relative scale can be gained by considering the scalar case where $N_i = N_h = N_o = 1$. In this scenario, it is easy to ensure that $w_1^2 = w_2^2$ satisfies $\lambda = 0$ while allowing for different absolute scales. For instance, $w_1 = w_2 = 0.001$ or $w_1 = w_2 = 5$. In such cases, the absolute scale is clearly decoupled from the relative scale. However, in more complex settings, the relative scale and absolute scale interact in non-trivial ways. While our study primarily focuses on the effects of relative scale, the influence of absolute scale is inherently embedded within our framework through the definitions of $\mathbf{B}$, $\mathbf{C}$, and $\mathbf{D}$ (see Equations 10). However, this influence is not immediately apparent from the main theorem. A clearer distinction between the roles of relative and absolute scale can be observed in Theorem 5.1. We investigate first how $\lambda$, the relative scale parameter, governs the transition between sigmoidal and exponential dynamical regimes. A similar argument applies to absolute scale, which appears explicitly as $s_\alpha(0)$ in these equations. Consider the case when $\lambda = 0$, the dynamics of $s_\alpha$ reduce to the classical solution of the Bernoulli differential

equation. In the limiting case where $s_\alpha(0) \to 0$, the system exhibits classic sigmoidal dynamics (characteristic of the rich regime), whereas the limit $s_\alpha(0) \to \infty$ yields exponential dynamics (characteristic of the lazy regime).

We performed an experiment to further examine the interplay between relative weight scale, absolute weight scale, and the network's learning regime in the general setting. We define the absolute scale of the weights as the norm of $\mathbf{W_2 W_1}$. We generate random initial weights of a given relative and absolute scale and train the network on a random input-output task. We compute the logarithmic kernel distance of the NTK from initialization and the logarithmic loss throughout training. We plot these values in a heat map for $\lambda$ in $[-9, 9]$ and relative scale in $(0, 20]$. We repeat this procedure for three architectures: a square network $(N_i = 2, N_h = 2, N_o = 2)$, a funnel network $(N_i = 4, N_h = 2, N_o = 2)$, and an anti-funnel network $(N_i = 2, N_h = 2, N_o = 4)$. These are the same architectures as in Figure 5 in the main text.

In all three different types of networks, at the start of training, the model loss depends entirely on an absolute scale, not the relative scale. **Throughout training** across networks, the learning dynamics are intricately influenced by both the absolute and relative scales and fully captured by our theoretical solution. In the square network, the loss increases with absolute scale but decreases with relative scale, as shown in Fig.8. Similarly, the kernel distance phase plot exhibits an intricate relationship with the relative and absolute scales, as illustrated in Fig.5. Strikingly, for large imbalanced $\lambda$, even at small scales, the network transitions into a lazy regime. The funnel and anti-funnel network architectures demonstrate antisymmetric behavior as shown in Fig. 9 and Fig. 10. Here, we focus on the anti-funnel network for brevity. The evolution of the loss reveals that negatively $\lambda$ initializations first converge, whereas positively $\lambda$ initializations retain larger loss values. Additionally, the kernel distance attains its maximum for positive $\lambda$, aligning with the results outlined in section 5. **At convergence**, the loss across all networks stabilizes uniformly, irrespective of initial conditions, confirming consistent convergence. This outcome aligns with the theoretical expectation for linear networks under gradient flow, which predictably converge to the same solution. Furthermore, in square networks, the kernel distance depends exclusively on the relative scale $\lambda$ and peaks at $\lambda = 0$ (results corresponding to the green curve in Fig. 5B.) This observation illustrates that the regime at $\lambda = 0$ is consistently rich, independent of the absolute scale as predicted by our theoretical results in C.3. For funnel and anti-funnel networks, the kernel distance exhibits an antisymmetric pattern. In the anti-funnel network, the kernel distance depends mostly on $\lambda$, achieving high values for positive $\lambda$ and approaching zero for negative $\lambda$ (matching the results in Fig. 5B ( pink line)). Conversely, in the funnel network, the kernel distance is high for negative $\lambda$ and approaches zero for positive $\lambda$, corroborating the results in Fig. 5B. (blue line). These results emphasize the interplay between relative and absolute scales, highlighting their critical roles in determining the system's behavior. Altogether, the absolute scale and relative scale of the weights play a critical role in describing the phase portrait of the learning regime, as first demonstrated in the Kunin et al. (2024) paper on ReLU networks.

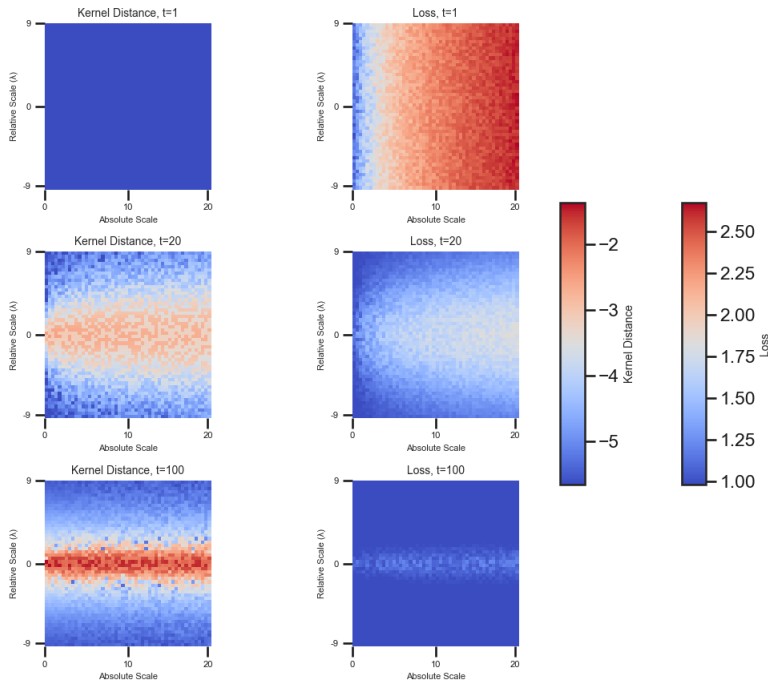

Figure 8: Square network: Phase plots showing (left) the logarithmic kernel distance of the NTK from initialization and (right) the corresponding logarithmic loss as functions of the relative scale $\lambda$ and the absolute scale. (Top to bottom) Different time steps during training $t = 1, t = 20, t = 100$.

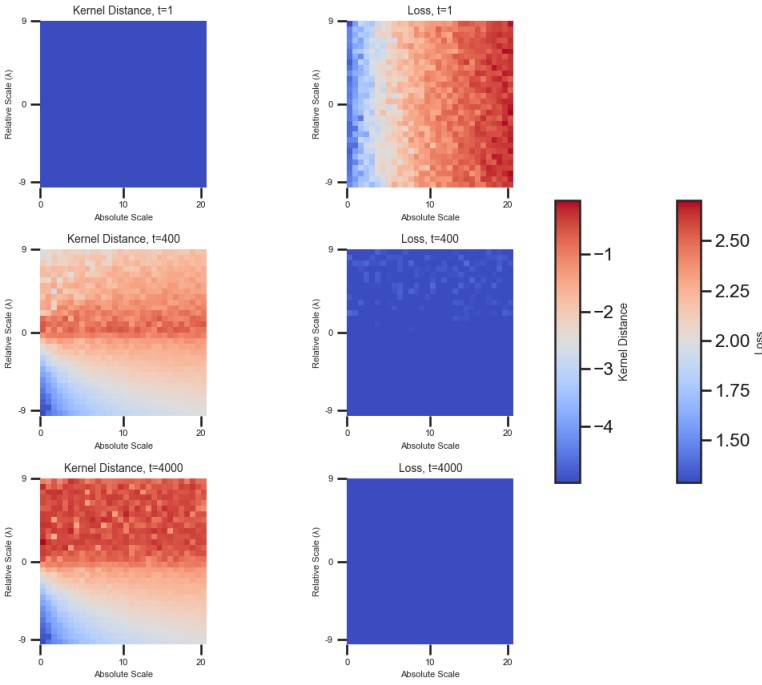

Figure 9: Anti-funnel network: Phase plots showing (left) the logarithmic kernel distance of the NTK from initialization and (right) the corresponding logarithmic loss as functions of the relative scale $\lambda$ and the absolute scale. (Top to bottom) Different time steps during training $t = 1, t = 400, t = 4000$.

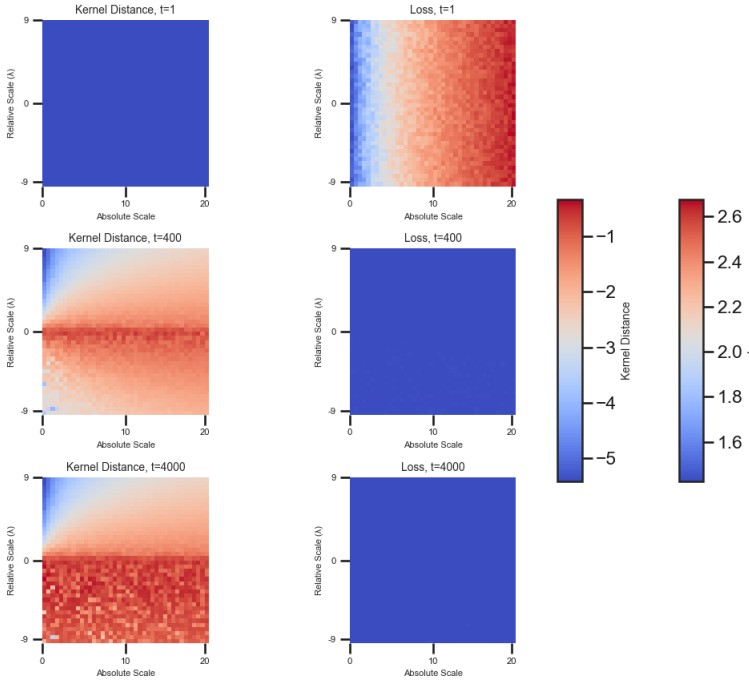

Figure 10: Funnel network: Phase plots showing (left) the logarithmic kernel distance of the NTK from initialization and (right) the corresponding logarithmic loss as functions of the relative scale $\lambda$ and the absolute scale. (Top to bottom) Different time steps during training $t = 1$, $t = 400$, $t = 4000$.

## B   APPENDIX: EXACT LEARNING DYNAMICS WITH PRIOR KNOWLEDGE

### B.1   APPENDIX: FUKUMIZU APPROACH

**Lemma B.1.** *We introduce the variables*

$$\mathbf{Q} = \begin{bmatrix} \mathbf{W}_1^T \\ \mathbf{W}_2 \end{bmatrix} \quad and \quad \mathbf{Q}\mathbf{Q}^T = \begin{bmatrix} \mathbf{W}_1^T\mathbf{W}_1 & \mathbf{W}_1^T\mathbf{W}_2^T \\ \mathbf{W}_2\mathbf{W}_1 & \mathbf{W}_2\mathbf{W}_2^T \end{bmatrix}. \tag{23}$$

*Defining*

$$\mathbf{F} = \begin{bmatrix} -\frac{\lambda}{2}I & (\tilde{\Sigma}^{yx})^T \\ \tilde{\Sigma}^{yx} & \frac{\lambda}{2}I \end{bmatrix}, \tag{24}$$

*the gradient flow dynamics of $\mathbf{Q}\mathbf{Q}^T(t)$ can be written as a differential matrix Riccati equation*

$$\tau\frac{d}{dt}(\mathbf{Q}\mathbf{Q}^T) = \mathbf{F}\mathbf{Q}\mathbf{Q}^T + \mathbf{Q}\mathbf{Q}^T\mathbf{F} - (\mathbf{Q}\mathbf{Q}^T)^2. \tag{25}$$

*Proof.* We introduce the variables

$$\mathbf{Q} = \begin{bmatrix} \mathbf{W}_1^T \\ \mathbf{W}_2 \end{bmatrix} \quad and \quad \mathbf{Q}\mathbf{Q}^T = \begin{bmatrix} \mathbf{W}_1^T\mathbf{W}_1 & \mathbf{W}_1^T\mathbf{W}_2^T \\ \mathbf{W}_2\mathbf{W}_1 & \mathbf{W}_2\mathbf{W}_2^T \end{bmatrix}. \tag{26}$$

We compute the time derivative

$$\tau\frac{d}{dt}(\mathbf{Q}\mathbf{Q}^T) = \tau \begin{bmatrix} \frac{d\mathbf{W}_1^T}{dt}\mathbf{W}_1 + \mathbf{W}_1^T\frac{d\mathbf{W}_1}{dt} & \frac{d\mathbf{W}_1^T}{dt}\mathbf{W}_2 + \mathbf{W}_1^T\frac{d\mathbf{W}_2}{dt} \\ \frac{d\mathbf{W}_2}{dt}\mathbf{W}_1 + \mathbf{W}_2\frac{d\mathbf{W}_1}{dt} & \frac{d\mathbf{W}_2^T}{dt}\mathbf{W}_2 + \mathbf{W}_2^T\frac{d\mathbf{W}_2}{dt} \end{bmatrix}. \tag{27}$$

Using equations 18 and 19, we compute the four quadrants separately giving

$$\tau\left(\frac{d\mathbf{W}_1^T}{dt}\mathbf{W}_1 + \mathbf{W}_1^T\frac{d\mathbf{W}_1}{dt}\right) = \tag{28}$$

$$= (\Sigma^{yx} - \mathbf{W}_2\mathbf{W}_1)^T\mathbf{W}_2\mathbf{W}_1 + \mathbf{W}_1^T\mathbf{W}_2^T(\Sigma^{yx} - \mathbf{W}_2\mathbf{W}_1) \tag{29}$$

$$= (\Sigma^{yx})^T\mathbf{W}_2\mathbf{W}_1 + \mathbf{W}_1^T\mathbf{W}_2^T\Sigma^{yx} - \mathbf{W}_1^T\mathbf{W}_2^T\mathbf{W}_2\mathbf{W}_1 - (\mathbf{W}_2\mathbf{W}_1)^T\mathbf{W}_2\mathbf{W}_1 \tag{30}$$

$$= (\Sigma^{yx})^T\mathbf{W}_2\mathbf{W}_1 + \mathbf{W}_1^T\mathbf{W}_2^T\Sigma^{yx} - \mathbf{W}_1^T\mathbf{W}_2^T\mathbf{W}_2\mathbf{W}_1 - \mathbf{W}_1^T\mathbf{W}_1\mathbf{W}_1^T\mathbf{W}_1 - \lambda\mathbf{W}_1^T\mathbf{W}_1, \tag{31}$$

$$\tau\left(\frac{d\mathbf{W}_1^T}{dt}\mathbf{W}_2^T + \mathbf{W}_1^T\frac{d\mathbf{W}_2^T}{dt}\right) = \tag{32}$$

$$= (\Sigma^{yx} - \mathbf{W}_2\mathbf{W}_1)^T\mathbf{W}_2\mathbf{W}_2^T + \mathbf{W}_1^T\mathbf{W}_1(\Sigma^{yx} - \mathbf{W}_2\mathbf{W}_1)^T \tag{33}$$

$$= (\Sigma^{yx})^T\mathbf{W}_2\mathbf{W}_2^T + \mathbf{W}_1^T\mathbf{W}_1(\Sigma^{yx})^T - \mathbf{W}_1^T\mathbf{W}_1\mathbf{W}_1^T\mathbf{W}_2^T - \mathbf{W}_1^T\mathbf{W}_2^T\mathbf{W}_2\mathbf{W}_2^T, \tag{34}$$

$$\tau\left(\frac{d\mathbf{W}_2}{dt}\mathbf{W}_1 + \mathbf{W}_2\frac{d\mathbf{W}_1}{dt}\right) = \tag{35}$$

$$= (\Sigma^{yx} - \mathbf{W}_2\mathbf{W}_1)\mathbf{W}_1^T\mathbf{W}_1 + \mathbf{W}_2\mathbf{W}_2^T(\Sigma^{yx} - \mathbf{W}_2\mathbf{W}_1) \tag{36}$$

$$= \Sigma^{yx}\mathbf{W}_1^T\mathbf{W}_1 + \mathbf{W}_2\mathbf{W}_2^T\Sigma^{yx} - \mathbf{W}_2\mathbf{W}_2^T\mathbf{W}_2\mathbf{W}_1 - \mathbf{W}_2\mathbf{W}_1\mathbf{W}_1^T\mathbf{W}_1, \tag{37}$$

$$\tau\left(\frac{d\mathbf{W}_2}{dt}\mathbf{W}_2^T + \mathbf{W}_2\frac{d\mathbf{W}_2^T}{dt}\right) = \tag{38}$$

$$(\tilde{\Sigma}^{yx} - \mathbf{W}_2\mathbf{W}_1)\mathbf{W}_1^T\mathbf{W}_2^T + \mathbf{W}_2\mathbf{W}_1(\tilde{\Sigma}^{yx} - \mathbf{W}_2\mathbf{W}_1)^T \tag{39}$$

$$= \tilde{\Sigma}^{yx}\mathbf{W}_1^T\mathbf{W}_2^T + \mathbf{W}_2\mathbf{W}_1(\tilde{\Sigma}^{yx})^T - \mathbf{W}_2\mathbf{W}_1\mathbf{W}_1^T\mathbf{W}_2^T - \mathbf{W}_2\mathbf{W}_1(\mathbf{W}_2\mathbf{W}_1)^T \tag{40}$$

$$= \tilde{\Sigma}^{yx}\mathbf{W}_1^T\mathbf{W}_2^T + \mathbf{W}_2\mathbf{W}_1(\tilde{\Sigma}^{yx})^T - \mathbf{W}_2\mathbf{W}_1\mathbf{W}_1^T\mathbf{W}_2^T - \mathbf{W}_2\mathbf{W}_1\mathbf{W}_1^T\mathbf{W}_2^T \tag{41}$$

$$= \tilde{\Sigma}^{yx}\mathbf{W}_1^T\mathbf{W}_2^T + \mathbf{W}_2\mathbf{W}_1(\tilde{\Sigma}^{yx})^T - \mathbf{W}_2\mathbf{W}_1\mathbf{W}_1^T\mathbf{W}_2^T - \mathbf{W}_2\mathbf{W}_2^T\mathbf{W}_2\mathbf{W}_2^T + \lambda\mathbf{W}_2\mathbf{W}_2^T. \tag{42}$$

Defining

$$\mathbf{F} = \begin{bmatrix} -\frac{\lambda}{2}I & (\tilde{\Sigma}^{yx})^T \\ \tilde{\Sigma}^{yx} & \frac{\lambda}{2}I \end{bmatrix}, \tag{43}$$

the gradient flow dynamics of $\mathbf{Q}\mathbf{Q}^T(t)$ can be written as a differential matrix Riccati equation

$$\tau\frac{d}{dt}(\mathbf{Q}\mathbf{Q}^T) = \mathbf{F}\mathbf{Q}\mathbf{Q}^T + \mathbf{Q}\mathbf{Q}^T\mathbf{F} - (\mathbf{Q}\mathbf{Q}^T)^2. \tag{44}$$

We write $\tau\frac{d}{dt}(\mathbf{Q}\mathbf{Q}^T)$ for completeness

$$\tau\frac{d}{dt}(\mathbf{Q}\mathbf{Q}^T) = \begin{bmatrix} -\frac{\lambda}{2} & (\tilde{\Sigma}^{yx})^T \\ \tilde{\Sigma}^{yx} & \frac{\lambda}{2} \end{bmatrix}\begin{bmatrix} \mathbf{W}_1^T\mathbf{W}_1 & \mathbf{W}_1^T\mathbf{W}_2 \\ \mathbf{W}_2\mathbf{W}_1 & \mathbf{W}_2\mathbf{W}_2^T \end{bmatrix} + \begin{bmatrix} \mathbf{W}_1^T\mathbf{W}_1 & \mathbf{W}_1^T\mathbf{W}_2 \\ \mathbf{W}_2\mathbf{W}_1 & \mathbf{W}_2\mathbf{W}_2^T \end{bmatrix}^T\begin{bmatrix} -\frac{\lambda}{2} & (\tilde{\Sigma}^{yx})^T \\ \tilde{\Sigma}^{yx} & \frac{\lambda}{2} \end{bmatrix}$$
$$- \begin{bmatrix} \mathbf{W}_1^T\mathbf{W}_1 & \mathbf{W}_1^T\mathbf{W}_2 \\ \mathbf{W}_2\mathbf{W}_1 & \mathbf{W}_2\mathbf{W}_2^T \end{bmatrix}^2 \tag{45}$$

$$= \begin{bmatrix} -\frac{\lambda}{2} & (\tilde{\Sigma}^{yx})^T \\ \tilde{\Sigma}^{yx} & \frac{\lambda}{2} \end{bmatrix}\begin{bmatrix} \mathbf{W}_1^T\mathbf{W}_1 & \mathbf{W}_1^T\mathbf{W}_2 \\ \mathbf{W}_2\mathbf{W}_1 & \mathbf{W}_2\mathbf{W}_2^T \end{bmatrix} + \begin{bmatrix} \mathbf{W}_1^T\mathbf{W}_1 & \mathbf{W}_1^T\mathbf{W}_2 \\ \mathbf{W}_2\mathbf{W}_1 & \mathbf{W}_2\mathbf{W}_2^T \end{bmatrix}^T\begin{bmatrix} -\frac{\lambda}{2} & (\tilde{\Sigma}^{yx})^T \\ \tilde{\Sigma}^{yx} & \frac{\lambda}{2} \end{bmatrix}$$
$$- \begin{bmatrix} \mathbf{W}_1^T\mathbf{W}_1 & \mathbf{W}_1^T\mathbf{W}_2 \\ \mathbf{W}_2\mathbf{W}_1 & \mathbf{W}_2\mathbf{W}_2^T \end{bmatrix}\begin{bmatrix} \mathbf{W}_1^T\mathbf{W}_1 & \mathbf{W}_1^T\mathbf{W}_2 \\ \mathbf{W}_2\mathbf{W}_1 & \mathbf{W}_2\mathbf{W}_2^T \end{bmatrix} \tag{46}$$

$$= \begin{bmatrix} -\frac{\lambda}{2}\mathbf{W}_1^T\mathbf{W}_1 + (\tilde{\Sigma}^{yx})^T\mathbf{W}_2\mathbf{W}_1 & -\frac{\lambda}{2}\mathbf{W}_1^T\mathbf{W}_2 + (\tilde{\Sigma}^{yx})^T\mathbf{W}_2\mathbf{W}_2^T \\ \tilde{\Sigma}^{yx}\mathbf{W}_1^T\mathbf{W}_1 + \frac{\lambda}{2}\mathbf{W}_2\mathbf{W}_1 & \tilde{\Sigma}^{yx}\mathbf{W}_1^T\mathbf{W}_2^T + \frac{\lambda}{2}\mathbf{W}_2\mathbf{W}_2^T \end{bmatrix}$$
$$+ \begin{bmatrix} -\frac{\lambda}{2}\mathbf{W}_1^T\mathbf{W}_1 + \mathbf{W}_1^T\mathbf{W}_1(\tilde{\Sigma}^{yx})^T & \frac{\lambda}{2}\mathbf{W}_1^T\mathbf{W}_2 + \mathbf{W}_1^T\mathbf{W}_2(\tilde{\Sigma}^{yx})^T \\ -\frac{\lambda}{2}\mathbf{W}_2^T\mathbf{W}_1 + \mathbf{W}_2\mathbf{W}_1(\tilde{\Sigma}^{yx})^T & \frac{\lambda}{2}\mathbf{W}_2\mathbf{W}_2^T + \mathbf{W}_2\mathbf{W}_2^T(\tilde{\Sigma}^{yx})^T \end{bmatrix}$$
$$- \begin{bmatrix} \mathbf{W}_1^T\mathbf{W}_1 & \mathbf{W}_1^T\mathbf{W}_2 \\ \mathbf{W}_2\mathbf{W}_1 & \mathbf{W}_2\mathbf{W}_2^T \end{bmatrix}\begin{bmatrix} \mathbf{W}_1^T\mathbf{W}_1 & \mathbf{W}_1^T\mathbf{W}_2 \\ \mathbf{W}_2\mathbf{W}_1 & \mathbf{W}_2\mathbf{W}_2^T \end{bmatrix} \tag{47}$$

$$
\begin{aligned}
&= \begin{bmatrix} -\frac{\lambda}{2}\mathbf{W}_1^T\mathbf{W}_1 + (\tilde{\Sigma}^{yx})^T\mathbf{W}_2\mathbf{W}_1 & -\frac{\lambda}{2}\mathbf{W}_1^T\mathbf{W}_2 + (\tilde{\Sigma}^{yx})^T\mathbf{W}_2\mathbf{W}_2^T \\ \tilde{\Sigma}^{yx}\mathbf{W}_1^T\mathbf{W}_1 + \frac{\lambda}{2}\mathbf{W}_2\mathbf{W}_1 & \tilde{\Sigma}^{yx}\mathbf{W}_1^T\mathbf{W}_2^T + \frac{\lambda}{2}\mathbf{W}_2\mathbf{W}_2^T \end{bmatrix} \\
&+ \begin{bmatrix} -\frac{\lambda}{2}\mathbf{W}_1^T\mathbf{W}_1 + \mathbf{W}_1^T\mathbf{W}_1(\tilde{\Sigma}^{yx})^T & \frac{\lambda}{2}\mathbf{W}_1^T\mathbf{W}_2 + \mathbf{W}_1^T\mathbf{W}_2(\tilde{\Sigma}^{yx})^T \\ -\frac{\lambda}{2}\mathbf{W}_2^T\mathbf{W}_1 + \mathbf{W}_2\mathbf{W}_1(\tilde{\Sigma}^{yx})^T & \frac{\lambda}{2}\mathbf{W}_2\mathbf{W}_2^T + \mathbf{W}_2\mathbf{W}_2^T(\tilde{\Sigma}^{yx})^T \end{bmatrix} \\
&- \begin{bmatrix} \mathbf{W}_1^T\mathbf{W}_1\mathbf{W}_1^T\mathbf{W}_1 + \mathbf{W}_1^T\mathbf{W}_2\mathbf{W}_2^T\mathbf{W}_1 & \mathbf{W}_1^T\mathbf{W}_1\mathbf{W}_1^T\mathbf{W}_2 + \mathbf{W}_1^T\mathbf{W}_2\mathbf{W}_2^T\mathbf{W}_2 \\ \mathbf{W}_2\mathbf{W}_1\mathbf{W}_1^T\mathbf{W}_1 + \mathbf{W}_2\mathbf{W}_2^T\mathbf{W}_2\mathbf{W}_1 & \mathbf{W}_2\mathbf{W}_1\mathbf{W}_1^T\mathbf{W}_2 + \mathbf{W}_2\mathbf{W}_2^T\mathbf{W}_2\mathbf{W}_2^T \end{bmatrix}
\end{aligned}
\tag{45}
$$

$\square$

The four quadrants of 27 are equivalent to equations 31, 34, 37, and 42 respectively.

## B.2 $\mathbf{Q}\mathbf{Q}^T$ DIAGONALISATION

**Lemma B.2.** *If $\boldsymbol{F} = \boldsymbol{P}\boldsymbol{\Lambda}\boldsymbol{P}^T$ is symmetric and diagonalizable, then the matrix Riccati differential equation $\frac{d}{dt}(\mathbf{Q}\mathbf{Q}^T) = \mathbf{F}\mathbf{Q}\mathbf{Q}^T + \mathbf{Q}\mathbf{Q}^T\mathbf{F} - (\mathbf{Q}\mathbf{Q}^T)^2$ with initialization $\mathbf{Q}\mathbf{Q}^T(0) = \mathbf{Q}(0)\mathbf{Q}(0)^T$ has a unique solution for all $t \geq 0$, and the solution is given by*

$$
\mathbf{Q}\mathbf{Q}^T(t) = e^{\mathbf{F}\frac{t}{\tau}}\mathbf{Q}(0)\left[\mathbf{I} + \mathbf{Q}(0)^T\boldsymbol{P}\left(\frac{e^{2\boldsymbol{\Lambda}\frac{t}{\tau}} - \mathbf{I}}{2\boldsymbol{\Lambda}}\right)\boldsymbol{P}^T\mathbf{Q}(0)\right]^{-1}\mathbf{Q}(0)^T e^{\mathbf{F}\frac{t}{\tau}}.
\tag{48}
$$

*This is true even when there exists $\boldsymbol{\Lambda}_i = 0$.*

*Proof.* First we show that there exists a unique solution to the initial value problem stated. This is true by Picard-Lindelöf theorem. Now we show that the provided solution satisfies the ODE. Let $\boldsymbol{L} = e^{\mathbf{F}\frac{t}{\tau}}\mathbf{Q}(0)$ and $\boldsymbol{C} = \mathbf{I} + \mathbf{Q}(0)^T\boldsymbol{P}\left(\frac{e^{2\boldsymbol{\Lambda}\frac{t}{\tau}}-\mathbf{I}}{2\boldsymbol{\Lambda}}\right)\boldsymbol{P}^T\mathbf{Q}(0)$ such that solution $\mathbf{Q}\mathbf{Q}^T(t) = \boldsymbol{L}\boldsymbol{C}^{-1}\boldsymbol{L}^T$. The time derivative of $\mathbf{Q}\mathbf{Q}^T$ is then given by

$$
\frac{d}{dt}(\mathbf{Q}\mathbf{Q}^T) = \frac{d}{dt}(\boldsymbol{L})\boldsymbol{C}^{-1}\boldsymbol{L}^T + \boldsymbol{L}\frac{d}{dt}(\boldsymbol{C}^{-1})\boldsymbol{L}^T + \boldsymbol{L}\boldsymbol{C}^{-1}\frac{d}{dt}(\boldsymbol{L}^T)
\tag{49}
$$

Solving for these derivatives individually, we find

$$
\frac{d}{dt}(\boldsymbol{L}) = \frac{d}{dt}e^{\mathbf{F}\frac{t}{\tau}}\mathbf{Q}(0) = \boldsymbol{F}e^{\mathbf{F}\frac{t}{\tau}}\mathbf{Q}(0) = \boldsymbol{F}\boldsymbol{L}
\tag{50}
$$

$$
\frac{d}{dt}(\boldsymbol{C}^{-1}) = -\boldsymbol{C}^{-1}\frac{d}{dt}(\boldsymbol{C})\boldsymbol{C}^{-1} = -\boldsymbol{C}^{-1}\mathbf{Q}(0)^T\boldsymbol{P}\frac{d}{dt}\left(\frac{e^{2\boldsymbol{\Lambda}\frac{t}{\tau}}-\mathbf{I}}{2\boldsymbol{\Lambda}}\right)\boldsymbol{P}^T\mathbf{Q}(0)\boldsymbol{C}^{-1}
\tag{51}
$$

We consider the derivative of the fraction serpately,

$$
\frac{d}{dt}\left(\frac{e^{2\boldsymbol{\Lambda}\frac{t}{\tau}}-\mathbf{I}}{2\boldsymbol{\Lambda}}\right) = e^{2\boldsymbol{\Lambda}\frac{t}{\tau}}
\tag{52}
$$

this is true even in the limit as $\lambda_i \to 0$. Plugging these derivatives back in we see that the solution satisfies the ODE. Lastly, let $t = 0$, we see that the the solution satisfies the initial conditions. $\square$

## B.3 $\boldsymbol{F}$ DIAGONALIZATION

**Lemma B.3.** *The eigendecomposition of $\mathbf{F} = \mathbf{P}\boldsymbol{\Lambda}\mathbf{P}^T$ where*

$$
\mathbf{P} = \frac{1}{\sqrt{2}}\begin{pmatrix} \tilde{\boldsymbol{V}}(\tilde{\boldsymbol{G}}-\tilde{\boldsymbol{H}}\tilde{\boldsymbol{G}}) & \tilde{\boldsymbol{V}}(\tilde{\boldsymbol{G}}+\tilde{\boldsymbol{H}}\tilde{\boldsymbol{G}}) & \sqrt{2}\tilde{\boldsymbol{V}}_\perp \\ \tilde{\boldsymbol{U}}(\tilde{\boldsymbol{G}}+\tilde{\boldsymbol{H}}\tilde{\boldsymbol{G}}) & -\tilde{\boldsymbol{U}}(\tilde{\boldsymbol{G}}-\tilde{\boldsymbol{H}}\tilde{\boldsymbol{G}}) & \sqrt{2}\tilde{\boldsymbol{U}}_\perp \end{pmatrix}, \quad \boldsymbol{\Lambda} = \begin{pmatrix} \tilde{\boldsymbol{S}}_\lambda & 0 & 0 \\ 0 & -\tilde{\boldsymbol{S}}_\lambda & 0 \\ 0 & 0 & \boldsymbol{\lambda}_\perp \end{pmatrix}
\tag{53}
$$

*and the matrices $\tilde{\boldsymbol{S}}_\lambda$, $\boldsymbol{\lambda}_\perp$, $\tilde{\boldsymbol{H}}$, and $\tilde{\boldsymbol{G}}$ are the diagonal matrices defined as:*

$$
\tilde{\boldsymbol{S}}_\lambda = \sqrt{\tilde{\boldsymbol{S}}^2 + \frac{\lambda^2}{4}\mathbf{I}}, \quad \boldsymbol{\lambda}_\perp = \text{sgn}(N_o - N_i)\frac{\lambda}{2}\mathbf{I}, \quad \tilde{\boldsymbol{H}} = \text{sgn}(\lambda)\sqrt{\frac{\tilde{\boldsymbol{S}}_\lambda - \tilde{\boldsymbol{S}}}{\tilde{\boldsymbol{S}}_\lambda + \tilde{\boldsymbol{S}}}}, \quad \tilde{\boldsymbol{G}} = \frac{1}{\sqrt{\mathbf{I} + \tilde{\boldsymbol{H}}^2}}.
\tag{54}
$$

Beyond the invertibility of $F$, notice from the equation (Fukumizu solution) we need to understand the relationship between $F$ and $Q(0)$. To do this the following lemma relates the structure between the SVD of the model with the SVD structure of the individual parameters.

*Proof.* We leave for the reader by computing

$$
\boldsymbol{F} = \boldsymbol{P}\boldsymbol{\Lambda}\boldsymbol{P}^T
\tag{55}
$$

$\square$

## B.4 SOLUTION UNEQUAL-INPUT-OUTPUT

**Theorem B.4.** *Under the assumptions of whitened inputs, 1, lambda-balanced weights 2, no bottleneck 3, the temporal dynamics of* $\mathbf{QQ}^T$ *are*

$$\mathbf{QQ}^T(t) = \begin{pmatrix} \textcolor{teal}{Z_1 A^{-1} Z_1^T} & \textcolor{orange}{Z_1 A^{-1} Z_2^T} \\ \textcolor{orange}{Z_2 A^{-1} Z_1^T} & \textcolor{blue}{Z_2 A^{-1} Z_2^T} \end{pmatrix},$$

*where the variables* $Z_1 \in \mathbb{R}^{N_i \times N_h}$, $Z_2 \in \mathbb{R}^{N_o \times N_h}$, *and* $A \in \mathbb{R}^{N_h \times N_h}$ *are defined as*

$$Z_1(t) = \frac{1}{2}\tilde{V}(\tilde{G} - \tilde{H}\tilde{G})e^{\tilde{S}_\lambda \frac{t}{\tau}}B^T - \frac{1}{2}\tilde{V}(\tilde{G} + \tilde{H}\tilde{G})e^{-\tilde{S}_\lambda \frac{t}{\tau}}C^T + \tilde{V}_\perp e^{\lambda_\perp \frac{t}{\tau}}\tilde{V}_\perp^T W_1(0)^T \tag{56}$$

$$Z_2(t) = \frac{1}{2}\tilde{U}(\tilde{G} + \tilde{H}\tilde{G})e^{\tilde{S}_\lambda \frac{t}{\tau}}B^T + \frac{1}{2}\tilde{U}(\tilde{G} - \tilde{H}\tilde{G})e^{-\tilde{S}_\lambda \frac{t}{\tau}}C^T + \tilde{U}_\perp e^{\lambda_\perp \frac{t}{\tau}}\tilde{U}_\perp^T W_2(0) \tag{57}$$

$$A(t) = \mathbf{I} + B\left(\frac{e^{2\tilde{S}_\lambda \frac{t}{\tau}} - \mathbf{I}}{4\tilde{S}_\lambda}\right)B^T - C\left(\frac{e^{-2\tilde{S}_\lambda \frac{t}{\tau}} - \mathbf{I}}{4\tilde{S}_\lambda}\right)C^T + \mathbf{W}_2(0)^T\tilde{U}_\perp\left(\frac{e^{\lambda_\perp \frac{t}{\tau}} - \mathbf{I}}{\lambda_\perp}\right)\tilde{U}_\perp^T\mathbf{W}_2(0)$$

$$+ \mathbf{W}_1(0)\tilde{V}_\perp\left(\frac{e^{\lambda_\perp \frac{t}{\tau}} - \mathbf{I}}{\lambda_\perp}\right)\tilde{V}_\perp^T\mathbf{W}_1(0)^T \tag{58}$$

*Proof.* We start and use the diagonalization of $\mathbf{F}$ to rewrite the matrix exponential of $\mathbf{F}$ and $\mathbf{F}$. Note that $\mathbf{P}^T\mathbf{P} = \mathbf{P}\mathbf{P}^T = \mathbf{I}$ and therefore $\mathbf{P}^T = \mathbf{P}^{-1}$.

$$e^{\mathbf{F}\frac{t}{\tau}} = \mathbf{P}e^{\mathbf{\Gamma}}\mathbf{P}^T$$

$$= \frac{1}{\sqrt{2}}\begin{bmatrix}\tilde{V}(\tilde{G} - \tilde{H}\tilde{G}) & \tilde{V}(\tilde{G} + \tilde{H}\tilde{G}) & \sqrt{2}\mathbf{V}_\perp \\ \tilde{U}(\tilde{G} + \tilde{H}\tilde{G}) & -\tilde{U}(\tilde{G} - \tilde{H}\tilde{G}) & \sqrt{2}\mathbf{U}_\perp\end{bmatrix}\begin{bmatrix}e^{\tilde{S}_\lambda \frac{t}{\tau}} & 0 & 0 \\ 0 & e^{-\tilde{S}_\lambda \frac{t}{\tau}} & 0 \\ 0 & 0 & e^{\lambda_\perp \frac{t}{\tau}}\end{bmatrix}\frac{1}{\sqrt{2}}\begin{bmatrix}\tilde{V}(\tilde{G} - \tilde{H}\tilde{G}) & \tilde{V}(\tilde{G} + \tilde{H}\tilde{G}) & \sqrt{2}\mathbf{V}_\perp \\ \tilde{U}(\tilde{G} + \tilde{H}\tilde{G}) & -\tilde{U}(\tilde{G} - \tilde{H}\tilde{G}) & \sqrt{2}\mathbf{U}_\perp\end{bmatrix}^T$$

$$= \frac{1}{\sqrt{2}}\begin{bmatrix}\tilde{V}(\tilde{G} - \tilde{H}\tilde{G}) & \tilde{V}(\tilde{G} + \tilde{H}\tilde{G}) \\ \tilde{U}(\tilde{G} - \tilde{H}\tilde{G}) & -\tilde{U}(\tilde{G} + \tilde{H}\tilde{G})\end{bmatrix}\begin{bmatrix}e^{\tilde{S}_\lambda \frac{t}{\tau}} & 0 \\ 0 & e^{-\tilde{S}_\lambda \frac{t}{\tau}}\end{bmatrix}\frac{1}{\sqrt{2}}\begin{bmatrix}\tilde{V}(\tilde{G} - \tilde{H}\tilde{G}) & \tilde{V}(\tilde{G} + \tilde{H}\tilde{G}) \\ \tilde{U}(\tilde{G} + \tilde{H}\tilde{G}) & -\tilde{U}(\tilde{G} - \tilde{H}\tilde{G})\end{bmatrix}^T + 2\frac{1}{\sqrt{2}}\begin{bmatrix}\tilde{V}_\perp \\ \tilde{U}_\perp\end{bmatrix}e^{\lambda_\perp \frac{t}{\tau}}\frac{1}{\sqrt{2}}\begin{bmatrix}\tilde{V}_\perp \\ \tilde{U}_\perp\end{bmatrix}^T$$

$$= \mathbf{O}e^{\mathbf{\Lambda}\frac{t}{\tau}}\mathbf{O} + 2\mathbf{M}e^{\lambda_\perp \frac{t}{\tau}}\mathbf{M}^T. \tag{59}$$

$$e^{\mathbf{F}\frac{t}{\tau}}\mathbf{F}^{-1}e^{\mathbf{F}\frac{t}{\tau}} - \mathbf{F}^{-1} = \mathbf{O}e^{\mathbf{\Lambda}\frac{t}{\tau}}\mathbf{O}^T\mathbf{O}\mathbf{\Lambda}^{-1}\mathbf{O}^T\mathbf{O}e^{\mathbf{\Lambda}\frac{t}{\tau}}\mathbf{O}^T - \mathbf{O}\mathbf{\Lambda}^{-1}\mathbf{O}^T + \mathbf{M}(e^{\lambda_\perp \frac{t}{\tau}} - \mathbf{I})(\lambda_\perp)^{-1}\mathbf{M}^T. \tag{60}$$

$$\mathbf{F} = \mathbf{O}\mathbf{\Lambda}\mathbf{O}^T + 2\mathbf{M}\lambda_\perp\mathbf{M}^T \tag{61}$$

Where $M = \frac{1}{\sqrt{2}}\begin{bmatrix}\tilde{\mathbf{V}}_\perp \\ \tilde{\mathbf{U}}_\perp\end{bmatrix}^T$. Placing these expressions into equation 48 gives

$$\mathbf{QQ}^T(t) = \left[\mathbf{O}e^{\mathbf{\Lambda}\frac{t}{\tau}}\mathbf{O}^T + 2\mathbf{M}e^{\lambda_\perp \frac{t}{\tau}}\mathbf{M}^T\right]\mathbf{Q}(0)$$

$$\left[\mathbf{I} + \frac{1}{2}\mathbf{Q}(0)^T\left(\mathbf{O}\left(e^{2\mathbf{\Lambda}\frac{t}{\tau}} - \mathbf{I}\right)\mathbf{\Lambda}^{-1}\mathbf{O}^T + \mathbf{M}(e^{\lambda_\perp \frac{t}{\tau}} - \mathbf{I})\lambda_\perp^{-1}\mathbf{M}^T\right)\mathbf{Q}(0)\right]^{-1} \tag{62}$$

$$\mathbf{Q}(0)^T\left[\mathbf{O}e^{\mathbf{\Lambda}\frac{t}{\tau}}\mathbf{O}^T + 2\mathbf{M}e^{\lambda_\perp \frac{t}{\tau}}\mathbf{M}^T\right]^T$$

$$O^T Q(0) = \frac{1}{\sqrt{2}}\begin{pmatrix}\tilde{V}(\tilde{G} - \tilde{H}\tilde{G}) & \tilde{V}(\tilde{G} + \tilde{H}\tilde{G}) \\ \tilde{U}(\tilde{G} + \tilde{H}\tilde{G}) & -\tilde{U}(\tilde{G} - \tilde{H}\tilde{G})\end{pmatrix}^T\begin{pmatrix}W_1^T(0) \\ W_2(0)\end{pmatrix}$$

$$= \frac{1}{\sqrt{2}}\begin{pmatrix}(\tilde{G} - \tilde{H}\tilde{G})\tilde{V}^T W_1^T(0) + (\tilde{G} + \tilde{H}\tilde{G})\tilde{U}^T W_2(0) \\ (\tilde{G} + \tilde{H}\tilde{G})\tilde{V}^T W_1^T(0) - (\tilde{G} - \tilde{H}\tilde{G})\tilde{U}^T W_2(0)\end{pmatrix}$$

$$= \frac{1}{\sqrt{2}}\begin{pmatrix}B^T \\ -C^T\end{pmatrix} \tag{63}$$

where

$$\mathbf{B} = \mathbf{W}_2(0)^T\tilde{U}(\tilde{G} + \tilde{H}\tilde{G}) + \mathbf{W}_1(0)\tilde{V}(\tilde{G} - \tilde{H}\tilde{G}) \in \mathbb{R}^{N_h \times N_h} \tag{64}$$

$$\mathbf{C} = \mathbf{W}_2(0)^T\tilde{U}(\tilde{G} - \tilde{H}\tilde{G}) - \mathbf{W}_1(0)\tilde{V}(\tilde{G} + \tilde{H}\tilde{G}) \in \mathbb{R}^{N_h \times N_h} \tag{65}$$

$$\boldsymbol{O}e^{\boldsymbol{\Lambda} t/\tau} = \frac{1}{\sqrt{2}} \begin{pmatrix} \tilde{\boldsymbol{V}}(\tilde{\boldsymbol{G}} - \tilde{\boldsymbol{H}}\tilde{\boldsymbol{G}}) & \tilde{\boldsymbol{V}}(\tilde{\boldsymbol{G}} + \tilde{\boldsymbol{H}}\tilde{\boldsymbol{G}}) \\ \tilde{\boldsymbol{U}}(\tilde{\boldsymbol{G}} + \tilde{\boldsymbol{H}}\tilde{\boldsymbol{G}}) & -\tilde{\boldsymbol{U}}(\tilde{\boldsymbol{G}} - \tilde{\boldsymbol{H}}\tilde{\boldsymbol{G}}) \end{pmatrix} \begin{pmatrix} e^{\tilde{\boldsymbol{S}}_\lambda \frac{t}{\tau}} & 0 \\ 0 & e^{-\tilde{\boldsymbol{S}}_\lambda \frac{t}{\tau}} \end{pmatrix}$$

$$= \frac{1}{\sqrt{2}} \begin{pmatrix} \tilde{\boldsymbol{V}}(\tilde{\boldsymbol{G}} - \tilde{\boldsymbol{H}}\tilde{\boldsymbol{G}})e^{\tilde{\boldsymbol{S}}_\lambda \frac{t}{\tau}} & \tilde{\boldsymbol{V}}(\tilde{\boldsymbol{G}} + \tilde{\boldsymbol{H}}\tilde{\boldsymbol{G}})e^{-\tilde{\boldsymbol{S}}_\lambda \frac{t}{\tau}} \\ \tilde{\boldsymbol{U}}(\tilde{\boldsymbol{G}} + \tilde{\boldsymbol{H}}\tilde{\boldsymbol{G}})e^{\tilde{\boldsymbol{S}}_\lambda \frac{t}{\tau}} & -\tilde{\boldsymbol{U}}(\tilde{\boldsymbol{G}} - \tilde{\boldsymbol{H}}\tilde{\boldsymbol{G}})e^{-\tilde{\boldsymbol{S}}_\lambda \frac{t}{\tau}} \end{pmatrix} \tag{66}$$

$$\boldsymbol{O}e^{\boldsymbol{\Lambda} t/\tau}\boldsymbol{O}^T\boldsymbol{Q}(0) = \frac{1}{2} \begin{pmatrix} \tilde{\boldsymbol{V}}(\tilde{\boldsymbol{G}} - \tilde{\boldsymbol{H}}\tilde{\boldsymbol{G}})e^{\tilde{\boldsymbol{S}}_\lambda \frac{t}{\tau}} & \tilde{\boldsymbol{V}}(\tilde{\boldsymbol{G}} + \tilde{\boldsymbol{H}}\tilde{\boldsymbol{G}})e^{-\tilde{\boldsymbol{S}}_\lambda \frac{t}{\tau}} \\ \tilde{\boldsymbol{U}}(\tilde{\boldsymbol{G}} + \tilde{\boldsymbol{H}}\tilde{\boldsymbol{G}})e^{\tilde{\boldsymbol{S}}_\lambda \frac{t}{\tau}} & -\tilde{\boldsymbol{U}}(\tilde{\boldsymbol{G}} - \tilde{\boldsymbol{H}}\tilde{\boldsymbol{G}})e^{-\tilde{\boldsymbol{S}}_\lambda \frac{t}{\tau}} \end{pmatrix} \begin{pmatrix} \boldsymbol{B}^T \\ -\boldsymbol{C}^T \end{pmatrix}$$

$$= \frac{1}{2} \begin{pmatrix} \tilde{\boldsymbol{V}}(\tilde{\boldsymbol{G}} - \tilde{\boldsymbol{H}}\tilde{\boldsymbol{G}})e^{\tilde{\boldsymbol{S}}_\lambda \frac{t}{\tau}}\boldsymbol{B}^T - \tilde{\boldsymbol{V}}(\tilde{\boldsymbol{G}} + \tilde{\boldsymbol{H}}\tilde{\boldsymbol{G}})e^{-\tilde{\boldsymbol{S}}_\lambda \frac{t}{\tau}}\boldsymbol{C}^T \\ \tilde{\boldsymbol{U}}(\tilde{\boldsymbol{G}} + \tilde{\boldsymbol{H}}\tilde{\boldsymbol{G}})e^{\tilde{\boldsymbol{S}}_\lambda \frac{t}{\tau}}\boldsymbol{B}^T + \tilde{\boldsymbol{U}}(\tilde{\boldsymbol{G}} - \tilde{\boldsymbol{H}}\tilde{\boldsymbol{G}})e^{-\tilde{\boldsymbol{S}}_\lambda \frac{t}{\tau}}\boldsymbol{C}^T \end{pmatrix} \tag{67}$$

$$2\mathbf{M}e^{\boldsymbol{\lambda}_\perp \frac{t}{\tau}}\mathbf{M}^T\mathbf{Q}(0) = 2\frac{1}{\sqrt{2}} \begin{bmatrix} \tilde{\mathbf{V}}_\perp \\ \tilde{\mathbf{U}}_\perp \end{bmatrix} \begin{bmatrix} e^{\boldsymbol{\lambda}_\perp \frac{t}{\tau}} & 0 \\ 0 & e^{\boldsymbol{\lambda}_\perp \frac{t}{\tau}} \end{bmatrix} \frac{1}{\sqrt{2}} \begin{bmatrix} \tilde{\mathbf{V}}_\perp \\ \tilde{\mathbf{U}}_\perp \end{bmatrix}^T \begin{bmatrix} \mathbf{W}_1(0)^T \\ \mathbf{W}_2(0) \end{bmatrix}$$

$$= \begin{bmatrix} \tilde{\mathbf{V}}_\perp e^{\boldsymbol{\lambda}_\perp \frac{t}{\tau}}\tilde{\mathbf{V}}_\perp^T & 0 \\ 0 & \tilde{\mathbf{U}}_\perp e^{\boldsymbol{\lambda}_\perp \frac{t}{\tau}}\tilde{\mathbf{U}}_\perp^T \end{bmatrix} \begin{bmatrix} \mathbf{W}_1(0)^T \\ \mathbf{W}_2(0) \end{bmatrix}$$

$$= \begin{bmatrix} \tilde{\mathbf{V}}_\perp e^{\boldsymbol{\lambda}_\perp \frac{t}{\tau}}\tilde{\mathbf{V}}_\perp^T\mathbf{W}_1(0)^T \\ \tilde{\mathbf{U}}_\perp e^{\boldsymbol{\lambda}_\perp \frac{t}{\tau}}\tilde{\mathbf{U}}_\perp^T\mathbf{W}_2(0) \end{bmatrix} \tag{68}$$

Putting it together we get the expressions for $\boldsymbol{Z_1}(t)$ and $\boldsymbol{Z_2}(t)$

$$\left[\mathbf{O}e^{\boldsymbol{\Lambda} \frac{t}{\tau}}\mathbf{O}^T + 2\mathbf{M}e^{\boldsymbol{\lambda}_\perp \frac{t}{\tau}}\mathbf{M}^T\right]\mathbf{Q}(0) =$$

$$= \frac{1}{2} \begin{pmatrix} \tilde{\boldsymbol{V}}(\tilde{\boldsymbol{G}} - \tilde{\boldsymbol{H}}\tilde{\boldsymbol{G}})e^{\tilde{\boldsymbol{S}}_\lambda \frac{t}{\tau}}\boldsymbol{B}^T - \tilde{\boldsymbol{V}}(\tilde{\boldsymbol{G}} + \tilde{\boldsymbol{H}}\tilde{\boldsymbol{G}})e^{-\tilde{\boldsymbol{S}}_\lambda \frac{t}{\tau}}\boldsymbol{C}^T \\ \tilde{\boldsymbol{U}}(\tilde{\boldsymbol{G}} + \tilde{\boldsymbol{H}}\tilde{\boldsymbol{G}})e^{\tilde{\boldsymbol{S}}_\lambda \frac{t}{\tau}}\boldsymbol{B}^T + \tilde{\boldsymbol{U}}(\tilde{\boldsymbol{G}} - \tilde{\boldsymbol{H}}\tilde{\boldsymbol{G}})e^{-\tilde{\boldsymbol{S}}_\lambda \frac{t}{\tau}}\boldsymbol{C}^T \end{pmatrix} + \begin{bmatrix} \tilde{\mathbf{V}}_\perp e^{\boldsymbol{\lambda}_\perp \frac{t}{\tau}}\tilde{\mathbf{V}}_\perp^T\mathbf{W}_1(0)^T \\ \tilde{\mathbf{U}}_\perp e^{\boldsymbol{\lambda}_\perp \frac{t}{\tau}}\tilde{\mathbf{U}}_\perp^T\mathbf{W}_2(0) \end{bmatrix} \tag{69}$$

$$\boldsymbol{Z_1}(t) = \frac{1}{2}\tilde{\boldsymbol{V}}(\tilde{\boldsymbol{G}} - \tilde{\boldsymbol{H}}\tilde{\boldsymbol{G}})e^{\tilde{\boldsymbol{S}}_\lambda \frac{t}{\tau}}\boldsymbol{B}^T - \frac{1}{2}\tilde{\boldsymbol{V}}(\tilde{\boldsymbol{G}} + \tilde{\boldsymbol{H}}\tilde{\boldsymbol{G}})e^{-\tilde{\boldsymbol{S}}_\lambda \frac{t}{\tau}}\boldsymbol{C}^T + \tilde{\mathbf{V}}_\perp e^{\boldsymbol{\lambda}_\perp \frac{t}{\tau}}\tilde{\mathbf{V}}_\perp^T\mathbf{W}_1(0)^T \tag{70}$$

$$\boldsymbol{Z_2}(t) = \frac{1}{2}\tilde{\boldsymbol{U}}(\tilde{\boldsymbol{G}} + \tilde{\boldsymbol{H}}\tilde{\boldsymbol{G}})e^{\tilde{\boldsymbol{S}}_\lambda \frac{t}{\tau}}\boldsymbol{B}^T + \frac{1}{2}\tilde{\boldsymbol{U}}(\tilde{\boldsymbol{G}} - \tilde{\boldsymbol{H}}\tilde{\boldsymbol{G}})e^{-\tilde{\boldsymbol{S}}_\lambda \frac{t}{\tau}}\boldsymbol{C}^T + \tilde{\mathbf{U}}_\perp e^{\boldsymbol{\lambda}_\perp \frac{t}{\tau}}\tilde{\mathbf{U}}_\perp^T\mathbf{W}_2(0) \tag{71}$$

We now compute the terms inside the inverse

$$\mathbf{Q}(0)^T\mathbf{M}(e^{\boldsymbol{\lambda}_\perp \frac{t}{\tau}})\boldsymbol{\lambda}_\perp^{-1}\mathbf{M}^T\mathbf{Q}(0)$$

$$= \begin{bmatrix} \mathbf{W}_1(0) & \mathbf{W}_2(0)^T \end{bmatrix} \frac{1}{\sqrt{2}} \begin{bmatrix} \tilde{\mathbf{V}}_\perp \\ \tilde{\mathbf{U}}_\perp \end{bmatrix} \begin{bmatrix} e^{\boldsymbol{\lambda}_\perp \frac{t}{\tau}} & 0 \\ 0 & e^{\boldsymbol{\lambda}_\perp \frac{t}{\tau}} \end{bmatrix} \begin{bmatrix} \boldsymbol{\lambda}_\perp & 0 \\ 0 & \boldsymbol{\lambda}_\perp \end{bmatrix}^{-1} \frac{1}{\sqrt{2}} \begin{bmatrix} \tilde{\mathbf{V}}_\perp \\ \tilde{\mathbf{U}}_\perp \end{bmatrix}^T \begin{bmatrix} \mathbf{W}_1(0)^T \\ \mathbf{W}_2(0) \end{bmatrix}$$

$$= \begin{bmatrix} \mathbf{W}_1(0) & \mathbf{W}_2(0)^T \end{bmatrix} \begin{bmatrix} e^{\boldsymbol{\lambda}_\perp \frac{t}{\tau}}\boldsymbol{\lambda}_\perp^{-1}\tilde{\mathbf{V}}_\perp\tilde{\mathbf{V}}_\perp^T\mathbf{W}_1(0)^T \\ e^{\boldsymbol{\lambda}_\perp \frac{t}{\tau}}\boldsymbol{\lambda}_\perp^{-1}\tilde{\mathbf{U}}_\perp\tilde{\mathbf{U}}_\perp^T\mathbf{W}_2(0) \end{bmatrix}$$

$$= \left[\left(\mathbf{W}_1(0)\tilde{\mathbf{V}}_\perp e^{\boldsymbol{\lambda}_\perp \frac{t}{\tau}}\boldsymbol{\lambda}_\perp^{-1}\tilde{\mathbf{V}}_\perp^T\mathbf{W}_1(0)^T + \mathbf{W}_2(0)^T\tilde{\mathbf{U}}_\perp e^{\boldsymbol{\lambda}_\perp \frac{t}{\tau}}\boldsymbol{\lambda}_\perp^{-1}\tilde{\mathbf{U}}_\perp^T\mathbf{W}_2(0)\right)\right] \tag{72}$$

$$\mathbf{Q}(0)^T\mathbf{M}\boldsymbol{\lambda}_\perp^{-1}\mathbf{M}^T\mathbf{Q}(0) = 2 \begin{bmatrix} \mathbf{W}_1(0) & \mathbf{W}_2(0)^T \end{bmatrix} \frac{1}{\sqrt{2}} \begin{bmatrix} \tilde{\mathbf{V}}_\perp \\ \tilde{\mathbf{U}}_\perp \end{bmatrix} \begin{bmatrix} \boldsymbol{\lambda}_\perp & 0 \\ 0 & \boldsymbol{\lambda}_\perp \end{bmatrix}^{-1} \frac{1}{\sqrt{2}} \begin{bmatrix} \tilde{\mathbf{V}}_\perp \\ \tilde{\mathbf{U}}_\perp \end{bmatrix}^T \begin{bmatrix} \mathbf{W}_1(0)^T \\ \mathbf{W}_2(0) \end{bmatrix}$$

$$= \begin{bmatrix} \mathbf{W}_1(0) & \mathbf{W}_2(0)^T \end{bmatrix} \begin{bmatrix} \tilde{\mathbf{V}}_\perp \\ \tilde{\mathbf{U}}_\perp \end{bmatrix} \begin{bmatrix} \boldsymbol{\lambda}_\perp^{-1}\tilde{\mathbf{V}}_\perp\tilde{\mathbf{V}}_\perp^T\mathbf{W}_1(0)^T \\ \boldsymbol{\lambda}_\perp^{-1}\tilde{\mathbf{U}}_\perp\tilde{\mathbf{U}}_\perp^T\mathbf{W}_2(0) \end{bmatrix}$$

$$= \begin{bmatrix} \mathbf{W}_1(0)\tilde{\mathbf{V}}_\perp\boldsymbol{\lambda}_\perp^{-1}\tilde{\mathbf{V}}_\perp^T\mathbf{W}_1(0)^T + \mathbf{W}_2(0)^T\tilde{\mathbf{U}}_\perp\boldsymbol{\lambda}_\perp^{-1}\tilde{\mathbf{U}}_\perp^T\mathbf{W}_2(0) \end{bmatrix} \tag{73}$$

Now

$$\frac{1}{2}\mathbf{Q}(0)^T\mathbf{O}\left(e^{2\mathbf{\Lambda}\frac{t}{\tau}}-\mathbf{I}\right)\mathbf{\Lambda}^{-1}\mathbf{O}^T = \frac{1}{4}\left[\boldsymbol{B}-\boldsymbol{C}\right]\left(e^{\mathbf{\Lambda}\frac{t}{\tau}}-\mathbf{I}\right)\mathbf{\Lambda}^{-1}\begin{pmatrix}\boldsymbol{B}^T\\-\boldsymbol{C}^T\end{pmatrix}$$

$$= \frac{1}{4}\left(\boldsymbol{B}\left(e^{2\tilde{\boldsymbol{S}}_\lambda\frac{t}{\tau}}-\mathbf{I}\right)(\tilde{\boldsymbol{S}}_\lambda)^{-1}\boldsymbol{B}^T - \boldsymbol{C}\left(e^{-2\tilde{\boldsymbol{S}}_\lambda\frac{t}{\tau}}-\mathbf{I}\right)(\tilde{\boldsymbol{S}}_\lambda)^{-1}\boldsymbol{C}^T\right) \tag{74}$$

Putting it all together

$$\boldsymbol{A}(t) = \mathbf{I} + \boldsymbol{B}\left(\frac{e^{2\tilde{\boldsymbol{S}}_\lambda\frac{t}{\tau}}-\mathbf{I}}{4\tilde{\boldsymbol{S}}_\lambda}\right)\boldsymbol{B}^T - \boldsymbol{C}\left(\frac{e^{-2\tilde{\boldsymbol{S}}_\lambda\frac{t}{\tau}}-\mathbf{I}}{4\tilde{\boldsymbol{S}}_\lambda}\right)\boldsymbol{C}^T + \mathbf{W}_2(0)^T\tilde{\mathbf{U}}_\perp\left(\frac{e^{\boldsymbol{\lambda}_\perp\frac{t}{\tau}}-\mathbf{I}}{\boldsymbol{\lambda}_\perp}\right)\tilde{\mathbf{U}}_\perp^T\mathbf{W}_2(0)$$

$$+ \mathbf{W}_1(0)\tilde{\mathbf{V}}_\perp\left(\frac{e^{\boldsymbol{\lambda}_\perp\frac{t}{\tau}}-\mathbf{I}}{\boldsymbol{\lambda}_\perp}\right)\tilde{\mathbf{V}}_\perp^T\mathbf{W}_1(0)^T \tag{75}$$

So, final form:

$$\mathbf{Q}\mathbf{Q}^T(t) =$$

$$\left[\begin{pmatrix}\frac{1}{2}\tilde{V}(\tilde{G}-\tilde{H}\tilde{G})e^{\tilde{S}_\lambda\frac{t}{\tau}}B^T - \frac{1}{2}\tilde{V}(\tilde{G}+\tilde{H}\tilde{G})e^{-\tilde{S}_\lambda\frac{t}{\tau}}C^T + \tilde{V}_\perp e^{\boldsymbol{\lambda}_\perp\frac{t}{\tau}}\tilde{V}_\perp^T\mathbf{W}_1(0)^T\\\frac{1}{2}\tilde{U}(\tilde{G}+\tilde{H}\tilde{G})e^{\tilde{S}_\lambda\frac{t}{\tau}}B^T + \frac{1}{2}\tilde{U}(\tilde{G}-\tilde{H}\tilde{G})e^{-\tilde{S}_\lambda\frac{t}{\tau}}C^T + \tilde{U}_\perp e^{\boldsymbol{\lambda}_\perp\frac{t}{\tau}}\tilde{U}_\perp^T\mathbf{W}_2(0)\end{pmatrix}\right]$$

$$\left[\mathbf{I}+\frac{1}{4}\left(B\left(\frac{e^{2\tilde{S}_\lambda\frac{t}{\tau}}-\mathbf{I}}{\tilde{S}_\lambda}\right)B^T - C\left(\frac{e^{-2\tilde{S}_\lambda\frac{t}{\tau}}-\mathbf{I}}{\tilde{S}_\lambda}\right)C^T\right)\right.$$

$$\left.+\mathbf{W}_2(0)^T\tilde{\mathbf{U}}_\perp\left(\frac{e^{\boldsymbol{\lambda}_\perp\frac{t}{\tau}}-\mathbf{I}}{\boldsymbol{\lambda}_\perp}\right)\tilde{\mathbf{U}}_\perp^T\mathbf{W}_2(0) + \mathbf{W}_1(0)\tilde{\mathbf{V}}_\perp\left(\frac{e^{\boldsymbol{\lambda}_\perp\frac{t}{\tau}}-\mathbf{I}}{\boldsymbol{\lambda}_\perp}\right)\tilde{\mathbf{V}}_\perp^T\mathbf{W}_1(0)^T\right]^{-1}$$

$$\left[\begin{pmatrix}\frac{1}{2}\tilde{V}(\tilde{G}-\tilde{H}\tilde{G})e^{\tilde{S}_\lambda\frac{t}{\tau}}B^T - \frac{1}{2}\tilde{V}(\tilde{G}+\tilde{H}\tilde{G})e^{-\tilde{S}_\lambda\frac{t}{\tau}}C^T + \tilde{V}_\perp e^{\boldsymbol{\lambda}_\perp\frac{t}{\tau}}\tilde{V}_\perp^T\mathbf{W}_1(0)^T\\\frac{1}{2}\tilde{U}(\tilde{G}+\tilde{H}\tilde{G})e^{\tilde{S}_\lambda\frac{t}{\tau}}B^T + \frac{1}{2}\tilde{U}(\tilde{G}-\tilde{H}\tilde{G})e^{-\tilde{S}_\lambda\frac{t}{\tau}}C^T + \tilde{U}_\perp e^{\boldsymbol{\lambda}_\perp\frac{t}{\tau}}\tilde{U}_\perp^T\mathbf{W}_2(0)\end{pmatrix}\right]^T \tag{76}$$

$\square$

### B.5 STABLE SOLUTION UNEQUAL-INPUT-OUTPUT

**Theorem B.5.** *Given the assumptions of Theorem 4.3 further assuming that $\boldsymbol{B}$ is invertible and defining $e^{\boldsymbol{\lambda}_\perp\frac{t}{\tau}} = \mathrm{sgn}(N_o - N_i)\frac{\lambda}{2}$, the temporal evolution of $\mathbf{Q}\mathbf{Q}^T$ is described as follows:*

$$\mathbf{Q}\mathbf{Q}^T(t) = \boldsymbol{Z}\left[e^{-\tilde{\boldsymbol{S}}_\lambda\frac{t}{\tau}}\boldsymbol{B}^{-1}\boldsymbol{B}^{-T}e^{-\tilde{\boldsymbol{S}}_\lambda\frac{t}{\tau}}\right. \tag{77}$$

$$+\left(\frac{\mathbf{I}-e^{-2\tilde{\boldsymbol{S}}_\lambda\frac{t}{\tau}}}{4\tilde{\boldsymbol{S}}_\lambda}\right) - e^{-\tilde{\boldsymbol{S}}_\lambda\frac{t}{\tau}}\boldsymbol{B}^{-1}\boldsymbol{C}\left(\frac{e^{-\tilde{\boldsymbol{S}}_\lambda\frac{t}{\tau}}-\mathbf{I}}{4\tilde{\boldsymbol{S}}_\lambda}\right)\boldsymbol{C}^T\boldsymbol{B}^{-T}e^{-\tilde{\boldsymbol{S}}_\lambda\frac{t}{\tau}}$$

$$- e^{-\tilde{\boldsymbol{S}}_\lambda\frac{t}{\tau}}\mathbf{B}^{-1}\mathbf{W}_2(0)^T\tilde{\mathbf{U}}_\perp\boldsymbol{\lambda}_\perp^{-1}\tilde{\mathbf{U}}_\perp^T\mathbf{W}_2(0)\mathbf{B}^{-T}e^{-\tilde{\boldsymbol{S}}_\lambda\frac{t}{\tau}}$$

$$e^{-\tilde{\boldsymbol{S}}_\lambda\frac{t}{\tau}}e^{\frac{\boldsymbol{\lambda}_\perp}{2}\frac{t}{\tau}}\mathbf{B}^{-1}\mathbf{W}_2(0)^T\tilde{\mathbf{U}}_\perp\boldsymbol{\lambda}_\perp^{-1}\tilde{\mathbf{U}}_\perp^T\mathbf{W}_2(0)\mathbf{B}^{-T}e^{-\tilde{\boldsymbol{S}}_\lambda\frac{t}{\tau}}$$

$$+ e^{-\tilde{\boldsymbol{S}}_\lambda\frac{t}{\tau}}e^{\frac{\boldsymbol{\lambda}}{2}\frac{t}{\tau}}\mathbf{B}^{-1}\mathbf{W}_1(0)\tilde{\mathbf{V}}_\perp\boldsymbol{\lambda}_\perp^{-1}\tilde{\mathbf{V}}_\perp^T\mathbf{W}_1(0)^T\mathbf{B}^{-T}e^{-\tilde{\boldsymbol{S}}_\lambda\frac{t}{\tau}}$$

$$\left. - e^{-\tilde{\boldsymbol{S}}_\lambda\frac{t}{\tau}}\mathbf{B}^{-1}\mathbf{W}_1(0)\tilde{\mathbf{V}}_\perp\boldsymbol{\lambda}_\perp^{-1}\tilde{\mathbf{V}}_\perp^T\mathbf{W}_1(0)^T\mathbf{B}^{-T}e^{-\tilde{\boldsymbol{S}}_\lambda\frac{t}{\tau}}\right]^{-1}\boldsymbol{Z}^T$$

$$\boldsymbol{Z} = \begin{pmatrix}\frac{1}{2}\tilde{V}\left[(\tilde{G}-\tilde{H}\tilde{G})-(\tilde{G}+\tilde{H}\tilde{G})e^{-\tilde{\boldsymbol{S}}_\lambda\frac{t}{\tau}}\boldsymbol{C}^T\boldsymbol{B}^{-T}e^{-\tilde{\boldsymbol{S}}_\lambda\frac{t}{\tau}}\right] + \tilde{V}_\perp\tilde{V}_\perp^T\mathbf{W}_1(0)\boldsymbol{B}^{-T}e^{\boldsymbol{\lambda}_\perp\frac{t}{\tau}}e^{-\tilde{\boldsymbol{S}}_\lambda\frac{t}{\tau}}\\\frac{1}{2}\tilde{U}\left[(\tilde{G}+\tilde{H}\tilde{G})+(\tilde{G}-\tilde{H}\tilde{G})e^{-\tilde{\boldsymbol{S}}_\lambda\frac{t}{\tau}}\boldsymbol{C}^T\boldsymbol{B}^{-T}e^{-\tilde{\boldsymbol{S}}_\lambda\frac{t}{\tau}}\right] + \tilde{U}_\perp\tilde{U}_\perp^T\mathbf{W}_2(0)^T\boldsymbol{B}^{-T}e^{\boldsymbol{\lambda}_\perp\frac{t}{\tau}}e^{-\tilde{\boldsymbol{S}}_\lambda\frac{t}{\tau}}\end{pmatrix} \tag{78}$$

*Proof.* We start from

$$
\mathbf{QQ}^T(t) =
$$

$$
\left[ \begin{pmatrix} \frac{1}{2}\tilde{V}(\tilde{G}-\tilde{H}\tilde{G})e^{\tilde{S}_\lambda \frac{t}{\tau}}B^T - \frac{1}{2}\tilde{V}(\tilde{G}+\tilde{H}\tilde{G})e^{-\tilde{S}_\lambda \frac{t}{\tau}}C^T + \tilde{\mathbf{V}}_\perp e^{\boldsymbol{\lambda}_\perp \frac{t}{\tau}}\tilde{\mathbf{V}}_\perp^T \mathbf{W}_1(0)^T \\ \frac{1}{2}\tilde{U}(\tilde{G}+\tilde{H}\tilde{G})e^{\tilde{S}_\lambda \frac{t}{\tau}}B^T + \frac{1}{2}\tilde{U}(\tilde{G}-\tilde{H}\tilde{G})e^{-\tilde{S}_\lambda \frac{t}{\tau}}C^T + \tilde{\mathbf{U}}_\perp e^{\boldsymbol{\lambda}_\perp \frac{t}{\tau}}\tilde{\mathbf{U}}_\perp^T \mathbf{W}_2(0) \end{pmatrix} \right]
$$

$$
\left[ \mathbf{I} + \frac{1}{4}\left( B\left(\frac{e^{2\tilde{S}_\lambda \frac{t}{\tau}} - \mathbf{I}}{\tilde{S}_\lambda}\right)B^T - C\left(\frac{e^{-2\tilde{S}_\lambda \frac{t}{\tau}} - \mathbf{I}}{\tilde{S}_\lambda}\right)C^T \right) \right.
$$

$$
+ \mathbf{W}_2(0)^T\tilde{\mathbf{U}}_\perp \left(\frac{e^{\boldsymbol{\lambda}_\perp \frac{t}{\tau}} - \mathbf{I}}{\boldsymbol{\lambda}_\perp}\right)\tilde{\mathbf{U}}_\perp^T \mathbf{W}_2(0) + \mathbf{W}_1(0)\tilde{\mathbf{V}}_\perp \left(\frac{e^{\boldsymbol{\lambda}_\perp \frac{t}{\tau}} - \mathbf{I}}{\boldsymbol{\lambda}_\perp}\right)\tilde{\mathbf{V}}_\perp^T \mathbf{W}_1(0)^T \right]^{-1}
$$

$$
\left[ \begin{pmatrix} \frac{1}{2}\tilde{V}(\tilde{G}-\tilde{H}\tilde{G})e^{\tilde{S}_\lambda \frac{t}{\tau}}B^T - \frac{1}{2}\tilde{V}(\tilde{G}+\tilde{H}\tilde{G})e^{-\tilde{S}_\lambda \frac{t}{\tau}}C^T + \tilde{\mathbf{V}}_\perp e^{\boldsymbol{\lambda}_\perp \frac{t}{\tau}}\tilde{\mathbf{V}}_\perp^T \mathbf{W}_1(0)^T \\ \frac{1}{2}\tilde{U}(\tilde{G}+\tilde{H}\tilde{G})e^{\tilde{S}_\lambda \frac{t}{\tau}}B^T + \frac{1}{2}\tilde{U}(\tilde{G}-\tilde{H}\tilde{G})e^{-\tilde{S}_\lambda \frac{t}{\tau}}C^T + \tilde{\mathbf{U}}_\perp e^{\boldsymbol{\lambda}_\perp \frac{t}{\tau}}\tilde{\mathbf{U}}_\perp^T \mathbf{W}_2(0) \end{pmatrix} \right]^T
$$

$$\tag{79}$$

We extract $B^{-T}e^{-\tilde{S}_\lambda \frac{t}{\tau}}$ from all terms as exemplified bellow

$$
O e^{\boldsymbol{\Lambda}t/\tau}O^T Q(0) = \frac{1}{2}\left( \begin{matrix} \tilde{V}\left[(\tilde{G}-\tilde{H}\tilde{G}) - (\tilde{G}+\tilde{H}\tilde{G})e^{-\tilde{S}_\lambda \frac{t}{\tau}}C^T B^{-T}e^{-\tilde{S}_\lambda \frac{t}{\tau}}\right] \\ \tilde{U}\left[(\tilde{G}+\tilde{H}\tilde{G}) + (\tilde{G}-\tilde{H}\tilde{G})e^{-\tilde{S}_\lambda \frac{t}{\tau}}C^T B^{-T}e^{-\tilde{S}_\lambda \frac{t}{\tau}}\right] \end{matrix} \right) B^T e^{\tilde{S}_\lambda \frac{t}{\tau}} \tag{80}
$$

and rewrite the dynamis as

$$
\mathbf{QQ}^T(t) =
$$

$$
\left[ \begin{pmatrix} \frac{1}{2}\tilde{V}(\tilde{G}-\tilde{H}\tilde{G}) - \frac{1}{2}\tilde{V}(\tilde{G}+\tilde{H}\tilde{G})e^{-\tilde{S}_\lambda \frac{t}{\tau}}C^T B^{-T}e^{-\tilde{S}_\lambda \frac{t}{\tau}} + \tilde{\mathbf{V}}_\perp e^{\boldsymbol{\lambda}_\perp \frac{t}{\tau}}\tilde{\mathbf{V}}_\perp^T \mathbf{W}_1(0)^T B^{-T}e^{-\tilde{S}_\lambda \frac{t}{\tau}} \\ \frac{1}{2}\tilde{U}(\tilde{G}+\tilde{H}\tilde{G}) + \frac{1}{2}\tilde{U}(\tilde{G}-\tilde{H}\tilde{G})e^{-\tilde{S}_\lambda \frac{t}{\tau}}C^T B^{-T}e^{-\tilde{S}_\lambda \frac{t}{\tau}} + \tilde{\mathbf{U}}_\perp e^{\boldsymbol{\lambda}_\perp \frac{t}{\tau}}\tilde{\mathbf{U}}_\perp^T \mathbf{W}_2(0)B^{-T}e^{-\tilde{S}_\lambda \frac{t}{\tau}} \end{pmatrix} \right]
$$

$$
\left[ e^{-\tilde{S}_\lambda \frac{t}{\tau}}B^{-1}B^{-T}e^{-\tilde{S}_\lambda \frac{t}{\tau}} + \frac{1}{4}\left( \left(\frac{\mathbf{I} - e^{-2\tilde{S}_\lambda \frac{t}{\tau}}}{\tilde{S}_\lambda}\right) - e^{-\tilde{S}_\lambda \frac{t}{\tau}}B^{-1}C\left(\frac{e^{-2\tilde{S}_\lambda \frac{t}{\tau}} - \mathbf{I}}{\tilde{S}_\lambda}\right)C^T B^{-T}e^{-\tilde{S}_\lambda \frac{t}{\tau}} \right) \right.
$$

$$
+ e^{-\tilde{S}_\lambda \frac{t}{\tau}}B^{-1}\mathbf{W}_2(0)^T\tilde{\mathbf{U}}_\perp \left(\frac{e^{\boldsymbol{\lambda}_\perp \frac{t}{\tau}} - \mathbf{I}}{\boldsymbol{\lambda}_\perp}\right)\tilde{\mathbf{U}}_\perp^T \mathbf{W}_2(0)B^{-T}e^{-\tilde{S}_\lambda \frac{t}{\tau}}
$$

$$
\left. + e^{-\tilde{S}_\lambda \frac{t}{\tau}}B^{-1}\mathbf{W}_1(0)\tilde{\mathbf{V}}_\perp \left(\frac{e^{\boldsymbol{\lambda}_\perp \frac{t}{\tau}} - \mathbf{I}}{\boldsymbol{\lambda}_\perp}\right)\tilde{\mathbf{V}}_\perp^T \mathbf{W}_1(0)^T B^{-T}e^{-\tilde{S}_\lambda \frac{t}{\tau}} \right]^{-1}
$$

$$
\left[ \begin{pmatrix} \frac{1}{2}\tilde{V}(\tilde{G}-\tilde{H}\tilde{G}) - \frac{1}{2}\tilde{V}(\tilde{G}+\tilde{H}\tilde{G})e^{-\tilde{S}_\lambda \frac{t}{\tau}}C^T B^{-T}e^{-\tilde{S}_\lambda \frac{t}{\tau}} + \tilde{\mathbf{V}}_\perp e^{\boldsymbol{\lambda}_\perp \frac{t}{\tau}}\tilde{\mathbf{V}}_\perp^T \mathbf{W}_1(0)^T B^{-T}e^{-\tilde{S}_\lambda \frac{t}{\tau}} \\ \frac{1}{2}\tilde{U}(\tilde{G}+\tilde{H}\tilde{G}) + \frac{1}{2}\tilde{U}(\tilde{G}-\tilde{H}\tilde{G})e^{-\tilde{S}_\lambda \frac{t}{\tau}}C^T B^{-T}e^{-\tilde{S}_\lambda \frac{t}{\tau}} + \tilde{\mathbf{U}}_\perp e^{\boldsymbol{\lambda}_\perp \frac{t}{\tau}}\tilde{\mathbf{U}}_\perp^T \mathbf{W}_2(0)B^{-T}e^{-\tilde{S}_\lambda \frac{t}{\tau}} \end{pmatrix} \right]^T
$$

$$\tag{81}$$

$\mathbf{Q}\mathbf{Q}^T(t) =$

$$\begin{pmatrix} \frac{1}{2}\tilde{\boldsymbol{V}}\left[(\tilde{\boldsymbol{G}} - \tilde{\boldsymbol{H}}\tilde{\boldsymbol{G}}) - (\tilde{\boldsymbol{G}} + \tilde{\boldsymbol{H}}\tilde{\boldsymbol{G}})e^{-\tilde{\boldsymbol{S}}_\lambda\frac{t}{\tau}}\boldsymbol{C}^T\boldsymbol{B}^{-T}e^{-\tilde{\boldsymbol{S}}_\lambda\frac{t}{\tau}}\right] + \tilde{\boldsymbol{V}}_\perp\tilde{\boldsymbol{V}}_\perp^T\boldsymbol{W}_1(0)\boldsymbol{B}^{-T}e^{\lambda_\perp\frac{t}{\tau}}e^{-\tilde{\boldsymbol{S}}_\lambda\frac{t}{\tau}} \\ \frac{1}{2}\tilde{\boldsymbol{U}}\left[(\tilde{\boldsymbol{G}} + \tilde{\boldsymbol{H}}\tilde{\boldsymbol{G}}) + (\tilde{\boldsymbol{G}} - \tilde{\boldsymbol{H}}\tilde{\boldsymbol{G}})e^{-\tilde{\boldsymbol{S}}_\lambda\frac{t}{\tau}}\boldsymbol{C}^T\boldsymbol{B}^{-T}e^{-\tilde{\boldsymbol{S}}_\lambda\frac{t}{\tau}}\right] + \tilde{\boldsymbol{U}}_\perp\tilde{\boldsymbol{U}}_\perp^T\boldsymbol{W}_2(0)^T\boldsymbol{B}^{-T}e^{\lambda_\perp\frac{t}{\tau}}e^{-\tilde{\boldsymbol{S}}_\lambda\frac{t}{\tau}} \end{pmatrix}$$

$$\left[e^{-\tilde{\boldsymbol{S}}_\lambda\frac{t}{\tau}}\boldsymbol{B}^{-1}\boldsymbol{B}^{-T}e^{-\tilde{\boldsymbol{S}}_\lambda\frac{t}{\tau}}\right.$$

$$+ \left(\frac{\mathbf{I} - e^{-2\tilde{\boldsymbol{S}}_\lambda\frac{t}{\tau}}}{4\tilde{\boldsymbol{S}}_\lambda}\right) - e^{-\tilde{\boldsymbol{S}}_\lambda\frac{t}{\tau}}\boldsymbol{B}^{-1}\boldsymbol{C}\left(\frac{e^{-\tilde{\boldsymbol{S}}_\lambda\frac{t}{\tau}} - \mathbf{I}}{4\tilde{\boldsymbol{S}}_\lambda}\right)\boldsymbol{C}^T\boldsymbol{B}^{-T}e^{-\tilde{\boldsymbol{S}}_\lambda\frac{t}{\tau}}$$

$$- e^{-\tilde{\boldsymbol{S}}_\lambda\frac{t}{\tau}}\mathbf{B}^{-1}\mathbf{W}_2(0)^T\tilde{\mathbf{U}}_\perp\boldsymbol{\lambda}_\perp^{-1}\tilde{\mathbf{U}}_\perp^T\mathbf{W}_2(0)\mathbf{B}^{-T}e^{-\tilde{\boldsymbol{S}}_\lambda\frac{t}{\tau}}$$

$$e^{-\tilde{\boldsymbol{S}}_\lambda\frac{t}{\tau}}e^{\frac{\lambda_\perp}{2}\frac{t}{\tau}}\mathbf{B}^{-1}\mathbf{W}_2(0)^T\tilde{\mathbf{U}}_\perp\boldsymbol{\lambda}_\perp^{-1}\tilde{\mathbf{U}}_\perp^T\mathbf{W}_2(0)\mathbf{B}^{-T}e^{-\tilde{\boldsymbol{S}}_\lambda\frac{t}{\tau}}$$

$$+ e^{-\tilde{\boldsymbol{S}}_\lambda\frac{t}{\tau}}e^{\frac{\lambda}{2}\frac{t}{\tau}}\mathbf{B}^{-1}\mathbf{W}_1(0)\tilde{\mathbf{V}}_\perp\boldsymbol{\lambda}_\perp^{-1}\tilde{\mathbf{V}}_\perp^T\mathbf{W}_1(0)^T\mathbf{B}^{-T}e^{-\tilde{\boldsymbol{S}}_\lambda\frac{t}{\tau}}$$

$$\left. - e^{-\tilde{\boldsymbol{S}}_\lambda\frac{t}{\tau}}\mathbf{B}^{-1}\mathbf{W}_1(0)\tilde{\mathbf{V}}_\perp\boldsymbol{\lambda}_\perp^{-1}\tilde{\mathbf{V}}_\perp^T\mathbf{W}_1(0)^T\mathbf{B}^{-T}e^{-\tilde{\boldsymbol{S}}_\lambda\frac{t}{\tau}}\right]^{-1}$$

$$\begin{pmatrix} \tilde{\boldsymbol{V}}\left[(\tilde{\boldsymbol{G}} - \tilde{\boldsymbol{H}}\tilde{\boldsymbol{G}}) - (\tilde{\boldsymbol{G}} + \tilde{\boldsymbol{H}}\tilde{\boldsymbol{G}})e^{-\tilde{\boldsymbol{S}}_\lambda\frac{t}{\tau}}\boldsymbol{C}^T\boldsymbol{B}^{-T}e^{-\tilde{\boldsymbol{S}}_\lambda\frac{t}{\tau}}\right] + \tilde{\boldsymbol{V}}_\perp\tilde{\boldsymbol{V}}_\perp^T\boldsymbol{W}_1(0)\boldsymbol{B}^{-T}e^{\lambda_\perp\frac{t}{\tau}}e^{-\tilde{\boldsymbol{S}}_\lambda\frac{t}{\tau}} \\ \tilde{\boldsymbol{U}}\left[(\tilde{\boldsymbol{G}} + \tilde{\boldsymbol{H}}\tilde{\boldsymbol{G}}) + (\tilde{\boldsymbol{G}} - \tilde{\boldsymbol{H}}\tilde{\boldsymbol{G}})e^{-\tilde{\boldsymbol{S}}_\lambda\frac{t}{\tau}}\boldsymbol{C}^T\boldsymbol{B}^{-T}e^{-\tilde{\boldsymbol{S}}_\lambda\frac{t}{\tau}}\right] + \tilde{\boldsymbol{U}}_\perp\tilde{\boldsymbol{U}}_\perp^T\boldsymbol{W}_2(0)^T\boldsymbol{B}^{-T}e^{\lambda_\perp\frac{t}{\tau}}e^{-\tilde{\boldsymbol{S}}_\lambda\frac{t}{\tau}} \end{pmatrix}^T$$

$$(82)$$

where $e^{\lambda_\perp\frac{t}{\tau}} = \text{sgn}(N_o - N_i)\frac{\lambda}{2}$ is a scalar $\qquad\square$

### B.5.1 PROOF EXACT LEARNING DYNAMICS WITH PRIOR KNOWLEDGE UNEQUAL DIMENSION

We follow a similar derivation presented in Braun et al. (2022) and start with the following equation

$$\mathbf{Q}\mathbf{Q}^T(t) = \underbrace{\left[\mathbf{O}e^{\boldsymbol{\Lambda}\frac{t}{\tau}}\mathbf{O}^T + 2\mathbf{M}e^{\boldsymbol{\lambda}_\perp\frac{t}{\tau}}\mathbf{M}^T\right]\mathbf{Q}(0)}_{\mathbf{L}}$$

$$\underbrace{\left[\mathbf{I} + \frac{1}{2}\mathbf{Q}(0)^T\left(\mathbf{O}\left(e^{2\boldsymbol{\Lambda}\frac{t}{\tau}} - \mathbf{I}\right)\boldsymbol{\Lambda}^{-1}\mathbf{O}^T + \mathbf{M}(e^{\boldsymbol{\lambda}_\perp\frac{t}{\tau}} - \mathbf{I})\boldsymbol{\lambda}_\perp^{-1}\mathbf{M}^T\right)\mathbf{Q}(0)\right]^{-1}}_{\mathbf{C}^{-1}} \quad (83)$$

$$\underbrace{\mathbf{Q}(0)^T\left[\mathbf{O}e^{\boldsymbol{\Lambda}\frac{t}{\tau}}\mathbf{O}^T + 2\mathbf{M}e^{\boldsymbol{\lambda}_\perp\frac{t}{\tau}}\mathbf{M}^T\right]}_{\mathbf{R}}$$

$$= \mathbf{L}\mathbf{C}^{-1}\mathbf{R}, \quad (84)$$

Substituting our solution into the matrix Riccati equation then yields

$$\tau\frac{d}{dt}\mathbf{Q}\mathbf{Q}^T = \mathbf{F}\mathbf{Q}\mathbf{Q}^T + \mathbf{Q}\mathbf{Q}^T\mathbf{F} - (\mathbf{Q}\mathbf{Q}^T)^2 \quad (85)$$

$$\Rightarrow \tau\frac{d}{dt}\mathbf{L}\mathbf{C}^{-1}\mathbf{R} \overset{?}{=} \mathbf{F}\mathbf{L}\mathbf{C}^{-1}\mathbf{R} + \mathbf{L}\mathbf{C}^{-1}\mathbf{R}\mathbf{F} - \mathbf{L}\mathbf{C}^{-1}\mathbf{R}\mathbf{L}\mathbf{C}^{-1}\mathbf{R}. \quad (86)$$

Using the chain rule $\partial(\mathbf{AB}) = (\partial\mathbf{A})\mathbf{B} + \mathbf{A}(\partial\mathbf{B})$ and the identities

$$\frac{d}{dt}(\mathbf{A}^{-1}) = \mathbf{A}^{-1}(\frac{d}{dt}\mathbf{A})\mathbf{A}^{-1} \qquad \text{and} \qquad \frac{d}{dt}(e^{t\mathbf{A}}) = \mathbf{A}e^{t\mathbf{A}} = e^{t\mathbf{A}}\mathbf{A} \quad (87)$$

$$\tau\frac{d}{dt}\mathbf{Q}\mathbf{Q}^T = \tau\frac{d}{dt}\mathbf{L}\mathbf{C}^{-1}\mathbf{R} \quad (88)$$

$$= \tau\left(\frac{d}{dt}\mathbf{L}\right)\mathbf{C}^{-1}\mathbf{R} + \tau\mathbf{L}\left(\frac{d}{dt}C^{-1}\mathbf{R}\right) \quad (89)$$

$$= \tau\left(\frac{d}{dt}\mathbf{L}\right)\mathbf{C}^{-1}\mathbf{R} + \tau\mathbf{L}\mathbf{C}^{-1}\left(\frac{d}{dt}\mathbf{R}\right) + \tau\mathbf{L}\left(\frac{d}{dt}\mathbf{C}^{-1}\right)\mathbf{R}, \quad (90)$$

Next, we note that

$$\mathbf{O} = \frac{1}{\sqrt{2}} \begin{pmatrix} \tilde{\boldsymbol{V}}(\tilde{\boldsymbol{G}} - \tilde{\boldsymbol{H}}\tilde{\boldsymbol{G}}) & \tilde{\boldsymbol{V}}(\tilde{\boldsymbol{G}} + \tilde{\boldsymbol{H}}\tilde{\boldsymbol{G}}) \\ \tilde{\boldsymbol{U}}(\tilde{\boldsymbol{G}} + \tilde{\boldsymbol{H}}\tilde{\boldsymbol{G}}) & -\tilde{\boldsymbol{U}}(\tilde{\boldsymbol{G}} - \tilde{\boldsymbol{H}}\tilde{\boldsymbol{G}}) \end{pmatrix}^T \tag{91}$$

$$\mathbf{O}^T\mathbf{O} = \frac{1}{\sqrt{2}} \begin{pmatrix} \tilde{\boldsymbol{V}}(\tilde{\boldsymbol{G}} - \tilde{\boldsymbol{H}}\tilde{\boldsymbol{G}}) & \tilde{\boldsymbol{V}}(\tilde{\boldsymbol{G}} + \tilde{\boldsymbol{H}}\tilde{\boldsymbol{G}}) \\ \tilde{\boldsymbol{U}}(\tilde{\boldsymbol{G}} + \tilde{\boldsymbol{H}}\tilde{\boldsymbol{G}}) & -\tilde{\boldsymbol{U}}(\tilde{\boldsymbol{G}} - \tilde{\boldsymbol{H}}\tilde{\boldsymbol{G}}) \end{pmatrix}^T \frac{1}{\sqrt{2}} \begin{pmatrix} \tilde{\boldsymbol{V}}(\tilde{\boldsymbol{G}} - \tilde{\boldsymbol{H}}\tilde{\boldsymbol{G}}) & \tilde{\boldsymbol{V}}(\tilde{\boldsymbol{G}} + \tilde{\boldsymbol{H}}\tilde{\boldsymbol{G}}) \\ \tilde{\boldsymbol{U}}(\tilde{\boldsymbol{G}} + \tilde{\boldsymbol{H}}\tilde{\boldsymbol{G}}) & -\tilde{\boldsymbol{U}}(\tilde{\boldsymbol{G}} - \tilde{\boldsymbol{H}}\tilde{\boldsymbol{G}}) \end{pmatrix} \tag{92}$$

$$= \mathbf{I} \tag{93}$$

$$\mathbf{O}^T\mathbf{M} = \frac{1}{\sqrt{2}} \begin{bmatrix} \tilde{\boldsymbol{V}}(\tilde{\boldsymbol{G}} - \tilde{\boldsymbol{H}}\tilde{\boldsymbol{G}}) & \tilde{\boldsymbol{V}}(\tilde{\boldsymbol{G}} + \tilde{\boldsymbol{H}}\tilde{\boldsymbol{G}}) \\ \tilde{\boldsymbol{U}}(\tilde{\boldsymbol{G}} + \tilde{\boldsymbol{H}}\tilde{\boldsymbol{G}}) & -\tilde{\boldsymbol{U}}(\tilde{\boldsymbol{G}} - \tilde{\boldsymbol{H}}\tilde{\boldsymbol{G}}) \end{bmatrix} \frac{1}{\sqrt{2}} \begin{bmatrix} \tilde{\mathbf{V}}_\perp \\ \tilde{\mathbf{U}}_\perp \end{bmatrix} \tag{94}$$

$$= \frac{1}{2} \begin{bmatrix} (\tilde{\boldsymbol{G}} - \tilde{\boldsymbol{H}}\tilde{\boldsymbol{G}})^T\tilde{\mathbf{V}}^T\tilde{\mathbf{V}}_\perp + (\tilde{\boldsymbol{G}} + \tilde{\boldsymbol{H}}\tilde{\boldsymbol{G}})^T\tilde{\mathbf{U}}^T\tilde{\mathbf{U}}_\perp \\ (\tilde{\boldsymbol{G}} + \tilde{\boldsymbol{H}}\tilde{\boldsymbol{G}})^T\tilde{\mathbf{V}}^T\tilde{\mathbf{V}}_\perp - (\tilde{\boldsymbol{G}} - \tilde{\boldsymbol{H}}\tilde{\boldsymbol{G}})^T\tilde{\mathbf{U}}^T\tilde{\mathbf{U}}_\perp \end{bmatrix} \tag{95}$$

$$= \mathbf{0} \tag{96}$$

and

$$\mathbf{M}^T\mathbf{O} = \frac{1}{\sqrt{2}} \begin{bmatrix} \tilde{\mathbf{V}}_\perp^T & \tilde{\mathbf{U}}_\perp^T \end{bmatrix} \begin{pmatrix} \tilde{\boldsymbol{V}}(\tilde{\boldsymbol{G}} - \tilde{\boldsymbol{H}}\tilde{\boldsymbol{G}}) & \tilde{\boldsymbol{V}}(\tilde{\boldsymbol{G}} + \tilde{\boldsymbol{H}}\tilde{\boldsymbol{G}}) \\ \tilde{\boldsymbol{U}}(\tilde{\boldsymbol{G}} + \tilde{\boldsymbol{H}}\tilde{\boldsymbol{G}}) & -\tilde{\boldsymbol{U}}(\tilde{\boldsymbol{G}} - \tilde{\boldsymbol{H}}\tilde{\boldsymbol{G}}) \end{pmatrix} \tag{97}$$

$$= \frac{1}{2} \begin{bmatrix} \tilde{\mathbf{V}}_\perp^T\tilde{\mathbf{V}}(\tilde{\boldsymbol{G}} - \tilde{\boldsymbol{H}}\tilde{\boldsymbol{G}}) + \tilde{\mathbf{U}}_\perp^T\tilde{\mathbf{U}}(\tilde{\boldsymbol{G}} + \tilde{\boldsymbol{H}}\tilde{\boldsymbol{G}}) \\ \tilde{\mathbf{V}}_\perp^T\tilde{\mathbf{V}}(\tilde{\boldsymbol{G}} + \tilde{\boldsymbol{H}}\tilde{\boldsymbol{G}}) - \tilde{\mathbf{U}}_\perp^T\tilde{\mathbf{U}}(\tilde{\boldsymbol{G}} - \tilde{\boldsymbol{H}}\tilde{\boldsymbol{G}}) \end{bmatrix} \tag{98}$$

$$= \mathbf{0}. \tag{99}$$

we get

$$\tau\frac{d}{dt}\mathbf{Q}\mathbf{Q}^T = \tau\frac{d}{dt}\left(\mathbf{L}\mathbf{C}^{-1}\mathbf{R}\right) \tag{100}$$

$$= \tau\left(\frac{d}{dt}\mathbf{L}\right)\mathbf{C}^{-1}\mathbf{R} + \tau\mathbf{L}\left(\frac{d}{dt}C^{-1}\mathbf{R}\right) \tag{101}$$

$$= \tau\left(\frac{d}{dt}\mathbf{L}\right)\mathbf{C}^{-1}\mathbf{R} + \tau\mathbf{L}\mathbf{C}^{-1}\left(\frac{d}{dt}\mathbf{R}\right) + \tau\mathbf{L}\left(\frac{d}{dt}\mathbf{C}^{-1}\right)\mathbf{R}, \tag{102}$$

with

$$\tau\left(\frac{d}{dt}\mathbf{L}\right)\mathbf{C}^{-1}\mathbf{R} = \tau\left(\mathbf{O}\frac{1}{\tau}\mathbf{\Lambda}e^{\mathbf{\Lambda}\frac{t}{\tau}}\mathbf{O}^T + 2\mathbf{M}\frac{\lambda_\perp\mathbf{I}}{2\tau}e^{\lambda_\perp\frac{t}{\tau}}\mathbf{M}^T\right)\mathbf{Q}(0)\mathbf{C}^{-1}\mathbf{R} \tag{103}$$

$$= \left(\mathbf{O}\mathbf{\Lambda}e^{\mathbf{\Lambda}\frac{t}{\tau}}\mathbf{O}^T + \mathbf{M}\lambda_\perp\mathbf{I}e^{\lambda_\perp\frac{t}{\tau}}\mathbf{M}^T\right)\mathbf{Q}(0)\mathbf{C}^{-1}\mathbf{R} \tag{104}$$

$$= (\mathbf{O}\lambda_\perp\mathbf{O}^T + 2\mathbf{M}\lambda_\perp\mathbf{M}^T)\left(\mathbf{O}e^{\mathbf{\Lambda}\frac{t}{\tau}}\mathbf{O}^T + 2\mathbf{M}e^{\lambda_\perp\frac{t}{\tau}}\mathbf{M}^T\right)\mathbf{Q}(0)\mathbf{C}^{-1}\mathbf{R} \tag{105}$$

$$= \mathbf{F}\mathbf{L}\mathbf{C}^{-1}\mathbf{R}, \tag{106}$$

$$\tau\mathbf{L}\mathbf{C}^{-1}\left(\frac{d}{dt}\mathbf{R}\right) = \tau\mathbf{L}\mathbf{C}^{-1}\mathbf{Q}(0)^T\left(\mathbf{O}\frac{1}{\tau}e^{\mathbf{\Lambda}\frac{t}{\tau}}\mathbf{\Lambda}\mathbf{O}^T + 2\mathbf{M}e^{\lambda_\perp\frac{t}{\tau}}\frac{\lambda_\perp\mathbf{I}}{2\tau}\mathbf{M}^T\right) \tag{107}$$

$$= \mathbf{L}\mathbf{C}^{-1}\mathbf{Q}(0)^T\left(\mathbf{O}\frac{1}{\tau}e^{\mathbf{\Lambda}\frac{t}{\tau}}\mathbf{\Lambda}\mathbf{O}^T + 2\mathbf{M}e^{\lambda_\perp\frac{t}{\tau}}\frac{\lambda_\perp\mathbf{I}}{2\tau}\mathbf{M}^T\right) \tag{108}$$

$$= \mathbf{L}\mathbf{C}^{-1}\mathbf{R}\mathbf{F} \tag{109}$$

and

$$\tau \mathbf{L} \left( \frac{d}{dt} \mathbf{C}^{-1} \right) \mathbf{R} = -\tau \mathbf{L} \mathbf{C}^{-1} \left( \frac{d}{dt} \mathbf{C} \right) \mathbf{C}^{-1} \mathbf{R} \tag{110}$$

$$= -\mathbf{L} \mathbf{C}^{-1} \left[ \tau \frac{1}{2} \mathbf{Q}(0)^T \mathbf{O} 2 \frac{1}{\tau} e^{2\mathbf{\Lambda} \frac{t}{\tau}} \mathbf{\Lambda} \mathbf{\Lambda}^{-1} \mathbf{O}^T \mathbf{Q}(0) \tag{111} \right.$$

$$\left. + \tau \frac{1}{2} \mathbf{Q}(0)^T 4 \frac{1}{\tau} \mathbf{M} e^{\mathbf{\lambda}_\perp \frac{t}{\tau}} \mathbf{\lambda}_\perp \left( \mathbf{\lambda}_\perp \right)^{-1} \mathbf{M}^T \mathbf{Q}(0) \right] \mathbf{C}^{-1} \mathbf{R}$$

$$= -\mathbf{L} \mathbf{C}^{-1} \left[ \mathbf{Q}(0)^T \mathbf{O} e^{2\mathbf{\Lambda} \frac{t}{\tau}} \mathbf{O}^T \mathbf{Q}(0) + 2 \mathbf{Q}(0)^T \mathbf{M} e^{\mathbf{\lambda}_\perp \frac{t}{\tau}} \mathbf{M}^T \mathbf{Q}(0) \right] \mathbf{C}^{-1} \mathbf{R}$$
$$\tag{112}$$

$$= -\mathbf{L} \mathbf{C}^{-1} \left[ \mathbf{Q}(0)^T \mathbf{O} e^{\mathbf{\Lambda} \frac{t}{\tau}} \mathbf{O}^T \mathbf{O} e^{\mathbf{\Lambda} \frac{t}{\tau}} \mathbf{O}^T \mathbf{Q}(0) \right.$$

$$+ 2 \mathbf{Q}(0)^T \mathbf{O} e^{\mathbf{\Lambda} \frac{t}{\tau}} \underbrace{\mathbf{O}^T \mathbf{M}}_{\mathbf{0}} e^{\mathbf{\lambda}_\perp \frac{t}{\tau}} \mathbf{M}^T \mathbf{Q}(0) \tag{113}$$

$$+ 2 \mathbf{Q}(0)^T \mathbf{M} e^{\mathbf{\lambda}_\perp \frac{t}{\tau}} \underbrace{\mathbf{M}^T \mathbf{O}}_{\mathbf{0}} e^{\mathbf{\Lambda} \frac{t}{\tau}} \mathbf{O}^T \mathbf{Q}(0)$$

$$\left. + 4 \mathbf{Q}(0)^T \mathbf{M} e^{\mathbf{\lambda}_\perp \frac{t}{\tau}} \mathbf{M}^T \mathbf{M} e^{\mathbf{\lambda}_\perp \frac{t}{\tau}} \mathbf{M}^T \mathbf{Q}(0) \right] \mathbf{C}^{-1} \mathbf{R}$$

$$= -\mathbf{L} \mathbf{C}^{-1} \mathbf{R} \mathbf{L} \mathbf{C}^{-1} \mathbf{R}. \tag{114}$$

Finally, substituting equations 103, 107 and 110 into the left hand side of equation 86 proves equality. $\square$

## C  RICH-LAZY

### C.1  DYNAMICS OF THE SINGULAR VALUES

**Theorem C.1.** *Under the assumptions of Theorem 4.3 and with a task-aligned initialization given by $\boldsymbol{W}_1(0) = \boldsymbol{R} \boldsymbol{S}_1 \tilde{\boldsymbol{V}}^T$ and $\boldsymbol{W}_2(0) = \tilde{\boldsymbol{U}} \boldsymbol{S}_2 \boldsymbol{R}^T$, where $\boldsymbol{R} \in \mathbb{R}^{N_h \times N_h}$ is an orthonormal matrix, then the network function is given by the expression $\boldsymbol{W}_2 \boldsymbol{W}_1(t) = \tilde{\boldsymbol{U}} \boldsymbol{S}(t) \tilde{\boldsymbol{V}}^T$ where $\boldsymbol{S}(t) \in \mathbb{R}^{N_h \times N_h}$ is a diagonal matrix of singular values with elements $s_\alpha(t)$ that evolve according to the equation,*

$$s_\alpha(t) = s_\alpha(0) + \gamma_\alpha(t; \lambda) \left( \tilde{s}_\alpha - s_\alpha(0) \right), \tag{115}$$

*where $\tilde{s}_\alpha$ is the $\alpha$ singular value of $\tilde{\boldsymbol{S}}$ and $\gamma_\alpha(t; \lambda)$ is a $\lambda$-dependent monotonic transition function for each singular value that increases from $\gamma_\alpha(0; \lambda) = 0$ to $\lim_{t \to \infty} \gamma_\alpha(t; \lambda) = 1$ defined as*

$$\gamma_\alpha(t; \lambda) = \frac{\tilde{s}_{\lambda,\alpha} s_{\lambda,\alpha} \sinh\left( 2\tilde{s}_{\lambda,\alpha} \frac{t}{\tau} \right) + \left( \tilde{s}_\alpha s_\alpha + \frac{\lambda^2}{4} \right) \cosh\left( 2\tilde{s}_{\lambda,\alpha} \frac{t}{\tau} \right) - \left( \tilde{s}_\alpha s_\alpha + \frac{\lambda^2}{4} \right)}{\tilde{s}_{\lambda,\alpha} s_{\lambda,\alpha} \sinh\left( 2\tilde{s}_{\lambda,\alpha} \frac{t}{\tau} \right) + \left( \tilde{s}_\alpha s_\alpha + \frac{\lambda^2}{4} \right) \cosh\left( 2\tilde{s}_{\lambda,\alpha} \frac{t}{\tau} \right) + \tilde{s}_\alpha \left( \tilde{s}_\alpha - s_\alpha \right)}, \tag{116}$$

*where $\tilde{s}_{\lambda,\alpha} = \sqrt{\tilde{s}_\alpha^2 + \frac{\lambda^2}{4}}$, $s_{\lambda,\alpha} = \sqrt{s_\alpha(0)^2 + \frac{\lambda^2}{4}}$, and $s_\alpha = s_\alpha(0)$. We find that under different limits of $\lambda$, the transition function converges pointwise to the sigmoidal ($\lambda \to 0$) and exponential ($\lambda \to \pm\infty$) transition functions,*

$$\gamma_\alpha(t; \lambda) \to \begin{cases} \frac{e^{2\tilde{s}_\alpha \frac{t}{\tau}} - 1}{e^{2\tilde{s}_\alpha \frac{t}{\tau}} - 1 + \frac{\tilde{s}_\alpha}{s_\alpha(0)}} & \text{as } \lambda \to 0, \\ 1 - e^{-|\lambda| \frac{t}{\tau}} & \text{as } \lambda \to \pm\infty \end{cases}. \tag{117}$$

*Proof.* According to Theorem 4.3, the network function is given by the equation

$$\boldsymbol{W}_2 \boldsymbol{W}_1(t) = \boldsymbol{Z}_2(t) \boldsymbol{A}^{-1}(t) \boldsymbol{Z}_1^T(t), \tag{118}$$

which depends on the variables of the initialization $\mathbf{B}$ and $\mathbf{C}$. Plugging the expressions for a task-aligned initialization $\boldsymbol{W}_1(0)$ and $\boldsymbol{W}_2(0)$ into these variables we get the following simplified expres-

sions,

$$\mathbf{B} = \boldsymbol{R} \underbrace{\left( \boldsymbol{S}_2(\tilde{\boldsymbol{G}} + \tilde{\boldsymbol{H}}\tilde{\boldsymbol{G}}) + \boldsymbol{S}_1(\tilde{\boldsymbol{G}} - \tilde{\boldsymbol{H}}\tilde{\boldsymbol{G}}) \right)}_{\boldsymbol{D}_B}, \tag{119}$$

$$\mathbf{C} = \boldsymbol{R} \underbrace{\left( \boldsymbol{S}_2(\tilde{\boldsymbol{G}} - \tilde{\boldsymbol{H}}\tilde{\boldsymbol{G}}) - \boldsymbol{S}_1(\tilde{\boldsymbol{G}} + \tilde{\boldsymbol{H}}\tilde{\boldsymbol{G}}) \right)}_{\boldsymbol{D}_C}, \tag{120}$$

where we define the diagonal matrices $\boldsymbol{D}_B$ and $\boldsymbol{D}_C$ for ease of notation. Using these expressions, we now get the following time-dependent expressions for $\boldsymbol{Z}_2(t)$, $\boldsymbol{A}^{-1}(t)$, and $\boldsymbol{Z}_1(t)$,

$$\boldsymbol{Z}_1(t) = \frac{1}{2}\tilde{\boldsymbol{V}} \left( (\tilde{\boldsymbol{G}} - \tilde{\boldsymbol{H}}\tilde{\boldsymbol{G}})e^{\tilde{\boldsymbol{S}}_\lambda \frac{t}{\tau}}\boldsymbol{D}_B - (\tilde{\boldsymbol{G}} + \tilde{\boldsymbol{H}}\tilde{\boldsymbol{G}})e^{-\tilde{\boldsymbol{S}}_\lambda \frac{t}{\tau}}\boldsymbol{D}_C \right) \boldsymbol{R}^T \tag{121}$$

$$\boldsymbol{Z}_2(t) = \frac{1}{2}\tilde{\boldsymbol{U}} \left( (\tilde{\boldsymbol{G}} + \tilde{\boldsymbol{H}}\tilde{\boldsymbol{G}})e^{\tilde{\boldsymbol{S}}_\lambda \frac{t}{\tau}}\boldsymbol{D}_B + (\tilde{\boldsymbol{G}} - \tilde{\boldsymbol{H}}\tilde{\boldsymbol{G}})e^{-\tilde{\boldsymbol{S}}_\lambda \frac{t}{\tau}}\boldsymbol{D}_C \right) \boldsymbol{R}^T \tag{122}$$

$$\boldsymbol{A}(t) = \boldsymbol{R} \left( \mathbf{I} + \left( \frac{e^{2\tilde{\boldsymbol{S}}_\lambda \frac{t}{\tau}} - \mathbf{I}}{4\tilde{\boldsymbol{S}}_\lambda} \right) \boldsymbol{D}_B^2 - \left( \frac{e^{-2\tilde{\boldsymbol{S}}_\lambda \frac{t}{\tau}} - \mathbf{I}}{4\tilde{\boldsymbol{S}}_\lambda} \right) \boldsymbol{D}_C^2 \right) \boldsymbol{R}^T \tag{123}$$

Plugging these expressions into the expression for the network function, notice that the $\boldsymbol{R}$ terms cancel each other resulting in following equation

$$\boldsymbol{W}_2\boldsymbol{W}_1(t) = \tilde{\boldsymbol{U}} \underbrace{\left( \frac{\left( (\tilde{\boldsymbol{G}} - \tilde{\boldsymbol{H}}\tilde{\boldsymbol{G}})e^{\tilde{\boldsymbol{S}}_\lambda \frac{t}{\tau}}\boldsymbol{D}_B - (\tilde{\boldsymbol{G}} + \tilde{\boldsymbol{H}}\tilde{\boldsymbol{G}})e^{-\tilde{\boldsymbol{S}}_\lambda \frac{t}{\tau}}\boldsymbol{D}_C \right)\left( (\tilde{\boldsymbol{G}} + \tilde{\boldsymbol{H}}\tilde{\boldsymbol{G}})e^{\tilde{\boldsymbol{S}}_\lambda \frac{t}{\tau}}\boldsymbol{D}_B + (\tilde{\boldsymbol{G}} - \tilde{\boldsymbol{H}}\tilde{\boldsymbol{G}})e^{-\tilde{\boldsymbol{S}}_\lambda \frac{t}{\tau}}\boldsymbol{D}_C \right)}{4\mathbf{I} + \left( \frac{e^{2\tilde{\boldsymbol{S}}_\lambda \frac{t}{\tau}} - \mathbf{I}}{\tilde{\boldsymbol{S}}_\lambda} \right) \boldsymbol{D}_B^2 - \left( \frac{e^{-2\tilde{\boldsymbol{S}}_\lambda \frac{t}{\tau}} - \mathbf{I}}{\tilde{\boldsymbol{S}}_\lambda} \right) \boldsymbol{D}_C^2} \right)}_{\boldsymbol{S}(t)} \tilde{\boldsymbol{V}}^T, \tag{124}$$

Notice that the middle term is simply a product of diagonal matrices. We can factor the numerator of this expressions as,

$$(\tilde{\boldsymbol{G}}^2 - \tilde{\boldsymbol{H}}^2\tilde{\boldsymbol{G}}^2)e^{2\tilde{\boldsymbol{S}}_\lambda \frac{t}{\tau}}\boldsymbol{D}_B^2 + \left( (\tilde{\boldsymbol{G}} - \tilde{\boldsymbol{H}}\tilde{\boldsymbol{G}})^2 - (\tilde{\boldsymbol{G}} + \tilde{\boldsymbol{H}}\tilde{\boldsymbol{G}})^2 \right) \boldsymbol{D}_B\boldsymbol{D}_C - (\tilde{\boldsymbol{G}}^2 - \tilde{\boldsymbol{H}}^2\tilde{\boldsymbol{G}}^2)e^{-2\tilde{\boldsymbol{S}}_\lambda \frac{t}{\tau}}\boldsymbol{D}_C^2 \tag{125}$$

We can further factor this expression as,

$$\tilde{\boldsymbol{G}}^2(\mathbf{I} - \tilde{\boldsymbol{H}}^2) \left( e^{2\tilde{\boldsymbol{S}}_\lambda \frac{t}{\tau}}\boldsymbol{D}_B^2 - e^{-2\tilde{\boldsymbol{S}}_\lambda \frac{t}{\tau}}\boldsymbol{D}_C^2 \right) - 4\tilde{\boldsymbol{G}}^2\tilde{\boldsymbol{H}}\boldsymbol{D}_B\boldsymbol{D}_C. \tag{126}$$

Putting it all together we find that $\boldsymbol{S}(t)$ can be expressed as,

$$\boldsymbol{S}(t) = \frac{\tilde{\boldsymbol{G}}^2(\mathbf{I} - \tilde{\boldsymbol{H}}^2) \left( e^{2\tilde{\boldsymbol{S}}_\lambda \frac{t}{\tau}}\boldsymbol{D}_B^2 - e^{-2\tilde{\boldsymbol{S}}_\lambda \frac{t}{\tau}}\boldsymbol{D}_C^2 \right) - 4\tilde{\boldsymbol{G}}^2\tilde{\boldsymbol{H}}\boldsymbol{D}_B\boldsymbol{D}_C}{4\mathbf{I} + \left( \frac{e^{2\tilde{\boldsymbol{S}}_\lambda \frac{t}{\tau}} - \mathbf{I}}{\tilde{\boldsymbol{S}}_\lambda} \right) \boldsymbol{D}_B^2 - \left( \frac{e^{-2\tilde{\boldsymbol{S}}_\lambda \frac{t}{\tau}} - \mathbf{I}}{\tilde{\boldsymbol{S}}_\lambda} \right) \boldsymbol{D}_C^2}. \tag{127}$$

Now using the relationship between $\tilde{\boldsymbol{H}}$ and $\tilde{\boldsymbol{G}}$ we use the following two identities:

$$\tilde{\boldsymbol{G}}^2(\mathbf{I} - \tilde{\boldsymbol{H}}^2) = \frac{\tilde{\boldsymbol{S}}}{\tilde{\boldsymbol{S}}_\lambda}, \qquad 4\tilde{\boldsymbol{G}}^2\tilde{\boldsymbol{H}} = \frac{\lambda}{\tilde{\boldsymbol{S}}_\lambda} \tag{128}$$

Plugging these identities into the previous expression and multiplying the numerator and denominator by $\tilde{\boldsymbol{S}}_\lambda$ gives,

$$\boldsymbol{S}(t) = \frac{\tilde{\boldsymbol{S}} \left( e^{2\tilde{\boldsymbol{S}}_\lambda \frac{t}{\tau}}\boldsymbol{D}_B^2 - e^{-2\tilde{\boldsymbol{S}}_\lambda \frac{t}{\tau}}\boldsymbol{D}_C^2 \right) - \lambda\boldsymbol{D}_B\boldsymbol{D}_C}{4\tilde{\boldsymbol{S}}_\lambda + e^{2\tilde{\boldsymbol{S}}_\lambda \frac{t}{\tau}}\boldsymbol{D}_B^2 - e^{-2\tilde{\boldsymbol{S}}_\lambda \frac{t}{\tau}}\boldsymbol{D}_C^2 + \boldsymbol{D}_C^2 - \boldsymbol{D}_B^2}. \tag{129}$$

Add and subtract $\tilde{\boldsymbol{S}} \left( 4\tilde{\boldsymbol{S}}_\lambda + \boldsymbol{D}_C^2 - \boldsymbol{D}_B^2 \right)$ from the numerator such that

$$\boldsymbol{S}(t) = \tilde{\boldsymbol{S}} - \frac{\tilde{\boldsymbol{S}} \left( 4\tilde{\boldsymbol{S}}_\lambda + \boldsymbol{D}_C^2 - \boldsymbol{D}_B^2 \right) + \lambda\boldsymbol{D}_B\boldsymbol{D}_C}{4\tilde{\boldsymbol{S}}_\lambda + e^{2\tilde{\boldsymbol{S}}_\lambda \frac{t}{\tau}}\boldsymbol{D}_B^2 - e^{-2\tilde{\boldsymbol{S}}_\lambda \frac{t}{\tau}}\boldsymbol{D}_C^2 + \boldsymbol{D}_C^2 - \boldsymbol{D}_B^2}. \tag{130}$$

Using the form of $\boldsymbol{D}_B$ and $\boldsymbol{D}_C$ notice the following two identities:

$$\boldsymbol{D}_B\boldsymbol{D}_C = \frac{\lambda}{\tilde{\boldsymbol{S}}_\lambda}\left(\tilde{\boldsymbol{S}} - \boldsymbol{S}_2\boldsymbol{S}_1\right), \qquad \boldsymbol{D}_C^2 - \boldsymbol{D}_B^2 = -\frac{4}{\tilde{\boldsymbol{S}}_\lambda}\left(\tilde{\boldsymbol{S}}\boldsymbol{S}_2\boldsymbol{S}_1 + \frac{\lambda^2}{4}\mathbf{I}\right) \tag{131}$$

From the second identity we can derive a third identity,

$$4\tilde{\boldsymbol{S}}_\lambda + \boldsymbol{D}_C^2 - \boldsymbol{D}_B^2 = 4\frac{\tilde{\boldsymbol{S}}}{\tilde{\boldsymbol{S}}_\lambda}\left(\tilde{\boldsymbol{S}} - \boldsymbol{S}_2\boldsymbol{S}_1\right) \tag{132}$$

Plugging the first and third identities into the numerator for the previous expression gives,

$$\boldsymbol{S}(t) = \tilde{\boldsymbol{S}} - \frac{\frac{\left(4\tilde{\boldsymbol{S}}^2 + \lambda^2\mathbf{I}\right)}{\tilde{\boldsymbol{S}}_\lambda}\left(\tilde{\boldsymbol{S}} - \boldsymbol{S}_2\boldsymbol{S}_1\right)}{4\tilde{\boldsymbol{S}}_\lambda + e^{2\tilde{\boldsymbol{S}}_\lambda\frac{t}{\tau}}\boldsymbol{D}_B^2 - e^{-2\tilde{\boldsymbol{S}}_\lambda\frac{t}{\tau}}\boldsymbol{D}_C^2 + \boldsymbol{D}_C^2 - \boldsymbol{D}_B^2}. \tag{133}$$

Multiply numerator and denominator by $\frac{\tilde{\boldsymbol{S}}_\lambda}{4}$ and simplify terms gives the expression,

$$\boldsymbol{S}(t) = \tilde{\boldsymbol{S}} - \frac{\tilde{\boldsymbol{S}}_\lambda^2}{\tilde{\boldsymbol{S}}_\lambda^2 + \frac{\tilde{\boldsymbol{S}}_\lambda}{4}\left(e^{2\tilde{\boldsymbol{S}}_\lambda\frac{t}{\tau}}\boldsymbol{D}_B^2 - e^{-2\tilde{\boldsymbol{S}}_\lambda\frac{t}{\tau}}\boldsymbol{D}_C^2\right) - \frac{\tilde{\boldsymbol{S}}_\lambda}{4}\left(\boldsymbol{D}_B^2 - \boldsymbol{D}_C^2\right)}\left(\tilde{\boldsymbol{S}} - \boldsymbol{S}_2\boldsymbol{S}_1\right). \tag{134}$$

Thus we have found the transition function,

$$\gamma(t;\lambda) = \frac{\frac{\tilde{\boldsymbol{S}}_\lambda}{4}\left(e^{2\tilde{\boldsymbol{S}}_\lambda\frac{t}{\tau}}\boldsymbol{D}_B^2 - e^{-2\tilde{\boldsymbol{S}}_\lambda\frac{t}{\tau}}\boldsymbol{D}_C^2\right) + \frac{\tilde{\boldsymbol{S}}_\lambda}{4}\left(\boldsymbol{D}_C^2 - \boldsymbol{D}_B^2\right)}{\frac{\tilde{\boldsymbol{S}}_\lambda}{4}\left(e^{2\tilde{\boldsymbol{S}}_\lambda\frac{t}{\tau}}\boldsymbol{D}_B^2 - e^{-2\tilde{\boldsymbol{S}}_\lambda\frac{t}{\tau}}\boldsymbol{D}_C^2\right) + \frac{\tilde{\boldsymbol{S}}_\lambda}{4}\left(4\tilde{\boldsymbol{S}}_\lambda + \boldsymbol{D}_C^2 - \boldsymbol{D}_B^2\right)}. \tag{135}$$

We will use our previous identities and the definitions of $\boldsymbol{D}_B^2$ and $\boldsymbol{D}_C^2$ to simplify this expression. Notice the following identity,

$$\frac{\tilde{\boldsymbol{S}}_\lambda}{4}\left(e^{2\tilde{\boldsymbol{S}}_\lambda\frac{t}{\tau}}\boldsymbol{D}_B^2 - e^{-2\tilde{\boldsymbol{S}}_\lambda\frac{t}{\tau}}\boldsymbol{D}_C^2\right) = \tilde{\boldsymbol{S}}_\lambda\boldsymbol{S}_\lambda\sinh\left(2\tilde{\boldsymbol{S}}_\lambda\frac{t}{\tau}\right) + \left(\tilde{\boldsymbol{S}}\boldsymbol{S}(0) + \frac{\lambda^2}{4}\mathbf{I}\right)\cosh\left(2\tilde{\boldsymbol{S}}_\lambda\frac{t}{\tau}\right) \tag{136}$$

Putting it all together we get

$$\gamma(t;\lambda) = \frac{\tilde{\boldsymbol{S}}_\lambda\boldsymbol{S}_\lambda\sinh\left(2\tilde{\boldsymbol{S}}_\lambda\frac{t}{\tau}\right) + \left(\tilde{\boldsymbol{S}}\boldsymbol{S}(0) + \frac{\lambda^2}{4}\mathbf{I}\right)\cosh\left(2\tilde{\boldsymbol{S}}_\lambda\frac{t}{\tau}\right) - \left(\tilde{\boldsymbol{S}}\boldsymbol{S}(0) + \frac{\lambda^2}{4}\mathbf{I}\right)}{\tilde{\boldsymbol{S}}_\lambda\boldsymbol{S}_\lambda\sinh\left(2\tilde{\boldsymbol{S}}_\lambda\frac{t}{\tau}\right) + \left(\tilde{\boldsymbol{S}}\boldsymbol{S}(0) + \frac{\lambda^2}{4}\mathbf{I}\right)\cosh\left(2\tilde{\boldsymbol{S}}_\lambda\frac{t}{\tau}\right) + \tilde{\boldsymbol{S}}\left(\tilde{\boldsymbol{S}} - \boldsymbol{S}(0)\right)} \tag{137}$$

We will now show why under certain limits of $\lambda$ this expression simplifies to the sigmoidal and exponential dynamics discussed in the previous section.

**Sigmoidal dynamics.** When $\lambda = 0$, then $\tilde{\boldsymbol{S}}_\lambda = \tilde{\boldsymbol{S}}$ and $\boldsymbol{S}_\lambda = \boldsymbol{S}(0)$. Notice, that the coefficients for the hyperbolic functions all simplify to $\tilde{\boldsymbol{S}}\boldsymbol{S}(0)$. Using the hyperbolic identity $\sinh(x) + \cosh(x) = e^x$, we can simplify the expression for the transition function to

$$\gamma(t;\lambda) = \frac{\tilde{\boldsymbol{S}}\boldsymbol{S}(0)e^{2\tilde{\boldsymbol{S}}\frac{t}{\tau}} - \tilde{\boldsymbol{S}}\boldsymbol{S}(0)}{\tilde{\boldsymbol{S}}\boldsymbol{S}(0)e^{2\tilde{\boldsymbol{S}}\frac{t}{\tau}} - \tilde{\boldsymbol{S}}\boldsymbol{S}(0) + \tilde{\boldsymbol{S}}^2}. \tag{138}$$

Dividing the numerator and denominator by $\tilde{\boldsymbol{S}}\boldsymbol{S}(0)$ gives the final expression.

**Exponential dynamics.** In the limit as $\lambda \to \pm\infty$ the expressions $\tilde{\boldsymbol{S}}_\lambda \to \frac{|\lambda|}{2}$ and $\boldsymbol{S}_\lambda \to \frac{|\lambda|}{2}$. Additionally, in these limits because $\frac{\lambda^2}{4}\mathbf{I} \gg \tilde{\boldsymbol{S}}\boldsymbol{S}(0)$ then $\left(\tilde{\boldsymbol{S}}\boldsymbol{S}(0) + \frac{\lambda^2}{4}\mathbf{I}\right) \to \frac{\lambda^2}{4}\mathbf{I}$. As a result of these simplifications the coefficients for the hyperbolic functions all simplify to $\frac{\lambda^2}{4}\mathbf{I}$. As a result we can again use the hyperbolic identity $\sinh(x) + \cosh(x) = e^x$ to simplify the expression as

$$\gamma(t;\lambda) = \frac{\frac{\lambda^2}{4}e^{|\lambda|\frac{t}{\tau}} - \frac{\lambda^2}{4}\mathbf{I}}{\frac{\lambda^2}{4}e^{|\lambda|\frac{t}{\tau}} + \tilde{\boldsymbol{S}}\left(\tilde{\boldsymbol{S}} - \boldsymbol{S}(0)\right)}. \tag{139}$$

Dividing the numerator and denominator by $\frac{\lambda^2}{4}$ results in all terms without a coefficient proportional to $\lambda^2$ vanishing, which simplifying further gives the final expression. $\qquad\square$

## C.2 DYNAMICS OF THE REPRESENTATION FROM THE LAZY TO THE RICH REGIME

The *lazy* and *rich* regimes are defined by the dynamics of the NTK of the network. *Lazy* learning occurs when the NTK is constant, *rich* learning occurs when it is not. (Farrell et al. (2023))
The NTK intuitively measures the movement of the network representations through training. As shown in (Braun et al. (2022)), in specific experimental setup, we can calculate the NTK of the network in terms of the internal representations in a straightforward way:

$$\text{NTK} = \mathbf{I}_{N_o} \otimes \mathbf{X}^T \mathbf{W}_1^T \mathbf{W}_1(t) \mathbf{X} + \mathbf{W}_2 \mathbf{W}_2^T(t) \otimes \mathbf{X}^T \mathbf{X} \tag{140}$$

In order to better understand the effect of $\lambda$ on NTK dynamics, we first prove some theorems involving the Singular Values of the $\lambda$-*balanced* weights, and the representations of a $\lambda$-*balanced* network.

### C.2.1 LAMBDA-BALANCED SINGULAR VALUE

**Theorem C.2.** *Under a $\lambda$-Balanced initialization 2, if the network function $\mathbf{W}_2 \mathbf{W}_1(t) = \mathbf{U}(t)\mathbf{S}(t)\mathbf{V}^T(t)$ is full rank and we define $\mathbf{S}_{\boldsymbol{\lambda}}(t) = \sqrt{\mathbf{S}^2(t) + \frac{\lambda^2}{4}\mathbf{I}}$ , then we can recover the parameters $\mathbf{W}_2(t) = \mathbf{U}(t)\mathbf{S}_2(t)\mathbf{R}^T(t)$, $\mathbf{W}_1(t) = \mathbf{R}(t)\mathbf{S}_1(t)\mathbf{V}^T(t)$ up to time-dependent orthogonal transformation $\mathbf{R}(t)$ of size $N_h \times N_h$, where*

$$\mathbf{S}_1(t) = \left( \left( \mathbf{S}_{\boldsymbol{\lambda}}(t) - \frac{\lambda \mathbf{I}}{2} \right)^{\frac{1}{2}} \quad 0_{\max(0, N_i - N_o)} \right) \qquad \mathbf{S}_2(t) = \left( \left( \mathbf{S}_{\boldsymbol{\lambda}}(t) + \frac{\lambda \mathbf{I}}{2} \right)^{\frac{1}{2}} 0_{\max(0, N_o - N_i)} \right) \tag{141}$$

*Proof.* We prove the case $N_i \leq N_o$ and $N_h = min(N_i, N_o)$. The proof for $N_o \leq N_i$ follows the same structure. Let $\mathbf{U}\mathbf{S}\mathbf{V}^T = \mathbf{W}_2(t)\mathbf{W}_1(t)$ be the Singular Value Decomposition of the product of the weights at training step $t$. We will use $\mathbf{W}_2 = \mathbf{W}_2(t), \mathbf{W}_1 = \mathbf{W}_1(t)$ as a shorthand.

We write the initialisation for our setting $\mathbf{W}_2 \mathbf{W}_1 = \mathbf{U}\mathbf{S}\mathbf{V}^T$. We therefore can write without loss of generality the weight matrices $\mathbf{W}_2 = \mathbf{U}\mathbf{S}_2 \mathbf{G}_2$ and $\mathbf{W}_1 = \mathbf{G}_1 \mathbf{S}_1 \mathbf{V}^T$. In this case we requiere that $\mathbf{G}_2 = \mathbf{G}_1^{-1}$ and $\mathbf{S}_1 \mathbf{S}_2 = \mathbf{S}$.

We assume the balanced property such that $\mathbf{W}_2^T \mathbf{W}_2 - \mathbf{W}_1 \mathbf{W}_1^T = \lambda \mathbf{I}$. We know this holds for any $t$ since this is a conserved quantity in linear networks. The matrices $\mathbf{W}_1 \mathbf{W}_1^T$ and $\mathbf{W}_2^T \mathbf{W}_2$ are symmetric, which consequently implies that their singular vectors are orthogonal. Consequently, in our specific scenario, we require that $\mathbf{G}_1$ and $\mathbf{G}_2$ are orthogonal matrices and it follows that

$$\mathbf{G}_2 = \mathbf{G}_1^{-1} = \mathbf{G}_1^T \tag{142}$$

$$\mathbf{G}_2 = \mathbf{G}_1^T = \mathbf{R}^T \tag{143}$$

We can write $\mathbf{W}_2 = \mathbf{U}\mathbf{S}_2 \mathbf{R}^T, \mathbf{W}_1 = \mathbf{R}\mathbf{S}_1 \mathbf{V}^T$, where $\mathbf{R}$ is an orthonormal matrix and $\mathbf{S}_2, \mathbf{S}_1$ are diagonal (possibly rectangular) matrices.

Hence

$$\mathbf{R}\mathbf{S}_2^T \mathbf{S}_2 \mathbf{R}^T - \mathbf{R}\mathbf{S}_1 \mathbf{S}_1 \mathbf{R}^T = \lambda \mathbf{I} \tag{144}$$

$$\mathbf{S}_2^T \mathbf{S}_2 - \mathbf{S}_1 \mathbf{S}_1 = \lambda \mathbf{I} \tag{145}$$

The matrices $\mathbf{S}_1, \mathbf{S}_2$, have shapes $(N_h, N_i), (N_o, N_h)$ respectively. We introduce the diagonal matrices $\hat{\mathbf{S}}_1$ of shape $(N_h, N_i)$, $\hat{\mathbf{S}}_2$ of shape $(N_i, N_h)$ such that the zero matrix has size $(N_o - N_i, N_h)$ :

$$\mathbf{S}_1 = \left( \hat{\mathbf{S}}_1 \right), \quad \mathbf{S}_2 = \begin{pmatrix} \hat{\mathbf{S}}_2 \\ 0 \end{pmatrix} \tag{146}$$

Hence

$$\mathbf{S}_2^T \mathbf{S}_2 - \mathbf{S}_1 \mathbf{S}_1 = \lambda \mathbf{I} \tag{147}$$

From the equation above and the fact that $\hat{S}_1 \hat{S}_2 = S$ we derive that:

$$\hat{S}_2 = \left( \frac{\sqrt{\lambda^2 \mathbf{I} + 4S^2} + \lambda \mathbf{I}}{2} \right)^{\frac{1}{2}}, \quad \hat{S}_1 = \left( \frac{\sqrt{\lambda^2 \mathbf{I} + 4S^2} - \lambda \mathbf{I}}{2} \right)^{\frac{1}{2}}, \tag{148}$$

Hence

$$W_2 = U \left( \begin{array}{c} \left( \frac{\sqrt{\lambda^2 \mathbf{I} + 4S^2} + \lambda \mathbf{I}}{2} \right)^{\frac{1}{2}} \\ 0_{\max(0, N_o - N_i)} \end{array} \right), R^T, \quad W_1 = R \left( \left( \frac{\sqrt{\lambda^2 \mathbf{I} + 4S^2} - \lambda \mathbf{I}}{2} \right)^{\frac{1}{2}} \quad 0_{\max(0, N_i - N_o)} \right) V^T \tag{149}$$

$\square$

### C.2.2 CONVERGENCE PROOF

With our solution, $\mathbf{QQ}^T(t)$, which captures the temporal dynamics of the similarity between hidden layer activations, we can analyze the network's internal representations in relation to the task. This allows us to determine whether the network adopts a *rich* or *lazy* representation, depending on the value of $\lambda$. Consider a $\lambda$-Balanced network training on data $\Sigma^{yx} = \tilde{U}\tilde{S}\tilde{V}^T$. We assume that the convergence is toward global minima and B is invertible

**Theorem C.3.** *Under the assumptions of Theorem B.5, the network function converges to $\tilde{U}\tilde{S}\tilde{V}^T$ and acquires the internal representation, that is $\mathbf{W}_1^T\mathbf{W}_1 = \tilde{V}\tilde{S}_1^2\tilde{V}^T$ and $\mathbf{W}_2\mathbf{W}_2^T = \tilde{U}\tilde{S}_2^2\tilde{U}^T$*

*Proof.* As training time increases, all terms including a matrix exponential with negative exponent in Equation 77 vanish to zero, as $S_\lambda = \tilde{S}_\lambda$ is a diagonal matrix with entries larger zero
As training time increases, all terms in the equations vanish to zero. Terms in Equation 77 decay as

$$\lim_{t \to \infty} e^{-\sqrt{\tilde{S}^2 + \frac{\lambda^2 \mathbf{I}}{4}} \frac{t}{\tau}} = \mathbf{0}. \tag{150}$$

and

$$\lim_{t \to \infty} e^{\lambda_\perp \frac{t}{\tau}} e^{-\sqrt{\tilde{S}^2 + \frac{\lambda^2}{4} \mathbf{I}} \frac{t}{\tau}} = \mathbf{0}. \tag{151}$$

where $\tilde{S}_\lambda = \tilde{S}_\lambda$ is a diagonal matrix with entries larger zero
Therefore, in the temporal limit, eq. 77 reduces to

$$\lim_{t \to \infty} \mathbf{QQ}^T(t) = \lim_{t \to \infty} \begin{bmatrix} \mathbf{W}_1^T\mathbf{W}_1(t) & \mathbf{W}_1^T\mathbf{W}_2^T(t) \\ \mathbf{W}_2\mathbf{W}_1(t) & \mathbf{W}_2^T\mathbf{W}_2(t) \end{bmatrix} \tag{152}$$

$$= \begin{bmatrix} \tilde{V}(\tilde{G} - \tilde{H}\tilde{G}) \\ \tilde{U}(\tilde{H}\tilde{G} + \tilde{G}) \end{bmatrix} \left[ \tilde{S}_\lambda^{-1} \right]^{-1} \left[ (\tilde{V}(\tilde{G} - \tilde{H}\tilde{G}))^T \quad (\tilde{U}(\tilde{H}\tilde{G} + \tilde{G}))^T \right] \tag{153}$$

$$= \begin{bmatrix} \tilde{V}(\tilde{G} - \tilde{H}\tilde{G})\tilde{S}_\lambda(\tilde{G} - \tilde{H}\tilde{G})^T\tilde{V}^T & \tilde{V}(\tilde{G} - \tilde{H}\tilde{G})\tilde{S}_\lambda(\tilde{H}\tilde{G} + \tilde{G})^T\tilde{U}^T \\ \tilde{U}(\tilde{H}\tilde{G} + \tilde{G})\tilde{S}_\lambda(\tilde{G} - \tilde{H}\tilde{G})^T\tilde{V}^T & \tilde{U}(\tilde{H}\tilde{G} + \tilde{G})\tilde{S}_\lambda(\tilde{H}\tilde{G} + \tilde{G})^T\tilde{U}^T \end{bmatrix}. \tag{154}$$

$$(\tilde{G} - \tilde{H}\tilde{G})\tilde{S}_\lambda(\tilde{G} + \tilde{H}\tilde{G}) = \frac{S_\lambda(1 - \tilde{H}^2)}{1 + \tilde{H}^2} = \tilde{S} \tag{155}$$

$$\tilde{S}_\lambda(\tilde{G} - \tilde{H}\tilde{G})^2 = \frac{\tilde{S}_\lambda(1 + \tilde{H}^2)}{1 + \tilde{H}^2} - \frac{\tilde{S}_\lambda(2\tilde{H})}{1 + \tilde{H}^2} = \frac{\sqrt{4\tilde{S}^2 + \lambda^2 \mathbf{I}} - \lambda \mathbf{I}}{2} \tag{156}$$

$$\tilde{S}_\lambda(\tilde{G} + \tilde{H}\tilde{G})^2 = \frac{\tilde{S}_\lambda(1 + \tilde{H}^2)}{1 + \tilde{H}^2} + \frac{\tilde{S}_\lambda(2\tilde{H})}{1 + \tilde{H}^2} = \frac{\sqrt{4\tilde{S}^2 + \lambda^2 \mathbf{I}} + \lambda \mathbf{I}}{2} \tag{157}$$

$$\lim_{t \to \infty} \mathbf{QQ}^T(t) = \lim_{t \to \infty} \begin{bmatrix} \mathbf{W}_1^T\mathbf{W}_1(t) & \mathbf{W}_1^T\mathbf{W}_2^T(t) \\ \mathbf{W}_2\mathbf{W}_1(t) & \mathbf{W}_2^T\mathbf{W}_2(t) \end{bmatrix} \tag{158}$$

$$= \begin{bmatrix} \tilde{V}S_1^2\tilde{V}^T & \tilde{V}\tilde{S}\tilde{U}^T \\ \tilde{U}\tilde{S}\tilde{V}^T & \tilde{U}S_2^2\tilde{U}^T \end{bmatrix}. \tag{159}$$

$\square$

### C.2.3 Representation in the limit

**Theorem C.4.** *Under the assumptions of Theorem B.5, training on data* $\boldsymbol{\Sigma}^{yx} = \tilde{\boldsymbol{U}}\tilde{\boldsymbol{S}}\tilde{\boldsymbol{V}}^T$, *as* $\lambda \to \infty$ *the representation tends to*

$$\boldsymbol{W}_2\boldsymbol{W}_2^T = \tilde{\boldsymbol{U}} \begin{pmatrix} \lambda\mathbf{I} & 0_{\max(0,N_o-N_i)} \\ 0_{\max(0,N_o-N_i)} & 0 \end{pmatrix} \tilde{\boldsymbol{U}}^T \quad \boldsymbol{W}_1^T\boldsymbol{W}_1 = \frac{1}{\lambda}\tilde{\boldsymbol{V}} \begin{pmatrix} \tilde{\boldsymbol{S}}^2 & 0_{max(0,N_i-N_o)} \\ 0_{\max(0,N_i-N_o)} & 0 \end{pmatrix} \tilde{\boldsymbol{V}}^T$$

*As* $\lambda \to -\infty$

$$\boldsymbol{W}_2\boldsymbol{W}_2^T = -\frac{1}{\lambda}\tilde{\boldsymbol{U}} \begin{pmatrix} \tilde{\boldsymbol{S}}^2 & 0_{\max(0,N_o-N_i)} \\ 0_{\max(0,N_o-N_i)} & 0 \end{pmatrix} \tilde{\boldsymbol{U}}^T, \quad \boldsymbol{W}_1^T\boldsymbol{W}_1 = \tilde{\boldsymbol{V}} \begin{pmatrix} -\lambda\mathbf{I} & 0_{\max(0,N_i-N_o)} \\ 0_{\max(0,N_i-N_o)} & 0 \end{pmatrix} \tilde{\boldsymbol{V}}^T$$

*As* $\lambda \to -\infty$

$$\boldsymbol{W}_2\boldsymbol{W}_2^T = -\frac{1}{\lambda}\tilde{\boldsymbol{U}} \begin{pmatrix} \tilde{\boldsymbol{S}}^2 & 0_{\max(0,N_o-N_i)} \\ 0_{\max(0,N_o-N_i)} & 0 \end{pmatrix} \tilde{\boldsymbol{U}}^T, \quad \boldsymbol{W}_1^T\boldsymbol{W}_1 = \tilde{\boldsymbol{V}} \begin{pmatrix} -\lambda\mathbf{I} & 0_{\max(0,N_i-N_o)} \\ 0_{\max(0,N_i-N_o)} & 0 \end{pmatrix} \tilde{\boldsymbol{V}}^T$$

*Proof.* We start from the representation derived in C.3 and using the Taylor expansion of $f(x) = \sqrt{1+x^2}$, we compute

$$\frac{\sqrt{\lambda^2\mathbf{I} + 4\tilde{\boldsymbol{S}}^2} + \lambda\mathbf{I}}{2} = \frac{|\lambda|\sqrt{1 + \left(\frac{2\tilde{\boldsymbol{S}}}{\lambda}\right)^2} + \lambda\mathbf{I}}{2} \tag{160}$$

$$\frac{|\lambda|\left(1 + \left(\frac{2\tilde{\boldsymbol{S}}}{\lambda}\right)^2 + O(\lambda^{-4})\right) + \lambda\mathbf{I}}{2} = \frac{|\lambda| + \lambda}{2} + \frac{\tilde{\boldsymbol{S}}^2}{|\lambda|} + O(\lambda^{-3}) \tag{161}$$

Hence

$$\lim_{\lambda\to\infty} \frac{\sqrt{\lambda^2\mathbf{I} + 4\tilde{\boldsymbol{S}}^2} + \lambda\mathbf{I}}{2} = \lambda\mathbf{I}, \quad \lim_{\lambda\to-\infty} \frac{\sqrt{\lambda^2\mathbf{I} + 4\tilde{\boldsymbol{S}}^2} + \lambda\mathbf{I}}{2} = \frac{\tilde{\boldsymbol{S}}^2}{|\lambda|} = -\frac{\tilde{\boldsymbol{S}}^2}{\lambda} \tag{162}$$

Similarly,

$$\frac{\sqrt{\lambda^2\mathbf{I} + 4\tilde{\boldsymbol{S}}^2} - \lambda\mathbf{I}}{2} = \frac{|\lambda| - \lambda}{2} + \frac{\tilde{\boldsymbol{S}}^2}{|\lambda|} + O(\lambda^{-3}) \tag{163}$$

$$\lim_{\lambda\to\infty} \frac{\sqrt{\lambda^2\mathbf{I} + 4\tilde{\boldsymbol{S}}^2} - \lambda\mathbf{I}}{2} = \frac{\tilde{\boldsymbol{S}}^2}{\lambda}, \quad \lim_{\lambda\to-\infty} \frac{\sqrt{\lambda^2\mathbf{I} + 4\tilde{\boldsymbol{S}}^2} - \lambda\mathbf{I}}{2} = \frac{\tilde{\boldsymbol{S}}^2}{|\lambda|} = -\lambda\mathbf{I} \tag{164}$$

Since $\tilde{\boldsymbol{U}}, \tilde{\boldsymbol{V}}$ are independent of $\lambda$:

$$\lim_{\lambda\to\pm\infty} \boldsymbol{W}_2\boldsymbol{W}_2^T = \tilde{\boldsymbol{U}} \left(\lim_{\lambda\to\pm\infty} \boldsymbol{S}_2\right) \tilde{\boldsymbol{U}}^T \tag{165}$$

$$\lim_{\lambda\to\pm\infty} \boldsymbol{W}_1^T\boldsymbol{W}_1 = \tilde{\boldsymbol{V}} \left(\lim_{\lambda\to\pm\infty} \boldsymbol{S}_1\right) \tilde{\boldsymbol{V}}^T \tag{166}$$

$\square$

As $|\lambda| \to \infty$, one of the network representations approaches a scaled identity matrix, while the other tends toward zero. Intuitively, this suggests that the representations shift less and less as $|\lambda|$ increases. Next, we demonstrate that the NTK becomes progressively less variable as $|\lambda|$ grows and ultimately converges to zero.

### C.2.4 NTK movement

Relationship between $\lambda$ and the NTK of the network

**Theorem C.5.** *Under the assumptions of Theorem B.5, consider a linear network training on data* $\boldsymbol{\Sigma}^{yx} = \tilde{\boldsymbol{U}}\tilde{\boldsymbol{S}}\tilde{\boldsymbol{V}}^T$. *At any arbitrary training time* $t \geq 0$, *let* $\boldsymbol{W}_2(t)\boldsymbol{W}_1(t) = \boldsymbol{U}^*\boldsymbol{S}^*\boldsymbol{V}^{*T}$. *Then,*

1. *For any $\lambda \in \mathbf{R}$:*

$$NTK(0) = \mathbf{I}_{N_o} \otimes \boldsymbol{X}^T \boldsymbol{V} \begin{pmatrix} \frac{\sqrt{\lambda^2 \mathbf{I} + 4\boldsymbol{S}^{*2}} - \lambda \mathbf{I}}{2} & 0 \\ 0 & 0 \end{pmatrix} \boldsymbol{V}^T \boldsymbol{X}$$
$$+ \boldsymbol{U} \begin{pmatrix} \frac{\sqrt{\lambda^2 \mathbf{I} + 4\boldsymbol{S}^{*2}} + \lambda \mathbf{I}}{2} & 0 \\ 0 & 0 \end{pmatrix} \boldsymbol{U}^T \otimes \boldsymbol{X}^T \boldsymbol{X} \tag{167}$$

$$NTK(t) = \mathbf{I}_{N_o} \otimes \boldsymbol{X}^T \boldsymbol{V}^* \begin{pmatrix} \frac{\sqrt{\lambda^2 \mathbf{I} + 4\boldsymbol{S}^{*2}} - \lambda \mathbf{I}}{2} & 0 \\ 0 & 0 \end{pmatrix} \boldsymbol{V}^{*T}$$
$$+ \boldsymbol{U}^* \begin{pmatrix} \frac{\sqrt{\lambda^2 \mathbf{I} + 4\boldsymbol{S}^{*2}} + \lambda \mathbf{I}}{2} & 0 \\ 0 & 0 \end{pmatrix} \boldsymbol{U}^{*T} \otimes \boldsymbol{X}^T \boldsymbol{X} \tag{168}$$

2. *As $\lambda \to \infty$:*

$$NTK(t) - NTK(0) \to \frac{1}{\lambda} \left( \mathbf{I}_{N_o} \otimes \boldsymbol{X}^T \boldsymbol{V}^* \tilde{\boldsymbol{S}}^{*2} \boldsymbol{V}^{*T} \boldsymbol{X} - \mathbf{I}_{N_o} \otimes \boldsymbol{X}^T \boldsymbol{V} \tilde{\boldsymbol{S}}^2 \boldsymbol{V}^T \boldsymbol{X} \right) \to 0 \tag{169}$$

3. *As $\lambda \to -\infty$:*

$$NTK(t) - NTK(0) \to \frac{1}{\lambda} \left( \boldsymbol{U} \tilde{\boldsymbol{S}}^2 \boldsymbol{U}^T \otimes \boldsymbol{X}^T \boldsymbol{X} - \boldsymbol{U}^* \tilde{\boldsymbol{S}}^{*2} \boldsymbol{U}^{*T} \otimes \boldsymbol{X}^T \boldsymbol{X} \right) \to 0 \tag{170}$$

**Proof.** Follows by substituting the expressions for the network representations in terms of $\lambda$ from (Braun et al. (2022))'s expression for the NTK of a linear network. Similarly, follows from substituting the limit expressions for the network representations and the fact that the Kronecker product is linear in both arguments. $\qquad \square$

The theorem above demonstrates that as $|\lambda| \to \infty$, the NTK of a $\lambda$-Balanced network remains constant. This indicates that the network operates in the *lazy* regime throughout all training steps. The $\lambda$-balanced condition imposes a relationship between the singular values of the two weight matrices. Specifically, if $\mathbf{W}_2$ and $\mathbf{W}_1$ are $\lambda$-balanced and satisfy $\mathbf{W}_2 \mathbf{W}_1 = \boldsymbol{\Sigma}_{\mathbf{yx}}$, then for arbitrary singular values $a_i, b_i$, and $s_i$, the following relations hold:

$$a_i^2 - b_i^2 = \lambda, \quad a_i \cdot b_i = s_i.$$

As $\lambda$ increases, the value of $b_i$ must decrease. In the limit as $\lambda \to \infty$, $a_i^2 \to \lambda$ and $b_i^2 \to 0$. From the first equation, when $b_i^2 \to 0$, $a_i^2 \to \lambda$. Since these equations apply to all singular values of the matrices, it follows that for all $i$, $a_i^2 \to \lambda$, leading to the conclusion that:

$$\mathbf{W}_2^T \mathbf{W}_2 = \lambda \mathbf{I},$$

as expected. Consequently, the task representation becomes task-agnostic in $\mathbf{W}_1$. The intuition here is that the weights are constrained by the need to fit the data, which bounds their overall norms. The $\lambda$-balanced condition further specifies a relationship between these norms, and as $|\lambda|$ increases, this constraint tightens, driving $\mathbf{W}_2$ toward the identity matrix. In this regime, the network behaves similarly to a shallow network, with $\lambda$ acting as a toggle between deep and shallow learning dynamics. This finding is significant as it highlights the impact of weight initialization on learning regimes.

## C.3 REPRESENTATION ROBUSTNESS AND SENSITIVITY TO NOISE

As derived in (Braun et al., 2025), the expected mean squared error under additive, independent and identically distributed input noise with mean $\mu = 0$ and variance $\sigma_{\mathbf{x}}^2$ is

$$\left\langle \frac{1}{2P} \sum_{i=1}^{P} ||\mathbf{W}_2 \mathbf{W}_1 (\mathbf{x}_i + \xi_{\mathbf{x}}) - \mathbf{y}_i||_2^2 \right\rangle_{\xi_{\mathbf{x}}} = \sigma_{\mathbf{x}}^2 ||\mathbf{W}_2 \mathbf{W}_1||_F^2 + c, \tag{171}$$

where $c = \frac{1}{2} \text{Tr}(\tilde{\boldsymbol{\Sigma}}^{yy}) - \frac{1}{2} \text{Tr}(\tilde{\boldsymbol{\Sigma}}^{yx} \tilde{\boldsymbol{\Sigma}}^{yxT})$ is a noise independent constant that only depends on the statistics of the training data. In Theorem C.3 we show that the network function converges to $\tilde{\mathbf{U}} \tilde{\mathbf{S}} \tilde{\mathbf{V}}^T$ and therefore

$$\sigma_{\mathbf{x}}^2 ||\mathbf{W}_2 \mathbf{W}_1||_F^2 = \sigma_{\mathbf{x}}^2 ||\tilde{\mathbf{U}} \tilde{\mathbf{S}} \tilde{\mathbf{V}}^T||_F^2$$
$$= \sigma_{\mathbf{x}}^2 ||\tilde{\mathbf{S}}||_F^2$$
$$= \sigma_{\mathbf{x}}^2 \sum_{i=1}^{N_h} \tilde{\mathbf{S}}_i^2 \tag{172}$$

As derived in (Braun et al., 2025), under the assumption of whitened inputs (Assumption 1), in the case of additive parameter noise with $\mu = 0$ and variance $\sigma_{\mathbf{W}}^2$, the expected mean squared error is

$$\left\langle \frac{1}{2P} \sum_{i=1}^{P} || \left( \mathbf{W}_2 + \xi_{\mathbf{W}_2} \right) \left( \mathbf{W}_1 + \xi_{\mathbf{W}_1} \right) \mathbf{x}_i - \mathbf{y}_i ||_2^2 \right\rangle_{\xi_{\mathbf{W}_1}, \xi_{\mathbf{W}_2}} \tag{173}$$

$$= \frac{1}{2} N_i \sigma_{\mathbf{W}}^2 ||\mathbf{W}_2||_F^2 + \frac{1}{2} N_o \sigma_{\mathbf{W}}^2 ||\mathbf{W}_1||_F^2 + \frac{1}{2} N_i N_h N_o \sigma^4 + c.$$

Using Theorem C.3, we have

$$\begin{aligned}
||\mathbf{W}_1||_F^2 &= \mathrm{Tr}(\mathbf{W}_1^T \mathbf{W}_1) \\
&= \mathrm{Tr}\left( \frac{\sqrt{\lambda^2 \mathbf{I} + 4\tilde{\mathbf{S}}^2} + \lambda \mathbf{I}}{2} \right) \\
&= \frac{1}{2} \left( \sum_{i=1}^{N_h} \sqrt{\lambda^2 + 4\tilde{\mathbf{S}}_i^2} + \lambda \right)
\end{aligned} \tag{174}$$

and

$$\begin{aligned}
||\mathbf{W}_2||_F^2 &= \mathrm{Tr}(\mathbf{W}_2 \mathbf{W}_2^T) \\
&= \mathrm{Tr}\left( \frac{\sqrt{\lambda^2 \mathbf{I} + 4\tilde{\mathbf{S}}^2} - \lambda \mathbf{I}}{2} \right) \\
&= \frac{1}{2} \left( \sum_{i=1}^{N_h} \sqrt{\lambda^2 + 4\tilde{\mathbf{S}}_i^2} - \lambda \right).
\end{aligned} \tag{175}$$

To find the $\lambda$ that minimises the expected loss, we substitute the equations for the norms, take the partial derivative with respect to $\lambda$ and set it to zero

$$\begin{aligned}
&\frac{\partial \langle \mathcal{L} \rangle_{\xi_{\mathbf{W}_1}, \xi_{\mathbf{W}_2}}}{\partial \lambda} \overset{!}{=} 0 \\
\Leftrightarrow &\frac{1}{4} N_i \sigma_{\mathbf{W}}^2 \frac{\partial}{\partial \lambda} \left( \sum_{i=1}^{N_h} \sqrt{\lambda^2 + 4\tilde{\mathbf{S}}_i^2} - \lambda \right) + \frac{1}{4} N_o \sigma_{\mathbf{W}}^2 \frac{\partial}{\partial \lambda} \left( \sum_{i=1}^{N_h} \sqrt{\lambda^2 + 4\tilde{\mathbf{S}}_i^2} + \lambda \right) = 0 \\
\Leftrightarrow &N_i \sum_{i=1}^{N_h} \frac{\lambda}{\sqrt{\lambda^2 + 4\tilde{\mathbf{S}}_i^2}} - N_i N_h + N_o \sum_{i=1}^{N_h} \frac{\lambda}{\sqrt{\lambda^2 + 4\tilde{\mathbf{S}}_i^2}} + N_o N_h = 0 \\
\Leftrightarrow &\sum_{i=1}^{N_h} \frac{\lambda}{\sqrt{\lambda^2 + 4\tilde{\mathbf{S}}_i^2}} = N_h \frac{N_i - N_o}{N_i + N_o}.
\end{aligned} \tag{176}$$

It follows, that under the assumption that $N_i = N_o$, the equation reduces to

$$\sum_{i=1}^{N_h} \frac{\lambda}{\sqrt{\lambda^2 + 4\tilde{\mathbf{S}}_i^2}} = 0. \tag{177}$$

We note, that the denominator is always positive and therefore, that the left-hand side of the equation is always larger zero for any $\lambda > 0$, and smaller than zero for any $\lambda < 0$. The euqation is therefore only solved for $\lambda = 0$.

## C.4 Effect of the architecture from the lazy to the Rich Regime

**Theorem C.6.** *Under the conditions of Theorem B.5, when $\lambda_\perp > 0$, the network enters a regime referred to as the delayed-rich phase. In this phase, the learning rate is determined by two competing exponential factors:*

$$e^{\lambda_\perp \frac{t}{\tau}} e^{-\sqrt{\tilde{S}^2 + \frac{\lambda^2}{4} \mathbf{I}} \frac{t}{\tau}}$$

*and*

$$e^{-\sqrt{\bar{S}^2 + \frac{\lambda^2}{4} \mathbf{I}} \frac{t}{\tau}}.$$

*As $\lambda$ increases, various parts of the network display different learning dynamics: some components adjust rapidly, converging exponentially with $\lambda$, while others adapt more slowly, with their convergence rate inversely proportional to $\lambda$, leading to a slow adaptation.*

*Proof.* The solution to Theorem B.5 is governed by two time-dependent terms:

$$e^{-\sqrt{\tilde{\boldsymbol{S}}^2 + \frac{\lambda^2 \mathbf{I}}{4}} \frac{t}{\tau}} \quad \text{and} \quad e^{\lambda_\perp \frac{t}{\tau}} e^{-\sqrt{\tilde{\boldsymbol{S}}^2 + \frac{\lambda^2}{4} \mathbf{I}} \frac{t}{\tau}}.$$

The first term exhibits exponential decay approaching zero as time progresses:

$$\lim_{t \to \infty} e^{-\sqrt{\tilde{\boldsymbol{S}}^2 + \frac{\lambda^2 \mathbf{I}}{4}} \frac{t}{\tau}} = \mathbf{0}.$$

In the limit as lambda gets large the rate of learning is given by

$$\lim_{\lambda \to \infty} \frac{\sqrt{\lambda^2 \mathbf{I} + 4\tilde{\boldsymbol{S}}^2}}{2} = \frac{\lambda \mathbf{I}}{2}, \tag{178}$$

The second term also decays over time

$$\lim_{t \to \infty} e^{\lambda_\perp \frac{t}{\tau}} e^{-\sqrt{\tilde{\boldsymbol{S}}^2 + \frac{\lambda^2}{4} \mathbf{I}} \frac{t}{\tau}} = \mathbf{0}.$$

but in the limit as lambda gets large the rate of learning is given by

$$\lim_{\lambda \to \infty} \frac{\sqrt{\lambda^2 \mathbf{I} + 4\tilde{\boldsymbol{S}}^2} - \lambda \mathbf{I}}{2} = \frac{\tilde{\boldsymbol{S}}^2}{\lambda} \tag{179}$$

Thus, as $\lambda$ increases, the convergence rate slows for certain parts of the network, while other components continue to learn more quickly. This explains the delay observed in the delayed-rich regime. $\square$

## D APPENDIX: APPLICATION

### D.1 APPENDIX: CONTINUAL LEARNING

We build upon the derivation presented in Braun et al. (2022) to incorporate the dynamics of continual learning throughout the entire learning trajectory. Utilizing the assumption of whitened inputs, the entire batch loss for the $i$th task is

$$
\begin{aligned}
\mathcal{L}_i\left(\mathcal{T}_j\right) &= \frac{1}{2P} \left\|\mathbf{W}_2 \mathbf{W}_1 \mathbf{X}_i - \mathbf{Y}_i\right\|_F^2 \\
&= \frac{1}{2P} \operatorname{Tr}\left(\left(\mathbf{W}_2 \mathbf{W}_1 \mathbf{X}_i - \mathbf{Y}_i \mid\right)\left(\mathbf{W}_2 \mathbf{W}_1 \mathbf{X}_i - \mathbf{Y}_i \mid\right)^T\right) \\
&= \frac{1}{2P} \operatorname{Tr}\left(\mathbf{W}_2 \mathbf{W}_1 \mathbf{X}_i \mathbf{X}_i^T (\mathbf{W}_2 \mathbf{W}_1)^T\right) - \frac{1}{P} \operatorname{Tr}\left(\mathbf{W}_2 \mathbf{W}_1 \mathbf{X}_i \mathbf{Y}_i^T\right) + \frac{1}{2P} \operatorname{Tr}\left(\mathbf{Y}_i \mathbf{Y}_i^T\right) \\
&= \frac{1}{2} \operatorname{Tr}\left(\mathbf{W}_2 \mathbf{W}_1 (\mathbf{W}_2 \mathbf{W}_1)^T\right) - \operatorname{Tr}\left(\mathbf{W}_2 \mathbf{W}_1 \tilde{\boldsymbol{\Sigma}}_i^{yx^T}\right) + \frac{1}{2} \operatorname{Tr}\left(\tilde{\boldsymbol{\Sigma}}_i^{yy}\right) \\
&= \frac{1}{2} \operatorname{Tr}\left(\left(\mathbf{W}_2 \mathbf{W}_1 - \tilde{\boldsymbol{\Sigma}}_i^{yx}\right)\left(\mathbf{W}_2 \mathbf{W}_1 - \tilde{\boldsymbol{\Sigma}}_i^{yx}\right)^T - \tilde{\boldsymbol{\Sigma}}_i^{yx} \tilde{\boldsymbol{\Sigma}}_i^{yx^T}\right) + \frac{1}{2}\left(\tilde{\boldsymbol{\Sigma}}_i^{yy}\right) \\
&= \frac{1}{2} \left\|\mathbf{W}_2 \mathbf{W}_1 - \tilde{\boldsymbol{\Sigma}}_i^{yx}\right\|_F^2 \underbrace{- \frac{1}{2} \operatorname{Tr}\left(\tilde{\boldsymbol{\Sigma}}_i^{yx} \tilde{\boldsymbol{\Sigma}}_i^{yx^T}\right) + \frac{1}{2}\left(\tilde{\boldsymbol{\Sigma}}_i^{yy}\right)}_{c}.
\end{aligned}
$$

Hence, the extent of forgetting, denoted as $\mathcal{F}$ for task $\mathcal{T}_i$ during training on task $\mathcal{T}_k$ subsequent to training the network on task $\mathcal{T}_j$, specifically, the relative change in loss, is entirely dictated by the similarity structure among tasks.

$$
\begin{aligned}
\mathcal{F}_i\left(\mathcal{T}_j, \mathcal{T}_k\right) &= \mathcal{L}_i\left(\mathcal{T}_k\right) - \mathcal{L}_i\left(\mathcal{T}_j\right) \\
&= \frac{1}{2} \left\|\tilde{\boldsymbol{\Sigma}}_k^{yx} - \tilde{\boldsymbol{\Sigma}}_i^{yx}\right\|_F^2 + c - \frac{1}{2} \left\|\mathbf{W}_2 \mathbf{W}_1 - \tilde{\boldsymbol{\Sigma}}_i^{yx}\right\|_F^2 - c \\
&= \frac{1}{2} \left(\left\|\tilde{\boldsymbol{\Sigma}}_k^{yx} - \tilde{\boldsymbol{\Sigma}}_i^{yx}\right\|_F^2 - \left\|\mathbf{W}_2 \mathbf{W}_1 - \tilde{\boldsymbol{\Sigma}}_i^{yx}\right\|_F^2\right).
\end{aligned}
$$

It is important to note that the amount of forgetting is a function of the weight trajectories. Therefore, we have analytical solutions for trajectories of forgetting as well.

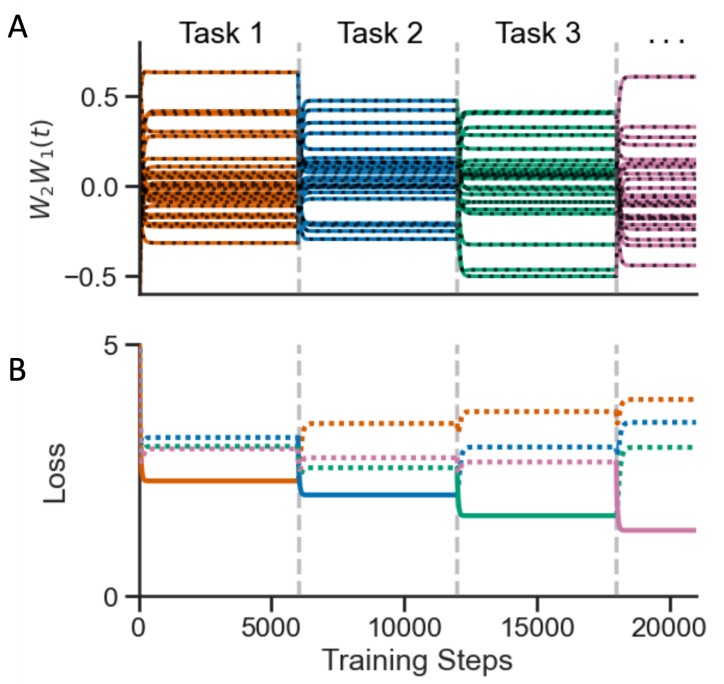

Figure 11: Continual learning. **A** Top: Network training from small zero-balanced weights across a sequence of tasks (colored lines represent simulations, and black dotted lines represent analytical results). Bottom: Evaluation loss for the tasks in the sequence (dotted lines) while training on the current task (solid lines). As the network optimizes its function on the current task, the loss on previously learned tasks increases.

Figure. D.1 panel was generated by training a linear network with $N_i = 5$, $N_h = 10$, $N_o = 6$ subsequently on four different random regression tasks with $N = 25$. The learning rate was $\eta = 0.05$ and the initial weights were small ($\sigma = 0.0001$).

## D.2 APPENDIX: REVERSAL LEARNING

As first introduced in Braun et al. (2022), in the following discussion, we assume that the input and output dimensions are equal. We denote the $i$-th columns of the left and right singular vectors as $\mathbf{u}_i$, $\tilde{\mathbf{u}}_i$, and $\mathbf{v}_i$, $\tilde{\mathbf{v}}_i$, respectively.
Reversal learning occurs when both the task and the initial network function share the same left and right singular vectors, i.e., $\mathbf{U} = \tilde{\mathbf{U}}$ and $\mathbf{V} = \tilde{\mathbf{V}}$, with the exception of one or more columns of the left singular vectors, where the direction is reversed: $-\mathbf{u}_i = \tilde{\mathbf{u}}_i$.
It is important to note that if a reversal occurs in the right singular vectors, such that $-\mathbf{v}_i = \tilde{\mathbf{v}}_i$, this can be equivalently represented as a reversal in the left singular vectors, as the signs of the right and left singular vectors are interchangeable.
In the reversal learning setting, both $\boldsymbol{B} = \boldsymbol{S}_2 \tilde{\boldsymbol{U}}^T \tilde{\boldsymbol{U}} (\tilde{\boldsymbol{G}} + \tilde{\boldsymbol{H}}\tilde{\boldsymbol{G}}) + \boldsymbol{S}_1 \boldsymbol{V}^T \tilde{\boldsymbol{V}} (\tilde{\boldsymbol{G}} - \tilde{\boldsymbol{H}}\tilde{\boldsymbol{G}})$ and $\boldsymbol{C} = \boldsymbol{S}_2 \tilde{\boldsymbol{U}}^T \tilde{\boldsymbol{U}} (\tilde{\boldsymbol{G}} - \tilde{\boldsymbol{H}}\tilde{\boldsymbol{G}}) - \boldsymbol{S}_1 \boldsymbol{V}^T \tilde{\boldsymbol{V}} (\tilde{\boldsymbol{G}} + \tilde{\boldsymbol{H}}\tilde{\boldsymbol{G}})$ are diagonal matrices.

In the case where lambda is zero, the same argument given in Braun et al. (2022) follows, the diagonal entries of $\mathbf{C}$ are zero if the singular vectors are aligned and non zero if they are reversed. Similarly, diagonal entries of $\mathbf{B}$ are non-zero if the singular vectors are aligned and zero if they are reversed. Therefore, in the case of reversal learning, $\mathbf{B}$ is a diagonal matrix with $0$ values and thus is not invertible. As a consequence, the learning dynamics cannot be described by Equation 56. However, as $\mathbf{B}$ and $\mathbf{C}$ are diagonal matrices, the learning dynamics simplify. Let $\mathbf{b}_i$, $\mathbf{c}_i$, $\mathbf{s}_i$ and $\tilde{\mathbf{s}}_i$

denote the $i$-th diagonal entry of $\mathbf{B}$, $\mathbf{C}$, $\mathbf{S}$ and $\tilde{\mathbf{S}}$ respectively, then the network dynamics can be rewritten as

$$
\mathbf{W}_2\mathbf{W}_1(t) = \frac{1}{2}\tilde{\mathbf{U}}\left[(\tilde{\boldsymbol{G}}+\tilde{\boldsymbol{H}}\tilde{\boldsymbol{G}})e^{\tilde{\boldsymbol{S}}_\lambda\frac{t}{\tau}}\mathbf{B}^T + (\tilde{\boldsymbol{G}}-\tilde{\boldsymbol{H}}\tilde{\boldsymbol{G}})e^{-\tilde{\boldsymbol{S}}_\lambda\frac{t}{\tau}}\mathbf{C}^T\right)
$$
$$
\left[\boldsymbol{S}_\lambda^{-1} + \frac{1}{4}\mathbf{B}\left(e^{2\tilde{\boldsymbol{S}}_\lambda\frac{t}{\tau}}-\mathbf{I}\right)\tilde{\mathbf{S}}_\lambda^{-1}\mathbf{B}^T - \frac{1}{4}\mathbf{C}\left(e^{-2\tilde{\boldsymbol{S}}_\lambda\frac{t}{\tau}}-\mathbf{I}\right)\tilde{\mathbf{S}}_\lambda^{-1}\mathbf{C}^T\right]^{-1} \quad (180)
$$
$$
\frac{1}{2}\left((\tilde{\boldsymbol{G}}-\tilde{\boldsymbol{H}}\tilde{\boldsymbol{G}})e^{\tilde{\boldsymbol{S}}_\lambda\frac{t}{\tau}}\mathbf{B} - (\tilde{\boldsymbol{G}}+\tilde{\boldsymbol{H}}\tilde{\boldsymbol{G}})e^{-\tilde{\boldsymbol{S}}_\lambda\frac{t}{\tau}}\mathbf{C}\right)\tilde{\mathbf{V}}^T
$$
$$
= \sum_{i=1}^{N_i}\frac{\mathbf{b}_i^2 e^{2\tilde{\mathbf{s}}_{\lambda i}\frac{t}{\tau}} - \mathbf{c}_i^2 e^{-2\tilde{\mathbf{s}}_{\lambda i}\frac{t}{\tau}}}{4\mathbf{s}_{\lambda i}^{-1}+\mathbf{b}_i^2 e^{2\tilde{\mathbf{s}}_{\lambda i}\frac{t}{\tau}}\tilde{\mathbf{s}}_{\lambda\mathbf{i}}^{-1}-\mathbf{b}_i^2\tilde{\mathbf{s}}_{\lambda\mathbf{i}}^{-1}-\mathbf{c}_i^2 e^{-2\tilde{\mathbf{s}}_{\lambda i}\frac{t}{\tau}}\tilde{\mathbf{s}}_{\lambda\mathbf{i}}^{-1}+\mathbf{c}_i^2\tilde{\mathbf{s}}_{\lambda\mathbf{i}}^{-1}}\tilde{\mathbf{u}}_i\tilde{\mathbf{v}}_i^T \quad (181)
$$
$$
= \sum_{i=1}^{N_i}\frac{\mathbf{s}_{\lambda i}\mathbf{b}_i^2\tilde{\mathbf{s}}_{\lambda\mathbf{i}}-\mathbf{s}_{\lambda i}\mathbf{c}_i^2\tilde{\mathbf{s}}_{\lambda\mathbf{i}}e^{-4\tilde{\mathbf{s}}_i\frac{t}{\tau}}}{4\tilde{\mathbf{s}}_{\lambda\mathbf{i}}e^{-2\tilde{\mathbf{s}}_i\frac{t}{\tau}}+\mathbf{s}_{\lambda i}\mathbf{b}_i^2\left(1-e^{-2\tilde{\mathbf{s}}_{\lambda i}\frac{t}{\tau}}\right)+\mathbf{s}_{\lambda i}\mathbf{c}_i^2\left(e^{-2\tilde{\mathbf{s}}_{\lambda i}\frac{t}{\tau}}-e^{-4\tilde{\mathbf{s}}_{\lambda i}\frac{t}{\tau}}\right)}\tilde{\mathbf{u}}_i\tilde{\mathbf{v}}_i^T
$$
$$
(182)
$$

It follows, that in the reversal learning case, i.e. $\mathbf{b}=0$, for each reversed singular vector, the dynamics vanish to zero

$$
\lim_{t\to\infty}\frac{-\mathbf{s}_{\lambda i}\mathbf{c}_i^2\tilde{\mathbf{s}}_i e^{-4\tilde{\mathbf{s}}_{\lambda i}\frac{t}{\tau}}}{4\tilde{\mathbf{s}}_{\lambda,\mathbf{i}}e^{-2\tilde{\mathbf{s}}_{\lambda i}\frac{t}{\tau}}+\mathbf{s}_i\mathbf{c}_i^2\left(e^{-2\tilde{\mathbf{s}}_{\lambda i}\frac{t}{\tau}}-e^{-4\tilde{\mathbf{s}}_{\lambda i}\frac{t}{\tau}}\right)}\tilde{\mathbf{u}}_i\tilde{\mathbf{v}}_i^T = 0. \quad (183)
$$

Analytically, the learning dynamics are initialized on and remain along the separatrix of a saddle point until the corresponding singular value of the network function decreases to zero and stays there, indicating convergence to the saddle point. In numerical simulations, however, the learning dynamics can escape the saddle points due to the imprecision of floating-point arithmetic. Despite this, numerical optimization still experiences significant delays, as escaping the saddle point is time-consuming Lee et al. (2022). In contrast, when the singular vectors are aligned ($\mathbf{c}=0$), the equation governing temporal dynamics, as described in Saxe et al. (2014), is recovered. Under these conditions, training succeeds, with the singular value of the network function converging to its target value.

$$
\lim_{t\to\infty}\sum_{i=1}^{N_i}\frac{\mathbf{s}_{\lambda\mathbf{i}}\mathbf{b}_i^2\tilde{\mathbf{s}}_{\lambda\mathbf{i}}}{4\tilde{\mathbf{s}}_{\lambda\mathbf{i}}e^{-2\tilde{\mathbf{s}}_{\lambda i}\frac{t}{\tau}}+\mathbf{s}_{\lambda\mathbf{i}}\mathbf{b}_i^2\left(1-e^{-2\tilde{\mathbf{s}}_{\lambda i}\frac{t}{\tau}}\right)}\tilde{\mathbf{u}}_i\tilde{\mathbf{v}}_i^T = \frac{\mathbf{s}_{\lambda\mathbf{i}}\mathbf{b}_i^2\tilde{\mathbf{s}}_{\lambda\mathbf{i}}}{\mathbf{s}_{\lambda\mathbf{i}}\mathbf{b}_i^2}\tilde{\mathbf{u}}_i\tilde{\mathbf{v}}_i^T \quad (184)
$$
$$
= \tilde{\mathbf{s}}_{\lambda\mathbf{i}}\tilde{\mathbf{u}}_i\tilde{\mathbf{v}}_i^T. \quad (185)
$$

In summary, in the case of aligned singular vectors, the learning dynamics can be described by the convergence of singular values. However in the case of reversal learning, analytically, training does not succeed. In simulations, the learning dynamics escape the saddle point due to numerical imprecision, but the learning dynamics are catastrophically slowed in the vicinity of the saddle point as shown in figure D.2 .

In the case where $\lambda$ is non-zero, the diagonal of $\mathbf{C}$ are also non-zero; this is true regardless of whether they are reversed or aligned. Similarly, the diagonal entries of $\mathbf{B}$ remain non-zero whether the singular vectors are aligned or reversed. Therefore, in the case of reversal learning, $\mathbf{B}$ is a diagonal matrix with elements that are zero. In figure D.2

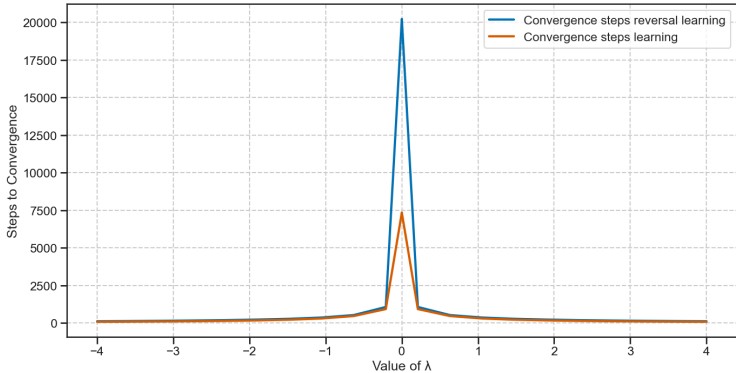

Figure 12: Plot showing the steps to convergence for two tasks: (1) the reversal learning task and (2) a randomly sampled continual learning task across a range of $\lambda$ values. The reversal learning task exhibits catastrophic slowing at $\lambda = 0$.

### D.3 APPENDIX: GENERALIZATION AND STRUCTURED LEARNING

We study how the representations learned for different $\lambda$ initializations impact generalization of properties of the data. To do this, we consider the case where a new feature is associated to a learned item in a dataset and how this new feature may then be related to other items based on prior knowledge. In particular, we first train each network (for different values of $-10 \leq \lambda \leq 10$) on the hierarchical semantic learning task in Section 5 and then add a new feature (e.g., 'eats worms') to a single item (e.g., the goldfish) (Fig. D.3A), correspondingly increasing the output dimension to represent the novel feature. In order to learn the new feature without affecting prior knowledge, we append a randomly initialized row to $\mathbf{W}_2$ and train it on the single item with the new feature, while keeping the rest of the network frozen. Thus, we only change the weights from the hidden layer to the new feature which may produce different behavior depending on how the hidden layer representations vary based on $\lambda$. After training on the new feature-item association, we query the network with the rest of the data to observe how the new feature is associated with the other items. We find that as $\lambda$ increases positively, the network better transfers the hierarchy such that it projects the feature onto items based on their distance to the trained item (Fig. D.3B,C). For example, after learning that a goldfish eats worms, the network can extrapolate the hierarchy to infer that another fish, or birds, may also eat worms; instead, plants are not likely to eat worms. Alternatively, as $\lambda$ becomes more negative, the network ceases to infer any hierarchical structure and only learns to map the new feature to the single item trained on. In this case, after learning that a goldfish eats worms, the network does not infer that other fish, birds, or plants may also eat worms.

Interestingly, this setting highlights how asymmetries in the representations yielded by different $\lambda$ can actually benefit transfer and generalization. This can be shown by observing that the learning of a new feature association only depends on the first layer $\mathbf{W}_1$. Let $\hat{\boldsymbol{y}}_f$ denote the vector of the representation of the new feature $f$ across items $i$ in the dataset. Additionally, let $\boldsymbol{w}_2^{(f)T}$ be the new row of weights appended to $\mathbf{W}_2$ which map the hidden layer to the new feature. Following Saxe et al. (2019b), if $\boldsymbol{w}_2^{(f)T}$ is initialized with small random weights and trained on item $\tilde{\boldsymbol{H}}_i$, it will converge to

$$\boldsymbol{w}_2^{(f)T} = \tilde{\boldsymbol{H}}_i^T \mathbf{W}_1^T / \|\mathbf{W}_1 \tilde{\boldsymbol{H}}_i\|_2^2 \tag{186}$$

$$\hat{\boldsymbol{y}}_f = (\tilde{\boldsymbol{H}}_i^T \mathbf{W}_1^T \mathbf{W}_1 \tilde{\boldsymbol{H}}) / \|\mathbf{W}_1 \tilde{\boldsymbol{H}}_i\|_2^2 \tag{187}$$

From this we can see that differences in the representations of the new feature across items $\hat{\boldsymbol{y}}_f$ across $\lambda$ are only influenced by $\mathbf{W}_1$.

In the case of the rich learning regime where $\lambda = 0$, the semantic relationship between features and items is distributed across both layers. Instead, when $\lambda > 0$, the second layer $\mathbf{W}_2$ exhibits *lazy* learning, yielding an output representation $\mathbf{W}_2 \mathbf{W}_2^T$ of a weighted identity matrix. However, the first layer $\mathbf{W}_1$ still learns a *rich* representation of the hierarchy, albeit at a smaller scaling. Furthermore, rather than distributing this learning across both layers, in the $\lambda > 0$ case, all learning of the hierarchy occurs in the first layer, allowing it to more readily transfer this structure to the learning of a new feature (which only depends on the first layer). Thus, in this case, the 'shallowing'

of the network into the first layer is actually beneficial. Finally, we can also observe the opposite case when $\lambda < 0$. Here, *rich* learning happens in the second layer, while the first layer is *lazy* and learns to represent a weighted identity matrix. As such, these networks do not learn to transfer the hierarchy of different items to the new feature.

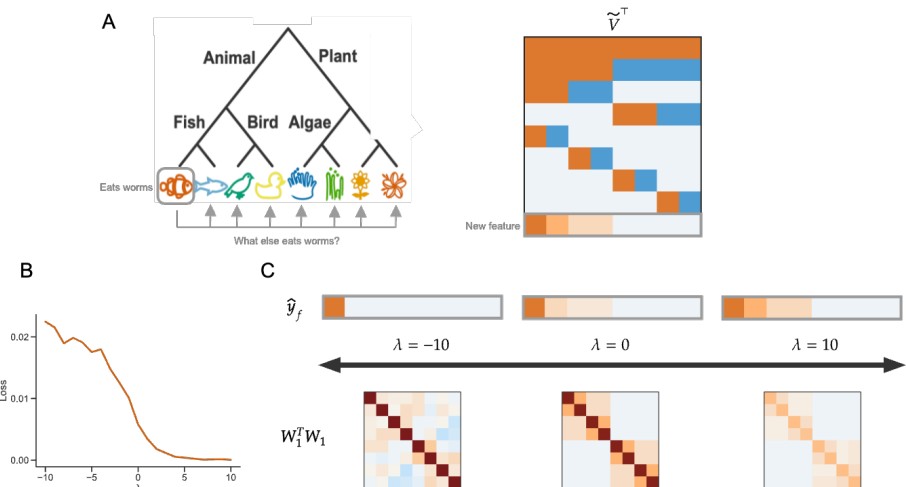

Figure 13: Transfer learning for different $\lambda$. **A** A new feature (such as 'eats worms') is introduced to the dataset after training on the hierarchical semantic learning task (Section 5). A randomly initialized row is added to $\mathbf{W}_2$ and trained on a single item with the new feature (for example, the goldfish), with the rest of the network frozen. The network is then tested on the transfer of the new feature to other items, such that items closer to the goldfish in the hierarchy are more likely to have the same feature. **B** The generalization loss on the untrained items with the new feature decreases as $\lambda$ increases. **C** As $\lambda$ increases positively, networks better transfer the hierarchical structure of the data to the representation of the new feature.

### D.4 Appendix: Finetuning

It is a common practice to pretrain neural networks on a large auxiliary task before fine-tuning them on a downstream task with limited samples. Despite the widespread use of this approach, the dynamics and outcomes of this method remain poorly understood. In our study, we provide a theoretical foundation for the empirical success of fine-tuning, aiming to improve our understanding of how performance depends on the initialization. We're interested in understanding how changing the $\lambda$-balancedness after pre-training may impact fine-tuning on a new dataset. We use $\lambda_{PT}$ to denote how networks are first initialized prior to pretraining, and $\lambda_{FT}$ to how they are *re-balanced* after pre-training and before fine-tuning on a new task. Similar to the previous section, we first train each network (for different values of $-10 \le \lambda_{PT} \le 10$) on the hierarchical semantic learning task. We then change the $\lambda$-balancedness of each network (for different values of $-10 \le \lambda_{FT} \le 10$) and retrain on a new dataset to observe how this impacts fine-tuning for different values and compare to networks that are not re-balanced to some $\lambda_{FT}$ ($\lambda_{FT} = \emptyset$) after initial pre-training.

In particular, to reset the $\lambda$-balancedness of a pretrained network to $\lambda_{FT}$, we rescale the singular values of each layer ($\mathbf{S}_1, \mathbf{S}_2$) using the singular values of the entire network function ($\mathbf{S} = \mathbf{U}^T \mathbf{W}_2 \mathbf{W}_1 \mathbf{V}$), while keeping the left and right singular vectors of the network unchanged.

We consider three different tasks to fine-tune the networks on. In the first, we add an existing feature from one item to another item in the hierarchy in order to disrupt the structure of the left and right singular vectors. In the second task, we consider the same reversal learning task discussed in Appendix D.2, where one column of the right singular vectors are reversed such that $-\boldsymbol{v}_i = \tilde{\boldsymbol{v}}_i$. Finally, we consider a scaled version of the hierarchy where each singular value is scaled by 2.

Across all the tasks we consider, we consistently find that fine-tuning performance improves as networks are re-balanced to larger values of $\lambda_{FT}$ and, conversely, decreases as $\lambda_{FT}$ approaches 0. Networks re-balanced to $\lambda_{FT} = 0$ also learn more slowly compared to $|\lambda_{FT}| > 0$. Interestingly, when studying networks that are *not* re-balanced prior to finetuning ($\lambda_{FT} = \emptyset$; but *are* first initial-

ized prior to pretraining to $\lambda_{PT}$), we see that they perform similarly on the new tasks to networks that are re-balanced to $\lambda_{FT} = \lambda_{PT}$.

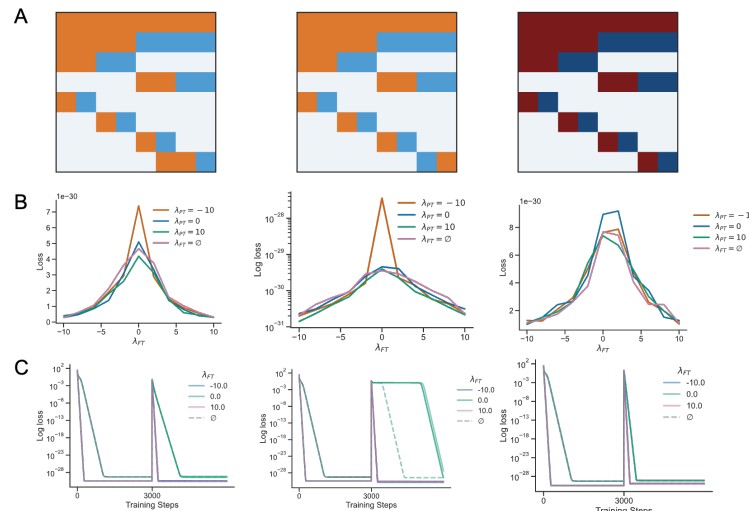

Figure 14: Fine-tuning performance on three tasks for different re-balancing $\lambda_{FT}$. **A** After training on the hierarchical semantic learning task (Section 5), networks are re-balanced and trained on one of three tasks: adding an existing feature from one item to another item in the hierarchy (*left*), the reversal learning task in Appendix D.2 (*center*), or a scaled version of the hierarchy where each singular value is scaled by 2 (*right*). **B** Change in loss on the new task across different $\lambda_{FT}$ for different $\lambda_{PT}$. As $\lambda_{FT}$ approaches 0, the loss on the new task increases across all $\lambda_{PT}$. Interestingly, networks that are not re-balanced prior to fine-tuning ($\lambda_{FT} = \emptyset$) perform similarly to networks that are re-balanced to the same values ($\lambda_{FT} = \lambda_{PT}$). **C** Dynamics of the loss across the first pre-training task and the new fine-tuning task. Networks re-balanced to $\lambda_{FT} = 0$ consistently learn slower across all tasks compared to networks that are re-balanced to larger magnitude values ($|\lambda_{FT}| > 0$ )

In this work, we derive the precise dynamics of two-layer linear. While straightforward in design, these architectures are foundational in numerous machine learning applications, particularly in the implementation of Low Rank Adapters (LoRA)Hu et al. (2022). A key innovation in LoRA is to parameterize the update of a large weight matrix $\mathbf{W} \in \mathbb{R}^{d \times d}$ within a language model as $\Delta \mathbf{W} = \mathbf{AB}$, the product of two low-rank matrices $\mathbf{A} \in \mathbb{R}^{d \times r}$ and $\mathbf{B} \in \mathbb{R}^{r \times d}$, where only $\mathbf{A}$ and $\mathbf{B}$ are trained. To ensure $\Delta \mathbf{W} = 0$ at initialization, it is standard practice to initialize $\mathbf{A} \sim \mathcal{N}(0, \sigma^2)$ and $\mathbf{B} = 0$ ( Hu et al. (2022); Hayou et al. (2024). It is noteworthy that this parameterization, $\Delta \mathbf{W} = \mathbf{AB}$, effectively embeds a two-layer linear network within the language model. When $r \ll d$, this initialization scheme approximately adheres to our $\lambda$-balanced condition, with $\sigma^2$ playing the role of the balance parameter $\lambda$. Investigating how the initialization scale of $\mathbf{A}$ and $\mathbf{B}$ influences fine-tuning dynamics under LoRA, and connecting this to our work on $\lambda$-balanced two-layer linear networks and their role in feature learning, represents an intriguing avenue for future exploration. This perspective aligns with recent studies suggesting that low-rank fine-tuning operates in a "lazy" regime, as well as work examining how the initialization of $\mathbf{A}$ or $\mathbf{B}$ affects fine-tuning performance Malladi et al. (2023); Hayou et al. (2024). Our framework offers a potential bridge to understanding these phenomena more comprehensively. While a detailed exploration of fine-tuning performance lies beyond the scope of this work, it remains an important direction for future research.

# E IMPLEMENTATION AND SIMULATIONS

The details of the simulation studies are described as follows. Specifically, $N_i$, $N_h$, and $N_o$ represent the dimensions of the input, hidden layer, and output (target), respectively. The total number of training samples is denoted by $N$, and the learning rate is defined as $\eta = \frac{1}{\tau}$.

### E.1 LAMBDA-BALANCED WEIGHT INITIALIZATION

In practice, to initialize the network with lambda-balanced weights, we use Algorithm E.1. In this algorithm, $\alpha$ serves as a scaling factor that controls the variance of the weights, allowing for adjustments between smaller and larger weight initializations.

---

**Algorithm 1** Get $\lambda$-*balanced*

---
1: **function** GET_LAMBDA_BALANCED($\lambda$, $in\_dim$, $hidden\_dim$, $out\_dim$, $\sigma = 1$)
2:     **if** $out\_dim > in\_dim$ and $\lambda < 0$ **then**
3:         **raise** Exception('Lambda must be positive if out_dim ¿ in_dim')
4:     **end if**
5:     **if** $in\_dim > out\_dim$ and $\lambda > 0$ **then**
6:         **raise** Exception('Lambda must be positive if in_dim ¿ out_dim')
7:     **end if**
8:     **if** $hidden\_dim < \min(in\_dim, out\_dim)$ **then**
9:         **raise** Exception('Network cannot be bottlenecked')
10:     **end if**
11:     **if** $hidden\_dim > \max(in\_dim, out\_dim)$ and $\lambda \neq 0$ **then**
12:         **raise** Exception('hidden_dim cannot be the largest dimension if lambda is not 0')
13:     **end if**
14:     $W_1 \leftarrow \sigma \cdot$ random normal matrix$(hidden\_dim, in\_dim)$
15:     $W_2 \leftarrow \sigma \cdot$ random normal matrix$(out\_dim, hidden\_dim)$
16:     $[U, S, Vt] \leftarrow \text{SVD}(W_2 \cdot W_1)$
17:     $R \leftarrow$ random orthonormal matrix$(hidden\_dim)$
18:     $S2_{equal\_dim} \leftarrow \sqrt{\left(\sqrt{\lambda^2 + 4 \cdot S^2} + \lambda\right)/2}$
19:     $S1_{equal\_dim} \leftarrow \sqrt{\left(\sqrt{\lambda^2 + 4 \cdot S^2} - \lambda\right)/2}$
20:     **if** $out\_dim > in\_dim$ **then**
21:         $S2 \leftarrow \begin{bmatrix} S2_{equal\_dim} & 0 \\ 0 & 0_{hidden\_dim-in\_dim} \end{bmatrix}$
22:         $S1 \leftarrow \begin{bmatrix} S1_{equal\_dim} \\ 0 \end{bmatrix}$
23:     **else if** $in\_dim > out\_dim$ **then**
24:         $S1 \leftarrow \begin{bmatrix} S1_{equal\_dim} & 0 \\ 0 & 0_{hidden\_dim-out\_dim} \end{bmatrix}$
25:         $S2 \leftarrow \begin{bmatrix} S2_{equal\_dim} & 0 \end{bmatrix}$
26:     **end if**
27:     $init\_W_2 \leftarrow U \cdot S2 \cdot R^T$
28:     $init\_W_1 \leftarrow R \cdot S1 \cdot Vt$
29:     **return** $(init\_W_1, init\_W_2)$
30: **end function**

---

### E.2 TASKS

In the following, we describe the different tasks that are used throughout the simulation studies.

#### E.2.1 RANDOM REGRESSION TASK

In the random regression task, the inputs $\mathbf{X} \in \mathbb{R}^{N_i \times N}$ are generated from a standard normal distribution, $\mathbf{X} \sim \mathcal{N}(\mu = 0, \sigma = 1)$. The input data $\mathbf{X}$ is then whitened to satisfy $\frac{1}{N}\mathbf{X}\mathbf{X}^T = \mathbf{I}$. The target values $\mathbf{Y} \in \mathbb{R}^{N_o \times N}$ are independently sampled from a normal distribution with variance scaled according to the number of output nodes, $\mathbf{Y} \sim \mathcal{N}(\mu = 0, \alpha = \frac{1}{\sqrt{N_o}})$. Consequently, the network inputs and target values are uncorrelated Gaussian noise, implying that a linear solution may not always exist.

#### E.2.2 SEMANTIC HIERARCHY

We use the same task as in Braun et al. (2022) and modify it to match the theoretical dynamics. The modification ensures that the inputs are whitened. In the semantic hierarchy task, input items are represented as one-hot vectors, i.e., $\mathbf{X} = \frac{\mathbf{I}}{8}$. The corresponding target vectors, $\mathbf{y}_i$, encode the

item's position within the hierarchical tree. Specifically, a value of 1 indicates that the item is a left child of a node, $-1$ denotes a right child, and 0 indicates that the item is not a child of that node. For example, consider the blue fish: it is a blue fish, a left child of the root node, a left child of the animal node, not part of the plant branch, a right child of the fish node, and not part of the bird, algae, or flower branches, resulting in the label $[1, 1, 1, 0, -1, 0, 0, 0]$. The labels for all objects in the semantic tree, as shown in Figure 4 A, are given by:

$$\mathbf{Y} = 8 * \begin{bmatrix} 1 & 1 & 1 & 1 & 1 & 1 & 1 & 1 \\ 1 & 1 & 1 & 1 & -1 & -1 & -1 & -1 \\ 1 & 1 & -1 & -1 & 0 & 0 & 0 & 0 \\ 0 & 0 & 0 & 0 & 1 & 1 & -1 & -1 \\ 1 & -1 & 0 & 0 & 0 & 0 & 0 & 0 \\ 0 & 0 & 1 & -1 & 0 & 0 & 0 & 0 \\ 0 & 0 & 0 & 0 & 1 & -1 & 0 & 0 \\ 0 & 0 & 0 & 0 & 0 & 0 & 1 & -1 \end{bmatrix}. \tag{188}$$

The singular value decomposition (SVD) of the corresponding correlation matrix, $\tilde{\mathbf{\Sigma}}^{yx}$, is not unique due to identical singular values: the first two, the third and fourth, and the last four values are the same. To align the numerical and analytical solutions, this permutation invariance is addressed by adding a small perturbation to each column $\mathbf{y}_i$, for $i \in 1, ..., N$, of the labels:

$$\mathbf{y}_i = \mathbf{y}_i \cdot \left(1 + \frac{0.1}{i}\right), \tag{189}$$

resulting in singular values that are nearly, but not exactly, identical.

### E.3 FIGURE 1

Panels B illustrates three simulations conducted on the same task with varying initial $\lambda$-balanced weights respectively $\lambda = -2$, $\lambda = 0$, $\lambda = 2$. The regression task parameters were set with ($\sigma = \sqrt{10}$). The network architecture consisted of $N_i = 3$, $N_h = 2$, $N_o = 2$, with a learning rate of $\eta = 0.0002$. The batch size is $N = 10$. The zero-balanced weights are initialized with variance $\sigma = 0.00001$. The lambda-balanced network are initialized with $sigmaxy = \sqrt{1}$ of a random regression task with same architecture.
On Panel C , we plot the ballancedness $\mathbf{W}_2(0)^T \mathbf{W}_2(0) - \mathbf{W}_1(0) \mathbf{W}_1(0)^T$ for a two layer network initialised with Lecun initialization with dimension $N_i = 40$ ,$N_h = 120$ ,$N_o = 250$

### E.4 FIGURE 2

Panel A, B, C illustrates three simulations conducted on the same task with varying initial $\lambda$-balanced weights respectively $\lambda = -2$, $\lambda = 0$, $\lambda = 2$ according to the initialization scheme described in E.7. The regression task parameters were set with ($\sigma = \sqrt{10}$). The network architecture consisted of $N_i = 3$, $N_h = 2$, $N_o = 2$ with a learning rate of $\eta = 0.0002$. The batch size is $N = 10$. The zero-balanced weights are initialized with variance $\sigma = 0.00001$. The lambda-balanced network are initialized with $sigmaxy = \sqrt{1}$ of a random regression task with same architecture.

### E.5 FIGURE 3

Panel A, B, C illustrates three simulations conducted on the same task with varying initial $\lambda$-balanced weights respectively $\lambda = -2$, $\lambda = 0$, $\lambda = 2$ according to the initialization scheme described in E.7. The regression task parameters were set with ($\sigma = \sqrt{12}$). The network architecture consisted of $N_i = 3$, $N_h = 3$, $N_o = 3$ with a learning rate of $\eta = 0.0002$. The batch size is $N = 5$. The zero-balanced weights are initialized with variance $\sigma = 0.0009$. The lambda-balanced network are initialized with $sigmaxy = \sqrt{12}$ of a random regression task with same architecture.

### E.6 FIGURE 4

In Panel A presents a semantic learning task with the SVD of the input-output correlation matrix of the task. $U$ and $V$ represent the singular vectors, and $S$ contains the singular values. This decomposition allows us to compute the respective RSMs as $USU^\top$ for the input and $VSV^\top$ for the output task. The rows and columns in the SVD and RSMs are ordered identically to the items in the hierarchical tree.

The results in Panel B display simulation outcomes, while Panel C presents theoretical input and output representation matrices at convergence for a network trained on the semantic task described in Braun et al. (2022); Saxe et al. (2013),. These matrices are generated using varying initial $\lambda$-balanced weights set at $\lambda = -2$, $\lambda = 0$, and $\lambda = 2$, following the initialization scheme outlined in E.7. The network architecture includes $N_i = 8$, $N_h = 8$, and $N_o = 8$ with a learning rate of $\eta = 0.001$ and a batch size of $N = 8$. Zero-balanced weights are initialized with a variance of $\sigma = 0.00001$, while $\lambda$-balanced networks are initialized with $\sigma_{xy} = \sqrt{1}$ based on a random regression task with the same architecture.

Panel D illustrates results from running the same task and network configuration but initialized with randomly large weights having a variance of $\sigma = 1$.

In panel E, we trained a two-layer linear network with $N_i = N_h = N_o = 4$ on a random regression task for $\lambda \in [-5, -4, -3, -2, -1, 0, 1, 2, 3, 4, 5]$ to convergence. Subsequently, we added Gaussian noise with $\mu = 0, \sigma \in [0, 0.5, 1]$ to the inputs (top panel) or synaptic weights (bottom panel) and calculated the expected mean squared error.

## E.7  FIGURE 5

Panel A illustrates schematic representations of the network architectures considered: from left to right, a funnel network ($N_i = 4$, $N_h = 2$, $N_o = 2$), a square network ($N_i = 4$, $N_h = 4$, $N_o = 4$), and an inverted-funnel network ($N_i = 2$, $N_h = 2$, $N_o = 4$).

Panel B shows the Neural Tangent Kernel (NTK) distance from initialization, as defined in Fort et al. (2020), across the three architectures shown schematically. The kernel distance is calculated as:

$$S(t) = 1 - \frac{\langle K_0, K_t \rangle}{\|K_0\|_F \|K_t\|_F}.$$

The simulations conducted on the same task with eleven varying initial $\lambda$-balanced weights in $[-9, 9]$. The regression task parameters were set with ($\sigma = \sqrt{3}$). The task has batch size $N = 10$. The network has with a learning rate of $\eta = 0.01$. The lambda-balanced network are initialized with $\sigma xy = \sqrt{1}$ of a random regression task.

Panel C shows the Neural Tangent Kernel (NTK) distance from initialization for the funnel architectures shown schematically with dimensions $N_i = 3$, $N_h = 2$, and $N_o = 2$. The simulations conducted on the same task with twenty one varying initial $\lambda$-balanced weights in $[-9, 9]$. The regression task parameters were set with ($\sigma = \sqrt{3}$). The task has batch size $N = 30$. The network has with a learning rate of $\eta = 0.002$. The lambda-balanced network are initialized with $\sigma xy = \sqrt{1}$ of a random regression task.

