# OpenReview forum: "From Lazy to Rich: Exact Learning Dynamics in Deep Linear Networks"
_ICLR.cc/2025/Conference — ICLR 2025 Poster_

### Official Review · Reviewer_5Z1o · 2024-11-01

**Soundness:** 4
**Presentation:** 3
**Contribution:** 3
**Rating:** 8
**Confidence:** 4

**Summary:**

This study derives exact solutions (Theorem 4.3) to examine how initialization --- particularly the relative scale of weights across layers --- influences rich versus lazy learning in linear feedforward neural networks. The study relaxes several assumptions from prior work, notably the balanced assumption in Braun et al., which is a special case in here when $\lambda=0$. Extending this, the authors show in Theorem C.4 that the network transitions into a rich regime as $\lambda$ approaches 0, and into a lazy regime as $\lambda$ approaches infinity. Overall, this work significantly contributes to the ongoing investigation into how various network parameters influence rich versus lazy learning regimes, with demonstrated relevance to both neuroscience and deep learning.

**Strengths:**

•	This work is novel and timely.

•	Strong theoretical foundation: the approach extends a previous theoretical framework (Braun et al., 2022) to derive exact solutions for NTK, representation similarity, gradient flow, and loss. This allows for precise determination of how initialization (specifically, the $\lambda$ parameter) influences representation and learning dynamics under the given assumptions. However, I did not have the bandwidth to verify the proofs in the appendix, which affects my confidence score.

•	Insightful experiments demonstrating how $\lambda$ interpolates between learning regimes, with applications to both neuroscience and machine learning.

•	The manuscript is well-written, with a comprehensive literature review.

**Weaknesses:**

•	This work relies on a list of assumptions and focuses on a simple two-layer linear feedforward network, which deviates from real-world settings. However, this didn’t significantly impact my score, as the assumptions are already more relaxed compared to previous works, and the authors have adequately addressed these limitations in the discussion.

•	I wish there were more intuitive explanations of how \(\lambda\) interpolates between learning regimes. The authors provide some results in this direction, such as Theorem 5.1, which shows that under different limits of \(\lambda\), the transition function converges to sigmoidal (\(\lambda \to 0\)) and exponential (\(\lambda \to \pm \infty\)). However, it would be helpful to have more insight into how the limit of \(\lambda \to \pm \infty\) leads to a task-agnostic identity representation.

•	I’m not sure if I have a good understanding of how the relative and absolute scales interact. While the authors demonstrated that in the extreme case where \(\lambda \to 0\), the rich regime occurs regardless of the absolute scale, I’m less certain about the case where \(\lambda \to \infty\), as the weight norm of one of the layers would presumably become very large. Large initialization can lead to lazy learning, so I’m unsure if it’s possible to decouple the contributions of \(\lambda\)-balanced initialization from large weight initialization. I believe some control experiments with large \(\lambda\) values using random (unbalanced) initialization but with the same \(W_1\) and \(W_2\) norms as the balanced case would help distinguish the two.

**Questions:**

•	It would be interesting to understand more how this relative scale interact with other knobs that impact learning regimes. For instance, could you provide more discussion on how the relative scale and absolute scale of the weights interact? See the weakness point above.

•	Could you offer more intuition on how \(\lambda \to \infty\) results in an identity representation (i.e., a task-agnostic representation)?

---

> ### Author Response · Authors · 2024-11-20
> **Detailed reply to reviewer 5Z1o**
>
> We are grateful for your thorough review and the time you dedicated to suggesting improvements for our study. We are pleased to hear that you view our work as a timely and significant contribution to the field. Your positive remarks on the paper’s strong theoretical foundation, novelty, clarity, and relevance are highly appreciated. We will address each of the weaknesses and questions you raised individually.
>
> **Assumptions and focus on a simple two-layer linear network**
> We acknowledge the limitations associated with the assumptions and the two-layer linear setting highlighted by the reviewer. In our general response, we aim to contrast these with prior literature, showcasing how they extend the scope of previous studies. We further demonstrate that relaxing these assumptions still leads to intriguing and qualitatively equivalent regimes.
>
> **More intuitive explanations of how ($\lambda$) interpolates between learning regimes**
> The $\lambda$-balanced condition imposes a relationship between the singular values of the two weight matrices. Specifically, if $W_2$ and $W_1$ are $\lambda$-balanced and satisfy $W_2 W_1 = \Sigma_{yx}$, then for arbitrary singular values $a_i$, $b_i$, and $s_i$ (singular values of $W_2$, $W_1$, and $\Sigma_{yx}$ respectively), the following relations hold: $ a_i^2 - b_i^2 = \lambda, \quad a_i \cdot b_i = s_i. $ As $\lambda$ increases, the value of $b_i$ must decrease. In the limit as $\lambda \to \infty$, $a_i^2 \to \lambda$ and $b_i^2 \to 0$. From the first equation, when $b_i^2 \to 0$, $a_i^2 \to \lambda$. Since these equations apply to all singular values of the matrices, it follows that for all $i$, $a_i^2 \to \lambda$, leading to the conclusion that $ W_2^T W_2 = \lambda I, $ as expected. Consequently, the task representation becomes task-agnostic in $W_1$. The intuition here is that the weights are constrained by the need to fit the data, which bounds their overall norms. The $\lambda$-balanced condition further specifies a relationship between these norms, and as $|\lambda|$ increases, this constraint tightens, driving $W_2$ toward the identity matrix. In this regime, the network behaves similarly to a shallow network, with $\lambda$ acting as a toggle between deep and shallow learning dynamics. In Appendix C of the paper, we rigorously prove this behavior and further demonstrate that, as a result, the Neural Tangent Kernel (NTK) remains static in this regime, confirming that the network operates in the lazy learning regime. We included this discussion in Appendix C 2.4 to improve clarity.
>
> **Scale with relative scale**
> You are correct that the relative scale and absolute scale interact in non-trivial ways. While our study primarily focuses on the effects of relative scale, the influence of absolute scale is inherently embedded within our framework through the definitions of $B$, $C$, and $D$ (see Equations 10, 11, and 12). However, this influence is not immediately apparent from the main theorem. A clearer distinction between the roles of relative and absolute scale can be observed in Theorem 5.1. Specifically, we investigated how $\lambda$, the relative scale parameter, governs the transition between sigmoidal and exponential dynamical regimes. A similar argument applies to absolute scale, which appears explicitly as $s_\alpha(0)$ in these equations. Consider the case when $\lambda = 0$. The dynamics of $s_\alpha$ reduce to the classical solution of the Bernoulli differential equation. In the limiting case where $s_\alpha(0) \to 0$, the system exhibits classic sigmoidal dynamics (characteristic of the rich regime), whereas the limit $s_\alpha(0) \to \infty$ yields exponential dynamics  (characteristic of the lazy regime). This interplay between relative and absolute scales underscores their critical roles in shaping the system's behavior. A straightforward intuition can also be gained by considering the scalar case where $N_i = N_h = N_o = 1$. In this scenario, it is easy to ensure that $w_1^2 = w_2^2$ satisfying $\lambda = 0$ while allowing for different absolute scales. For instance, $w_1 = w_2 = 0.001$ or $w_1 = w_2 = 5$. In such cases, the absolute scale is clearly decoupled from the relative scale. Altogether, the absolute scale and relative scale of the weights play a critical role in describing the phase portrait of the learning regime, as first demonstrated in the Kunin et al., 2024 paper on ReLU networks. Thank you for this question, this is an insightful point; we will add a discussion similar to the one above in Appendix A.4.
>
>
> **Questions**
> - Question a: See above
> - Question b: Please refer to the response above as well as the response to reviewer ifg2 question 4.
> .

---

> ### Comment · Reviewer_5Z1o · 2024-11-22
> **Thank you for the detailed rebuttal**
>
> I commend the authors for their additional clear explanations, expanded applications, and detailed responses to the reviewers' questions. The explanations regarding the intuition and interaction between absolute and relative scales are particularly helpful.
>
> That said, I share reviewer ifg2's concerns about the technical and theoretical novelty of this work. The method employed in this paper has largely been established in Braun et al. (2022), with this work extending it to the
> $\lambda$-balanced case. Nonetheless, I believe the paper should still be accepted due to its technical soundness and the novel insights derived from the relaxation.
>
> I would like to raise my score to a 7, but this does not appear to be an option in the ICLR score dropdown (the closest choices being 6 and 8). Thus, I will maintain my current score, which is already above the acceptance threshold. That said, I have increased the soundness score from 3 to 4 in light of the authors' rebuttal.

---

> > ### Author Response · Authors · 2024-11-26
> >
> > We sincerely appreciate your positive feedback and your expressed support for the acceptance of this paper.
> >
> > We would like to highlight that, while the method builds upon Braun et al. (2022), addressing the balanced case required significant and non-trivial modifications, as outlined in Appendix B.4. The resulting solution is substantially different and showcases the unique contributions of this work.
> >
> > We are committed to making the most of the discussion period. Thus, if the reviewer has any further suggestions which would improve their evaluation of the paper or strengthen their confidence in the merit of our work, we would gladly aim to incorporate these.
> >
> > Furthermore, given that the current scoring system does not include a 7 (accept) as an option— but instead 8 (accept) signifies acceptance—we kindly request that you consider adjusting your score to clearly reflect your endorsement of the paper’s acceptance. It would be unfortunate for this work to be disadvantaged due to the scoring system. We thank you or increasing the soundness to reflect our effort. Thank you for your thoughtful consideration and active participation in the review process.
> >
> > The Authors

---

> > > ### Author Response · Authors · 2024-11-28
> > > **Further response to Official Comment**
> > >
> > > Once again, we sincerely thank the reviewer for their valuable feedback. We have uploaded an updated manuscript that includes new results further addressing the question on the scale and relative scale. We have included the phase portrait of the loss and the kernel distance of the Neural Tangent Kernel (NTK) from initialization as functions of the scale and relative scale (see Figures 8, 9, and 10 in Appendix section A.4). These portraits, plotted at various training steps, examine how absolute and relative scales influence learning dynamics. We repeated the experiment for three architectures: a square network, a funnel network, and an anti-funnel network. The main takeaways are:
> > > - The scale and the relative scale non-trivially influence the learning dynamics as characterized by our theoretical solution.
> > > - Regardless of the scale, $\lambda$ = 0 will always lead to a rich regime in square linear networks.
> > > - For large $\lambda$ values, the network goes into a lazy regime in square linear networks, even at the small scale.
> > >
> > > We want to reiterate that our proposed solution significantly surpasses the previous approach presented by Braun et al. (2022), as the latter **could not** capture the effects above, being limited to the analysis of the restrictive case of $\lambda$ = 0. Additionally, we would like to respectfully point out that some of the weaknesses highlighted by Reviewer ifg2 are incorrect for the following reasons:
> > > - Our method does not impose stricter assumptions w.r.t. Braun et al. (2022), as both methods require similar assumptions of whitened inputs.
> > > - Our approach does not rely on the assumption of task alignment.
> > > - Our proposed solution differs from Braun et al. (2022) by relaxing three key assumptions: **invertibility of B, full rank, and $\lambda$-balancedness**. These relaxations stand as significant technical and theoretical novelty w.r.t. Braun et al. (2022), allowing us to understand the dynamics of the learning regimes.
> > >
> > > We are committed to making the most of the discussion period. We remain available for any further questions. Thank you for your active participation in the review process.

---

> > > > ### Comment · Reviewer_5Z1o · 2024-11-28
> > > > **Score raised**
> > > >
> > > > I appreciate the authors' insightful new results and additional explanations, which effectively highlight the nontrivial extension from Braun et al. (2022) and the novel insights gained through this work. The appendix of the manuscript is exceptionally strong. To enhance accessibility, it would be valuable for the authors to summarize more key insights from the appendix in the main text or provide more references to them in prominent sections. I trust the authors will make more effort on this in the camera-ready version, should the paper be accepted, and I have raised my score accordingly.

---

> > > > > ### Author Response · Authors · 2024-11-29
> > > > >
> > > > > We sincerely thank the reviewer for their positive and constructive feedback. We agree that summarizing key insights from the appendix in the main text and including clearer references will enhance accessibility. These improvements will be incorporated into the manuscript. Once again, thank you for your thoughtful input and engagement in the review process.

---

### Official Review · Reviewer_ifg2 · 2024-11-02

**Soundness:** 3
**Presentation:** 4
**Contribution:** 3
**Rating:** 6
**Confidence:** 4

**Summary:**

This work extends previous results on the exact dynamics of deep linear networks to include the lambda-balanced initialization conditions, where lambda is a tunable parameter that allows for exploration of a full range of dynamics from lazy to rich learning. By detailed analysis of the result, the authors showed that the transition from the lazy regime to the rich regime depends on the complex interaction of multiple factors, including the balance parameter lambda and the network architecture. This paper then also discusses potential application of the results on more complicated learning paradigms such as continual learning, reversal learning and transfer learning.

**Strengths:**

1.	The paper is very well structured and the theoretical results are clearly written.
2.	It is a novel and interesting idea to model the range of dynamics from lazy to rich with the balance parameter \lambda.
3.	The authors analyzed their theoretical results on interesting semantic task examples, and showed multiple implications of their theory on other learning paradigms.

**Weaknesses:**

1.	The technical and theoretical novelty of this work is limited, the method used in this paper has mostly been established in Braun et. al. 2022, this work extends to the \lambda-balanced case, which is more general. However, the extension is achieved by enforcing a stricter assumption that the input is whitened, limiting the applicability of the results.
2.	This paper makes several very strong assumptions such as task-aligned initialization, whitened inputs, linear networks, etc. It would be nice if the authors can show even some empirical evidence that these assumptions may be somewhat relaxed while the main results still qualitatively hold.
3.	The applications are only briefly discussed. I think the paper would benefit from making the theoretical results in section 4 more concise and moving some of the application results (transfer learning for example I think is very interesting) to main text.

**Questions:**

1.	A minor point: over-parameterization is commonly referring to the relation between the amount of parameters in the network (capacity of the network) vs. the amount of parameters needed to achieve zero training loss. I believe this may be different from ‘N_h > min(N_i, N_o)’?
2.	The assumption that the input data is whitened is relatively strong and not very applicable to realistic settings. Do you have any empirical evidence demonstrating that your observation at least qualitatively still holds for benchmark datasets without necessarily whitening the data?
3.	This may be just a typo. In Fig.2 b, in the caption it says lambda=0.001 but in the figure lambda=0.
4.	In previous works concerning lazy and rich dynamics, especially in infinitely wide networks, there’s usually the assumption that N goes to infinity and depending on how initialization scales with N the network resides in either the lazy or the rich regime. Here there doesn’t seem to be any assumptions about large N or the scaling of the initialization. Specifically since Nh=min(Ni, No), it seems that N is finite. However, it seems that the authors are able to get fixed NTK in the lazy regime without the assumption of large N in section 5. Is this specific to the linear network? Can you comment on how your results may scale to larger dimensional networks, and on the relation of your results and the lazy and rich regimes in the infinite width limit?
5.	In Fig.4 b the simulation w1w1 and w2w2 labels seem to be reversed compared to the theory.
6.	In Fig.5, how does the ratio between Nh and Ni (or Nh and No) play a role here?

---

> ### Author Response · Authors · 2024-11-20
> **Detailed reply to reviewer ifg2**
>
> Thank you for taking the time to review our paper carefully and point out areas that could benefit further clarification. We're glad to hear your positive feedback on the paper's novelty, clarity, and significance. We will address each of the concerns you highlighted in detail to strengthen your confidence in the importance of our research.
>
> **Weaknesses**
> - **The technical and theoretical novelty** This work builds on the methodology introduced by Braun et al. (2022) while making substantial advancements and extensions. The updated version eliminates several critical assumptions that constrained the earlier study, including full-rank conditions, the invertibility of the variable B, and the extension to the $\lambda$-balanced condition. Importantly, we emphasize that our general solution does not rely on task-aligned initialization (Theorem 5.1); this assumption is only later used to gain further intuition about the dynamics of singular values and the transition between the rich and lazy regimes. By removing these limitations, this work not only extends the scope of prior research but also broadens the analytical framework. In the general response to reviewers, we detail the unique contributions that distinguish our approach from those of previous studies. Additionally, we show that multiple assumptions can be relaxed without compromising the qualitative results of the primary findings, thereby addressing the first two weaknesses identified.
>
> - **Application** We extend the discussion of the applications in the new version of the paper to highlight their relevance. For instance, we delve deeper into how, in the transfer learning setting, increasing $\lambda$ enables networks to more effectively transfer the hierarchical structure to new features for untrained items. Additionally, we introduce a new application focused on fine-tuning dynamics and highlight its connection to LoRA fine-tuning techniques in large language models.

---

> ### Author Response · Authors · 2024-11-20
> **Detailed reply to reviewer ifg2**
>
> **Questions**
>
> 1. You raise a valid point. In the case of a linear network, the minimum required to learn a task in the most general sense is at least one layer with $ N_i \times N_o$ parameters. As $N_h > min(N_i, N_o)$, it ensures that the network exceeds this parameter count, thus fitting the definition of overparameterization. However, we acknowledge that this term can be confusing, as it is not commonly used in this way within practical machine learning contexts. We will adopt the simpler definition $N_h > min(N_i, N_o)$ to avoid ambiguity.
> 2. We respond to this point above in the discussion of the assumptions.
> 3. Thank you for pointing out the typo in Fig. 2. $\lambda$ should be equal to zero. We have corrected it in the updated version of the paper.
> 4. The claim aligns with the understanding that we achieve a fixed NTK without involving a large width. As outlined in the theorem, this is attributed to the NTK converging to an identity matrix, where the dominant singular values of the weights approach a scaled identity proportional to  $\lambda$, and the corresponding weights scale inversely with  $\lambda$. Intuitively, in this setup, the larger weights function as an identity-like projection, while the smaller weights adapt and align. However, due to their relatively small scale compared to the larger weights, their contribution to the NTK remains negligible. (It is important to recall that the NTK is determined by the similarity matrix of the input and output representations.) In the nonlinear setting, this behavior is not expected to hold, as an additional factor comes into play in the computation of the NTK: the activation coefficients of the nonlinearity, as demonstrated in Kunin et al., 2024. In that case, large relative weight (large positive $\lambda$) leads to a rapid rich regime. In the infinite-width regime, where weights are initialized from a Gaussian distribution with large variance, averaging effects cause both input and output representations to approximate identity matrices. In this scenario, the network learns with minimal parameter variation, operating in the lazy regime with a fixed Neural Tangent Kernel (NTK). This behavior contrasts with the dynamics observed in the current setting since both input and output representations are task agnostic.
> 5. Thank you for bringing the typo in Figure 4 to our attention. We have updated the figure in the manuscript.
> 6. As Nh​ approaches the maximum of Ni​ and No​, the importance of the delayed rich setting decreases. This happens because, in the delayed rich setting, no least-squares solution exists within the network's span at initialization. In such situations, the network shifts into a delayed rich regime where  $\lambda$ approaches infinity, with the magnitude of $\lambda$ dictating the extent of the delay. Initially, the network demonstrates lazy behavior, attempting to fit the solution. However, as constraints require adjustments in its directions, the network transitions to a rich phase. Consequently, when the network's dimensions are closer to spanning the full space, the delayed rich regime becomes less relevant.

---

### Official Review · Reviewer_6KfS · 2024-11-04

**Soundness:** 4
**Presentation:** 4
**Contribution:** 3
**Rating:** 6
**Confidence:** 3

**Summary:**

The paper theoretically derived the exact solutions for the learning dynamic of $\lambda$-balanced initializations in two-layer linear networks and discussed the implications in continual learning etc.

**Strengths:**

Originality: The paper built on top of existing work by extending from zero-balanced condition into $\lambda$-balanced scenarios, leading to a continuum from lazy to rich regime in terms of weight structure of two consecutive layers.

Quality: Solid theoretical derivations backed with rich simulation results.

Clarity: The paper is very well-written.

Significance: The $\lambda$-balanced discussed in the paper covers from architecture shapes to initialization schemes and relate these structural properties to the learning regimes, which will be of interest to the neuroscience community as well as continual learning and beyond communities.

**Weaknesses:**

The only weakness of the paper is on the weak demonstration of applications.

**Questions:**

a. Could the authors comment on how they would expect the results to generalize to more complex tasks?

b. Although it is mentioned some choice of $\lambda$ may be beneficial to transfer learning due to induced lazy learning in the shallow layer (D3), it would be at least better to show a performance increase.

---

> ### Author Response · Authors · 2024-11-20
> **Detailed reply to reviewer 6KfS**
>
> Thank you for taking the time to carefully review our paper and point out areas that could benefit from further clarification. We appreciate your positive feedback on the paper’s quality, clarity and significance. We will address each of the concerns you highlighted in detail, with the aim of strengthening your confidence in the importance of our research.
>
> **Application**
> 	We extend the discussion of the applications in the new version of the paper to highlight their relevance. For instance, we delve deeper into how, in the transfer learning setting, increasing \(\lambda\) enables networks to more effectively transfer the hierarchical structure to new features for untrained items. Additionally, we introduce a new application focused on fine-tuning dynamics and highlight its connection to LoRA fine-tuning techniques in large language models.
>
> **Results to generalize to more complex tasks** In this work, initializations are task-agnostic, meaning they do not depend on the specific structure of the task. This observation holds true for any learning task with a well-defined input-output correlation. Beyond the linear network setting, prior work suggests that many of the phenomena we study also take place in non-linear networks. For example, the work by Kunin et al., 2024 demonstrates that positive $\lambda$ initialization enhances the interpretability of early layers in CNNs and reduces the time required for grokking in modular arithmetic tasks. Therefore, although more research is needed, we are cautiously optimistic that our results also shed some light on the dynamics of non-linear networks in complex tasks.
>
>
> **Transfer learning shows a performance increase.**
> In the updated version of the manuscript, we emphasize in the main text the improvements in transfer learning performance as a function of the loss. Specifically, we highlight the results presented in Figure 11 of the appendix, which demonstrate that the generalization loss on untrained items with the new feature decreases as a function of increasing $\lambda$.  Therefore, as $\lambda$ increases, networks more effectively transfer the hierarchical structure of the network to the new feature for untrained items, leading to an increase in generalization performance. This finding effectively shows that performance improves with increasing  $\lambda$.

---

### Author Response · Authors · 2024-11-20
**General response to the review**

We thank the reviewers for their detailed and thoughtful feedback. We sincerely appreciate the time and effort dedicated to reviewing our manuscript, as your input has been instrumental in improving its quality. We have carefully addressed each of your comments and provided linked responses for overlapping points. We have uploaded a revised manuscript and summarized the key updates bellow:

- **Further Simplified Assumptions:** We have relaxed and removed several assumptions, both compared to the first version of this paper and the framework presented in Braun et al., 2022. For convenience, we will copy and paste the relevant new segment from the revised paper below.

- **New Application:** We have introduced an application to analyze fine-tuning dynamics over time and expanded the discussion on the connection to Low Rank Adaptation (LoRA), which plays a critical role in modern machine learning, particularly for fine-tuning large pre-trained models. For convenience, we copy-past below the relevant new segment from the revised paper.

- **Expanded Application and Implications:**  We further discussed the applications currently present in the paper - continual learning, reversal learning and transfer learning - and provided a more comprehensive discussion of its implications in practical, real-world scenarios. Due to space limitations, we have included part of the discussion in the Appendix.

 - **Clarifications and Typographical Corrections:** All sections highlighted for clarification by the reviewers have been revised accordingly. Additionally, all identified typographical errors have been corrected.
We hope these revisions enhance the clarity and impact of our work and that you find our manuscript a valuable contribution to the field.

We look forward to any further feedback you may have!

**New Application: Fine-tuning**
It is a common practice to pre-train neural networks on a large auxiliary task before fine-tuning them on a downstream task with limited samples. Despite the widespread use of this approach, the dynamics and outcomes of this method remain poorly understood. In our study, we provide a theoretical foundation for the empirical success of fine-tuning, aiming to improve our understanding of how performance depends on the initialization. Specifically, we explore how changes in $\lambda$-balancedness after pretraining might influence fine-tuning on a new dataset in Appendix~D.4 on fine-tuning. Across all the tasks we consider, we consistently find that fine-tuning performance improves and converges more quickly as networks are re-balanced to larger values of $\lambda_{FT}$ and, conversely, decreases as $\lambda_{FT}$ approaches 0. In this work, we consider two-layer linear networks. While straightforward in design, these architectures are foundational in numerous machine learning applications, particularly in the implementation of Low Rank Adapters (LoRA) (Hu et al., 2021). A key innovation in LoRA is to parameterize the update of a large weight matrix $W \in \mathbb{R}^{d \times d}$ within a language model as $\\Delta W = AB,$ the product of two low-rank matrices $A \in \mathbb{R}^{d \times r}$ and $B \in \mathbb{R}^{r \times d}$, where only $A$ and $B$ are trained. To ensure $\Delta W = 0$ at initialization, it is standard practice to initialize $A \sim \mathcal{N}(0, \sigma^2)$ and $B = 0$ (Hu et al., 2021; Hayou et al., 2024). It is noteworthy that this parameterization, $\Delta W = AB$, effectively embeds a two-layer linear network within the language model. When $r \ll d$, this initialization scheme approximately adheres to our $\lambda$-balanced condition, with $\sigma^2$ playing the role of the balance parameter $\lambda$. Investigating how the initialization scale of $A$ and $B$ influences fine-tuning dynamics under LoRA and connecting this to our work on $\lambda$-balanced two-layer linear networks and their role in feature learning represents an intriguing avenue for future exploration. This perspective aligns with recent studies suggesting that low-rank fine-tuning operates in a "lazy" regime, as well as work examining how the initialization of $A$ or $B$ affects fine-tuning performance (Malladi et al., 2023; Hayou et al., 2024). Our framework offers a potential bridge to understanding these phenomena more comprehensively. While a detailed exploration of fine-tuning performance (as a function of initialization) lies beyond the scope of this work, it remains an important direction for future research.

---

> ### Author Response · Authors · 2024-11-20
>
> **Further Simplified Assumptions:**
>
> - **Whitened Input Assumption** The assumption of whitened inputs, though strong, is a commonly used simplification in analytical studies to enable the derivation of exact solutions, as shown in previous works (e.g., Fukumizu et al., 1998; Kunin et al., 2024; Saxe et al., 2014) and notably in Braun et al., 2022. While relaxing this assumption prevents the exact description of network dynamics, Kunin et al., 2024, examine the implicit bias of the training trajectory without relying on whitened inputs. Their findings demonstrate a similar quantitative dependence on $\lambda$, governing the implicit bias transition between rich and lazy regimes. Furthermore, recent advancements, such as the "decorrelated backpropagation" technique introduced by Dalm et al., 2024,  which whitens inputs during training, showing that optimizing for whitened inputs can actually be done in practice and improve efficiency in real-world applications. Importantly, this study highlights that in certain real-world  scenarios, whitening can provide a more optimal learning condition. These approaches emphasize the potential advantages of input whitening for downstream tasks, reinforcing the validity of our assumption.
>
> - **Input-Output Dimensionality**
> Previous works imposed specific dimensionality constraints. For example:
> Fukumizu et al., 1998 assumed equal input and output dimensions ($N_i = N_o$) while allowing a bottleneck in the hidden dimension ($N_h \leq N_i = N_o$).
> Braun et al., 2022  extended these solutions to cases with unequal input and output dimensions ($N_i \neq N_o$) but restricted bottleneck networks ($N_h = \min(N_i, N_o)$) and introduced an additional invertibility condition on  ${ B}$.
> In our work, we allow for unequal input and output $N_i \neq N_o$ and do not introduce an additional invertibility assumption. This flexibility expands the applicability of our framework to a wider range of architectures.
>
> - **Balancedness Assumption**
> A significant departure from prior works is the relaxation of the balancedness assumption:
> Earlier studies, such as Fukumizu et al., 1998 and Braun et al., 2022, assumed strict zero-balancedness $W_1W_1^T = W_2^T W_2$, which constrained the networks to the rich regime. Our approach generalizes this to $\lambda$-balancedness, enabling exploration of the continuum between the rich and lazy regimes. While some efforts, such as Tarmoun et al., 2021, have explored removing the zero-balanced constraint, their solutions were limited to unstable or mixed forms.
> In contrast, our methodology systematically studies different learning regimes by varying initialization properties, particularly through the relative scale parameter. This allows controlled transitions between regimes, advancing understanding of neural network behavior across the spectrum. Other studies, such as Kunin et al., 2024 and Xu et al., 2023, have also relaxed the balancedness assumption, though their analysis was restricted to single-output neuron settings.
> We emphasize the importance of this balanced quantity by rigorously proving that standard network initializations (e.g., LeCun initialization, He initialization) are $\lambda$-balanced in expectation, and in the infinite width limit they are balanced in probability (Appendix A.3). Furthermore, previous studies such as Kunin et al.,2024 have demonstrated that the relative scaling of $\lambda$ significantly impacts the learning regime in practical scenarios, highlighting the crucial role of dynamical studies of networks as a function of this parameter.
>
> - **Full Rank**
> Previous work by Braun et al., 2022, imposed a full-rank initialization condition, defined as $\text{rank}(W_1(0)W_2(0)) = N_i = N_o$. However, this assumption is not necessary in our framework.
>
> We hope this clarifies the contribution. We updated the manuscript accordingly to improve clarity.

---

### Author Response · Authors · 2024-11-26

Dear Area Chair and Reviewers,

We thank the reviewers again for their feedback. We have provided detailed responses to all reviewers’ concerns and a general summary of the revisions made. An updated version of the manuscript has been uploaded, with the changes highlighted in blue for clarity. We kindly request confirmation on whether our replies sufficiently address the reviewers’ concerns and, if so, ask you to consider updating your scores accordingly—particularly given the positive response to the rebuttal from Reviewer 5Z1o. Your feedback is crucial to improving the quality of the paper, and we would greatly appreciate your engagement before the deadline.  We remain available for any further questions or concerns during the remainder of the discussion period.
Thank you for your time and consideration.

Best regards,
The Authors

---

### Meta-Review · Area_Chair_WdLV · 2024-12-20

**Metareview:**

This paper analyzes the learning dynamics of two-layer linear networks and derives explicit solutions for gradient flow. In particular, it focuses on how $\lambda$-balancedness of initialization affects the learning regime (including lazy and rich regimes). The results have implications for continual learning, reversal learning, and transfer learning. The reviewers all acknowledge the novel contributions of this paper and unanimously lean toward acceptance.

**Additional Comments On Reviewer Discussion:**

The reviewers had some concerns about the novelty of the techniques, the implications of the results, and the simplifying assumptions. The authors addressed these concerns satisfactorily.

---

### Decision · Program_Chairs · 2025-01-22

Accept (Poster)